# On the Identifiability of Switching Dynamical Systems

## Abstract

In the realm of interpretability and out-of-distribution generalisation, the identifiability of latent variable models has emerged as a captivating field of inquiry. In this work, we delve into the identifiability of Switching Dynamical Systems, taking an initial stride toward extending identifiability analysis to sequential latent variable models. We first prove the identifiability of Markov Switching Models, which commonly serve as the prior distribution for the continuous latent variables in Switching Dynamical Systems. We present identification conditions for first-order Markov dependency structures, whose transition distribution is parametrised via non-linear Gaussians. We then establish the identifiability of the latent variables and non-linear mappings in Switching Dynamical Systems up to affine transformations, by leveraging identifiability analysis techniques from identifiable deep latent variable models. We finally develop estimation algorithms for identifiable Switching Dynamical Systems. Throughout empirical studies, we demonstrate the practicality of identifiable Switching Dynamical Systems for segmenting high-dimensional time series such as videos, and showcase the use of identifiable Markov Switching Models for regime-dependent causal discovery in climate data.

## 1 Introduction

State-space models (SSMs) are well-established sequence modelling techniques where their linear versions have been extensively studied (Lindgren, 1978; Poritz, 1982; Hamilton, 1989). Meanwhile, recurrent neural networks (Hochreiter & Schmidhuber, 1997; Cho et al., 2014) have gained high popularity for sequence modelling thanks to their abilities in capturing non-linear and long-term dependencies. Nevertheless, significant progress has been made on fusing neural networks with SSMs, with Gu et al. (2022) as one of the latest examples. Many of these advances focus on building sequential latent variable models (LVMs) as flexible deep generative models (Chung et al., 2015; Li & Mandt, 2018; Babaeizadeh et al., 2018; Saxena et al., 2021), where SSMs have been incorporated as latent dynamical priors (Linderman et al., 2016; 2017; Fraccaro et al., 2017; Dong et al., 2020; Ansari et al., 2021; Smith et al., 2023). Despite these efforts in designing flexible SSM priors and developing stable training schemes, theoretical properties such as identifiability for theses sequential generative models are less studied, contrary to early literature on linear SSMs.

Identifiability, in general, establishes a one-to-one correspondence between the data likelihood and the model parameters (or latent variables), or an equivalence class of the latter. In causal discovery (Peters et al., 2017), identifiability refers to whether the underlying causal structure can be correctly pinpointed from infinite observational data. In independent component analysis (ICA, Comon (1994)), identifiability analysis focuses on both the latent sources and the mapping from the latents to the observed. While general non-linear ICA is ill-defined (Hyvärinen & Pajunen, 1999), recent results show that identifiability can be achieved using conditional priors (Khemakhem et al., 2020). Moreover, in a framework for deep (non-temporal) latent variable models, the required access to auxiliary variables can be relaxed using a finite mixture prior Kivva et al. (2022). Recent works have attempted to extend these results to sequential models using non-linear ICA (Hyvarinen & Morioka, 2017; Hyvarinen et al., 2019; Hälvä & Hyvarinen, 2020; Hälvä et al., 2021), or latent causal processes (Yao et al., 2022a;b; Lippe et al., 2023b;a).

In this work, we develop an identifiability analysis for Switching Dynamical Systems (SDSs) – a class of sequential LVMs with SSM priors that allow regime-switching behaviours, where their associated inference methods have been explored recently (Dong et al., 2020; Ansari et al., 2021).

Our approach differs fundamentally from non-linear ICA since the latent variables are no longer independent. To address this challenge, we first construct the latent dynamical prior using Markov Switching Models (MSMs) (Hamilton, 1989) – an extension of Hidden Markov Models (HMMs) with autoregressive connections, for which we provide the first identifiability analysis of this model family in non-linear transition settings. This also allows regime-dependent causal discovery (Saggioro et al., 2020) thanks to having access to the transition derivatives. We then extend the identifiability results to SDSs using recent identifiability analysis techniques for the non-temporal deep latent variable models (Kivva et al., 2022). Importantly, the identifiability conditions we provide are less restrictive, significantly broadening their applicability. In contrast, Yao et al. (2022a;b); Lippe et al. (2023a) require auxiliary information to handle regime-switching effects or distribution shifts, and Hälvä et al. (2021) imposes assumptions on the temporal correlations such as stationarity (Hyvarinen & Morioka, 2017) . Moreover, unlike the majority of existing works (Hälvä et al., 2021; Yao et al., 2022a;b), our identifiability analysis does not require the injectivity of the decoder mapping. Below we summarise our main contributions in both theoretical and empirical forms:

- We present conditions in which first-order MSMs with non-linear Gaussian transitions are identifiable up to permutations (Section 3.1). We further provide the first analysis of identifiability conditions for non-parametric first-order MSMs in Appendix B.

- We extend the previous result to SDSs, where we show conditions for identifiability up to affine transformations of the latent variables and non-linear emission (Section 3.2).

- We demonstrate the effectiveness of identifiable SDSs on causal discovery and sequence modelling tasks. These include discovery of time-dependent causal structures in climate data (Section 6.2), and time-series segmentation in complex data, e.g., videos (Section 6.3).

## 2 BACKGROUND

### 2.1 IDENTIFIABLE LATENT VARIABLE MODELS

In the non-temporal case, many works explore identifiability of latent variable models (Khemakhem et al., 2020; Kivva et al., 2022). Specifically, consider a generative model where its latent variables $\boldsymbol{z} \in \mathbb{R}^m$ are drawn from a Gaussian mixture prior with $K$ components ($K < +\infty$). Then $\boldsymbol{z}$ is transformed via a (noisy) non-linear mapping to obtain the observation $\boldsymbol{x} \in \mathbb{R}^n$, $n \geq m$:

$$\boldsymbol{x} = f(\boldsymbol{z}) + \boldsymbol{\epsilon}, \quad \boldsymbol{z} \sim p(\boldsymbol{z}) := \sum_{k=1}^{K} p(s = k) \mathcal{N}\left(\boldsymbol{z} | \boldsymbol{\mu}_k, \boldsymbol{\Sigma}_k\right), \quad \boldsymbol{\epsilon} \sim \mathcal{N}(\boldsymbol{0}, \boldsymbol{\Sigma}), \qquad (1)$$

Kivva et al. (2022) established that if the transformation $f$ is *weakly injective* (see definition below), the prior distribution $p(\boldsymbol{z})$ is identifiable up to affine transformations[1] from the observations. If we further assume $f$ is continuous and *injective*, both prior distribution $p(\boldsymbol{z})$ and non-linear mapping $f$ are identifiable up to affine transformations. In Section 3.2 we extend these results to establish identifiability for SDSs using similar proof strategies.

**Definition 2.1.** *A mapping $f : \mathbb{R}^m \to \mathbb{R}^n$ is said to be weakly injective if (i) there exsists $\boldsymbol{x}_0 \in \mathbb{R}^n$ and $\delta > 0$ s.t. $|f^{-1}(\{\boldsymbol{x}\})| = 1$ for every $\boldsymbol{x} \in B(\boldsymbol{x}_0, \delta) \cap f(\mathbb{R}^m)$, and (ii) $\{\boldsymbol{x} \in \mathbb{R}^n : |f^{-1}(\{\boldsymbol{x}\})| > 1\} \subseteq f(\mathbb{R}^m)$ has measure zero with respect to the Lebesgue measure on $f(\mathbb{R}^m)$.[2]*

**Definition 2.2.** *$f$ is said to be injective if $|f^{-1}(\{\boldsymbol{x}\})| = 1$ for every $\boldsymbol{x} \in f(\mathbb{R}^m)$.*

### 2.2 SWITCHING DYNAMICAL SYSTEMS

A Switching Dynamical System (SDS), with an example graphical model illustrated in Figure 1, is a sequential latent variable model with its dynamics governed by both discrete and continuous latent states, $s_t \in \{1, \ldots, K\}, \boldsymbol{z}_t \in \mathbb{R}^m$, respectively. At each time step $t$, the discrete state $s_t$ determines the regime of the dynamical prior that the current continuous latent variable $\boldsymbol{z}_t$ should follow, and the observation $\boldsymbol{x}_t$ is generated from $\boldsymbol{z}_t$ using a (noisy) non-linear transformation. This gives the following probabilistic model for the states and the observation variables:

$$p_{\boldsymbol{\theta}}(\boldsymbol{x}_{1:T}, \boldsymbol{z}_{1:T}, \mathbf{s}_{1:T}) = p_{\boldsymbol{\theta}}(\mathbf{s}_{1:T}) p_{\boldsymbol{\theta}}(\boldsymbol{z}_{1:T} | \mathbf{s}_{1:T}) \prod_{t=1}^{T} p_{\boldsymbol{\theta}}(\boldsymbol{x}_t | \boldsymbol{z}_t). \qquad (2)$$

---

[1]See Def. 2.2 in Kivva et al. (2022) or the equivalent adapted to SDSs (Def. 3.3 and Rem. 3.1).

[2]$B(\boldsymbol{x}_0, \delta) = \{\boldsymbol{x} \in \mathbb{R}^n : ||\boldsymbol{x} - \boldsymbol{x}_0|| < \delta\}$. We use the same notation as in Kivva et al. (2022), where we use sets as inputs to functions. I.e. for a set $\mathcal{B}$, we compute $f(\mathcal{B})$ as $\{f(\boldsymbol{x}) : \boldsymbol{x} \in \mathcal{B}\}$ or $f^{-1}(\mathcal{B})$ as $\{\boldsymbol{x} : f(\boldsymbol{x}) \in \mathcal{B}\}$.

As computing the marginal $p_{\boldsymbol{\theta}}(\boldsymbol{x}_{1:T})$ is intractable, recent works (Dong et al., 2020; Ansari et al., 2021) have developed inference techniques based on variational inference (Kingma & Welling, 2014). These works consider an additional dependency on the switch $s_t$ from $\boldsymbol{x}_t$ to allow more expressive segmentations, which we do not include in our theoretical analysis for simplicity.

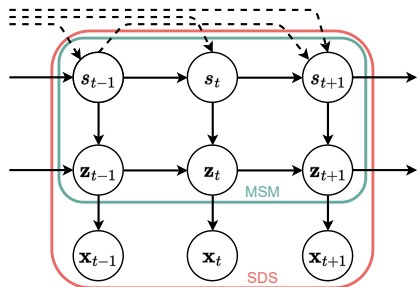

For the latent dynamic prior $p_{\boldsymbol{\theta}}(\boldsymbol{z}_{1:T})$, we consider a Markov Switching Model (MSM) which has also been referred to as autoregressive HMM (Ephraim & Roberts, 2005). This type of prior uses the latent "switch" $s_t$ to condition the distribution of $\boldsymbol{z}_t$ at each time-step, and the conditional dynamic model of $\boldsymbol{z}_{1:T}$ given $\mathbf{s}_{1:T}$ follows an autoregressive process. Under first-order Markov assumption for this conditional auto-regressive processes, this leads to the following probabilistic model for the prior:

Figure 1: The generative model considered in this work, where the MSM is indicated in green and the SDS is indicated in red. The dashed arrows indicate additional dependencies which are accommodated by our theoretical results.

$$p_{\boldsymbol{\theta}}(\boldsymbol{z}_{1:T}) = \sum_{\mathbf{s}_{1:T}} p_{\boldsymbol{\theta}}(\mathbf{s}_{1:T})p_{\boldsymbol{\theta}}(\boldsymbol{z}_{1:T}|\mathbf{s}_{1:T}), \quad p_{\boldsymbol{\theta}}(\boldsymbol{z}_{1:T}|\mathbf{s}_{1:T}) = p_{\boldsymbol{\theta}}(\boldsymbol{z}_1|s_1)\prod_{t=2}^{T}p_{\boldsymbol{\theta}}(\boldsymbol{z}_t|\boldsymbol{z}_{t-1}, s_t). \quad (3)$$

Note that the structure of the discrete latent state prior $p_{\boldsymbol{\theta}}(\mathbf{s}_{1:T})$ is not specified, and the identifiability results presented in the next section do not require further assumptions herein. As illustrated in Figure 1 (solid lines), in experiments we use a first-order Markov process for $p_{\boldsymbol{\theta}}(\mathbf{s}_{1:T})$, described by a transition matrix $Q \in \mathbb{R}^{K \times K}$ such that $p_{\boldsymbol{\theta}}(s_t = j|s_{t-1} = i) = Q_{ij}$, and an initial distribution $p_{\boldsymbol{\theta}}(s_1)$. In such case we also provide identifiability guarantees for the $Q$ matrix and the initial distribution.

## 3 THEORETICAL CONSIDERATIONS

This section establishes the identifiability of the SDS model (Eq. (2)) with MSM latent prior (Eq. (3)) under suitable assumptions. We address this challenge by leveraging ideas from Kivva et al. (2022) which uses finite mixture prior for static data generative models; importantly, this theory relies on the use of identifiable finite mixture priors (up to mixture component permutations). Inspired by this result, we first establish in Section 3.1 the identifiability of the Markov switching model $p_{\boldsymbol{\theta}}(\boldsymbol{z}_{1:T})$ as a finite mixture prior, which then allows us to extend the results of Kivva et al. (2022) to the temporal setting in Section 3.2 to prove identifiability of the SDS model. We drop the subscript $\boldsymbol{\theta}$ for simplicity.

### 3.1 IDENTIFIABLE MARKOV SWITCHING MODELS

The MSM $p(\boldsymbol{z}_{1:T})$ has an equivalent formulation as a finite mixture model which we will discuss as follows. Suppose the discrete states satisfy $s_t \in \{1, ..., K\}$ with $K < +\infty$, then for any given $T < +\infty$, one can define a bijective *path indexing function* $\varphi : \{1, ..., K^T\} \to \{1, ..., K\}^T$ such that each $i \in \{1, ..., K^T\}$ uniquely retrieves a set of states $\mathbf{s}_{1:T} = \varphi(i)$. Then we can use $c_i = p(\mathbf{s}_{1:T} = \varphi(i))$ to represent the joint probability of the states $\mathbf{s}_{1:T} = \varphi(i)$ under $p$. Let us further define the family of initial and transition distributions for the continuous states $\boldsymbol{z}_t$:

$$\Pi_{\mathcal{A}} := \{p_a(\boldsymbol{z}_1)|a \in \mathcal{A}\}, \quad \mathcal{P}_{\mathcal{A}} := \{p_a(\boldsymbol{z}_t|\boldsymbol{z}_{t-1})|a \in \mathcal{A}\}. \quad (4)$$

where $\mathcal{A}$ is an index set satisfying mild measure-theoretic conditions (Appendix A). Note $\mathcal{P}_{\mathcal{A}}$ assumes first-order Markov dynamics. Then, we can construct the family of first-order MSMs as

$$\mathcal{M}^T(\Pi_{\mathcal{A}}, \mathcal{P}_{\mathcal{A}}) := \left\{ \sum_{i=1}^{K^T} c_i p_{a_1^i}(\boldsymbol{z}_1) \prod_{t=2}^{T} p_{a_t^i}(\boldsymbol{z}_t|\boldsymbol{z}_{t-1}) \mid K < +\infty,\ p_{a_1^i} \in \Pi_{\mathcal{A}},\ p_{a_t^i} \in \mathcal{P}_{\mathcal{A}},\ t \geq 2, \right.$$
$$\left. a_t^i \in \mathcal{A}, \quad a_{1:T}^i \neq a_{1:T}^j, \forall i \neq j, \quad \sum_{i=1}^{K^T} c_i = 1 \right\}. \quad (5)$$

Since Eq. (5) requires $a_{1:T}^i \neq a_{1:T}^j$ for any $i \neq j$, this also builds an injective mapping $\phi(i) = a_{1:T}^i$ from $i \in \{1, ..., K^T\}$ to $\mathcal{A}$. Combined with the path indexing function, this establishes an injective mapping $\phi \circ \varphi^{-1}$ to uniquely map a set of states $\mathbf{s}_{1:T}$ to the $a_{1:T}$ indices, and we can view $p_{a_1^i}(\boldsymbol{z}_1)$ and $p_{a_t^i}(\boldsymbol{z}_t|\boldsymbol{z}_{t-1})$ as equivalent notations of $p(\boldsymbol{z}_1|s_1)$ and $p(\boldsymbol{z}_t|\boldsymbol{z}_{t-1}, s_t)$ respectively for $\mathbf{s}_{1:T} = \varphi(i)$.

This notation shows that the MSM extends finite mixture models to temporal settings as a finite mixture of $K^T$ trajectories composed by (conditional) distributions in $\Pi_{\mathcal{A}}$ and $\mathcal{P}_{\mathcal{A}}$.

Having established the finite mixture model view of Markov switching models, we will use this notation of MSM in the rest of Section 3.1, as we will use finite mixture modelling techniques to establish its identifiability. In detail, we first define the identification of $\mathcal{M}^T(\Pi_{\mathcal{A}}, \mathcal{P}_{\mathcal{A}})$ as follows.

**Definition 3.1.** *The family $\mathcal{M}^T(\Pi_{\mathcal{A}}, \mathcal{P}_{\mathcal{A}})$ that contains first-order MSMs is said to be identifiable up to permutations, when for $p_1(\boldsymbol{z}_{1:T}) = \sum_{i=1}^{K^T} c_i p_{a_1^i}(\boldsymbol{z}_1) \prod_{t=2}^{T} p_{a_t^i}(\boldsymbol{z}_t | \boldsymbol{z}_{t-1})$ and $p_2(\boldsymbol{z}_{1:T}) = \sum_{i=1}^{\hat{K}^T} \hat{c}_i p_{\hat{a}_1^i}(\boldsymbol{z}_1) \prod_{t=2}^{T} p_{\hat{a}_t^i}(\boldsymbol{z}_t | \boldsymbol{z}_{t-1})$, $p_1(\boldsymbol{z}_{1:T}) = p_2(\boldsymbol{z}_{1:T}), \forall \boldsymbol{z}_{1:T} \in \mathbb{R}^{Tm}$, if and only if $K = \hat{K}$ and for each $1 \leq i \leq K^T$ there is some $1 \leq j \leq \hat{K}^T$ s.t.*

1. *$c_i = \hat{c}_j$;*

2. *if $a_{t_1}^i = a_{t_2}^i$ for $t_1, t_2 \geq 2$ and $t_1 \neq t_2$, then $\hat{a}_{t_1}^j = \hat{a}_{t_2}^j$;*

3. *$p_{a_t^i}(\boldsymbol{z}_t | \boldsymbol{z}_{t-1}) = p_{\hat{a}_t^j}(\boldsymbol{z}_t | \boldsymbol{z}_{t-1}), \forall t \geq 2$, $\boldsymbol{z}_t, \boldsymbol{z}_{t-1} \in \mathbb{R}^m$;*

4. *$p_{a_1^i}(\boldsymbol{z}_1) = p_{\hat{a}_1^j}(\boldsymbol{z}_1), \forall \boldsymbol{z}_1 \in \mathbb{R}^m$.*

We note that the 2nd requirement eliminates the permutation equivalence of e.g., $\mathbf{s}_{1:4} = (1, 2, 3, 2)$ and $\hat{\mathbf{s}}_{1:4} = (3, 1, 2, 3)$ which would be valid in the finite mixture case with vector indexing.

For the purpose of building deep generative models, we seek to define identifiable parametric families and defer the study of the non-parametric case in Appendix B. In particular we use a non-linear Gaussian transition family as follows:

$$\mathcal{G}_{\mathcal{A}} = \{p_a(\boldsymbol{z}_t | \boldsymbol{z}_{t-1}) = \mathcal{N}(\boldsymbol{z}_t; \boldsymbol{m}(\boldsymbol{z}_{t-1}, a), \boldsymbol{\Sigma}(\boldsymbol{z}_{t-1}, a)) \mid a \in \mathcal{A}, \boldsymbol{z}_t, \boldsymbol{z}_{t-1} \in \mathbb{R}^m\}, \quad (6)$$

where $\boldsymbol{m}(\boldsymbol{z}_{t-1}, a)$ and $\boldsymbol{\Sigma}(\boldsymbol{z}_{t-1}, a)$ are non-linear with respect to $\boldsymbol{z}_{t-1}$ and denote the mean and covariance matrix of the Gaussian distribution. We further require the *unique indexing* assumption:

$$\forall a \neq a' \in \mathcal{A}, \exists \boldsymbol{z}_{t-1} \in \mathbb{R}^d, \ s.t. \ \boldsymbol{m}(\boldsymbol{z}_{t-1}, a) \neq \boldsymbol{m}(\boldsymbol{z}_{t-1}, a') \text{ or } \boldsymbol{\Sigma}(\boldsymbol{z}_{t-1}, a) \neq \boldsymbol{\Sigma}(\boldsymbol{z}_{t-1}, a'). \quad (7)$$

In other words, for such $\boldsymbol{z}_{t-1}$ we have $p_a(\boldsymbol{z}_t | \boldsymbol{z}_{t-1})$ and $p_{a'}(\boldsymbol{z}_t | \boldsymbol{z}_{t-1})$ as two different Gaussian distributions. We also introduce a family of *initial distributions* and assume unique indexing:

$$\mathcal{I}_{\mathcal{A}} := \{p_a(\boldsymbol{z}_1) = \mathcal{N}(\boldsymbol{z}_1; \boldsymbol{\mu}(a), \boldsymbol{\Sigma}_1(a)) \mid a \in \mathcal{A}\}, \quad (8)$$

$$a \neq a' \in \mathcal{A} \Leftrightarrow \boldsymbol{\mu}(a) \neq \boldsymbol{\mu}(a') \text{ or } \boldsymbol{\Sigma}_1(a) \neq \boldsymbol{\Sigma}_1(a'). \quad (9)$$

The above Gaussian distribution families paired with unique indexing assumptions satisfy conditions which favour identifiability of first-order MSMs under non-linear Gaussian transitions.

**Theorem 3.1.** *Define the following first-order Markov switching model family under the non-linear Gaussian families, $\mathcal{M}_{NL}^T = \mathcal{M}^T(\mathcal{I}_{\mathcal{A}}, \mathcal{G}_{\mathcal{A}})$ with $\mathcal{G}_{\mathcal{A}}, \mathcal{I}_{\mathcal{A}}$ defined by Eqs. (6), (8) respectively. Then, the Markov switching model is identifiable in terms of Def. 3.1 under the following assumptions:*

(a1) *Unique indexing for $\mathcal{G}_{\mathcal{A}}$ and $\mathcal{I}_{\mathcal{A}}$: Eqs. (7), (9) hold;*

(a2) *For any $a \in \mathcal{A}$, the mean and covariance in $\mathcal{G}_{\mathcal{A}}$, $\boldsymbol{m}(\cdot, a) : \mathbb{R}^m \to \mathbb{R}^m$ and $\boldsymbol{\Sigma}(\cdot, a) : \mathbb{R}^m \to \mathbb{R}^{m \times m}$, are analytic functions.*

*Proof sketch:* See Appendix B for the proof. The strategy can be summarised in 4 steps.

(i) Under the finite mixture model view of MSM, it suffices to show that $\{p_{a_{1:T}^i}(\boldsymbol{z}_{1:T}) \mid a_{1:T}^i \in \mathcal{A} \times \cdots \times \mathcal{A}\}$ contains linearly independent functions (Yakowitz & Spragins, 1968).

(ii) Due to the conditional first-order Markov assumption on $p_{a_{1:T}^i}(\boldsymbol{z}_{1:T})$, we just need to find conditions for the linear independence of $\{p_{a_{1:2}^i}(\boldsymbol{z}_{1:2})\}$, and then prove $T \geq 3$ by induction.

(iii) To ensure linear independence of functions in $\{p_{a_{1:2}^i}(\boldsymbol{z}_{1:2})\}$, we specify conditions on $\mathcal{P}_{\mathcal{A}} = \{p_a(\boldsymbol{z}_t | \boldsymbol{z}_{t-1})\}$ and $\Pi_{\mathcal{A}} = \{p_a(\boldsymbol{z}_1)\}$ in non-parametric case.

(iv) We show the Markov switching model family with (non-linear) Gaussian transitions $\mathcal{M}_{NL}^T = \mathcal{M}^T(\mathcal{I}_{\mathcal{A}}, \mathcal{G}_{\mathcal{A}})$ is a special case of (iii) when assuming (a1 - a2).

Assumption (a2) allows parametrisations via e.g., polynomials and neural networks with analytic activation functions (e.g. SoftPlus). In the latter case, the identifiability result applies to the functional form only, since network weights do not uniquely index the functions that they parameterise. Note the identifiability result holds independently of the choice of $p(\mathbf{s}_{1:T})$. In our experiments, we consider a stationary first-order Markov chain for $p(\mathbf{s}_{1:T})$ and leave other cases to future work. We also state the following identifiability result regarding the discrete state transitions for completeness.

**Corollary 3.1.** *Consider an identifiable MSM from Def. 3.1, where the prior distribution of the states $p(\mathbf{s}_{1:T})$ follows a first-order stationary Markov chain, i.e $p(\mathbf{s}_{1:T}) = \pi_{s_1} Q_{s_1, s_2} \ldots Q_{s_{T-1}, s_T}$, where $\boldsymbol{\pi}$ denotes the initial distribution: $p(s_1 = k) = \pi_k$, and $Q$ denotes the transition matrix: $p(s_t = k | s_{t-1} = l) = Q_{l,k}$. Then, $\boldsymbol{\pi}$ and $Q$ are identifiable up to permutations.*

## 3.2 Identifiable Switching Dynamical Systems

We now turn to the analysis of Switching Dynamical Systems, whose setup can be viewed as an extension of the setup from Kivva et al. (2022) (Section 2.1) to the temporal case. Assume the prior dynamic model $p(\boldsymbol{z}_{1:T})$ belongs to the non-linear Gaussian MSM family $\mathcal{M}_{NL}^T$ specified by Thm. 3.1. At each time step $t$, the latent $\boldsymbol{z}_t \in \mathbb{R}^m$ generates observed data $\boldsymbol{x}_t \in \mathbb{R}^n$ via a piece-wise linear transformation $f$. The generation process of such SDS model can be expressed as follows:

$$\boldsymbol{x}_t = f(\boldsymbol{z}_t) + \boldsymbol{\epsilon}_t, \quad \boldsymbol{z}_{1:T} \sim p(\boldsymbol{z}_{1:T}) \in \mathcal{M}_{NL}^T, \quad \boldsymbol{\epsilon}_t \sim \mathcal{N}(\mathbf{0}, \boldsymbol{\Sigma}). \quad (10)$$

Note that we can also write the decoding process as $\boldsymbol{x}_{1:T} = \mathcal{F}(\boldsymbol{z}_{1:T}) + \mathcal{E}$ with $\mathcal{E} = (\boldsymbol{\epsilon}_1, ..., \boldsymbol{\epsilon}_T)^\top$, where $\mathcal{F}$ is a factored transformation composed by $f$ as defined below. Importantly, this notion allows $\mathcal{F}$ to inherit e.g., piece-wise linearity and (weakly) injectivity properties from $f$.

**Definition 3.2.** *We say that a function $\mathcal{F} : \mathbb{R}^{mT} \to \mathbb{R}^{nT}$ is factored if it is composed by $f : \mathbb{R}^m \to \mathbb{R}^n$, such that for any $\boldsymbol{z}_{1:T} \in \mathbb{R}^{mT}$, $\mathcal{F}(\boldsymbol{z}_{1:T}) = (f(\boldsymbol{z}_1), \ldots, f(\boldsymbol{z}_T))^\top$.*

Below we carry out the identifiability analysis of the SDS model (Eq. (10)), which follows two steps (see Figure 2). (i) Following Kivva et al. (2022), we establish the identifiability for a prior $p(\boldsymbol{z}_{1:T}) \in \mathcal{M}_{NL}^T$ from $(\mathcal{F}_{\#}p)(\boldsymbol{x}_{1:T})$ in the noiseless case. Here $\mathcal{F}_{\#}p$ denotes the pushforward measure of $p$ by $\mathcal{F}$. (ii) When the noise $\mathcal{E}$ distribution is known, $\mathcal{F}_{\#}p$ is identifiable from $(\mathcal{F}_{\#}p) * \mathcal{E}$ using convolution tricks from Khemakhem et al. (2020). In detail, we first define the notion of identifiability for $p(\boldsymbol{z}_{1:T}) \in \mathcal{M}^T(\Pi_{\mathcal{A}}, \mathcal{P}_{\mathcal{A}})$ given noiseless observations.

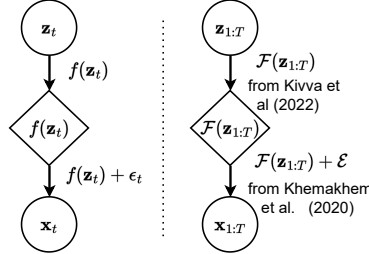

$\mathcal{F}(\mathbf{z}_{1:T}) = (f(\mathbf{z}_1), \ldots, f(\mathbf{z}_T))^T$

Figure 2: $\boldsymbol{z}_t$ is transformed via $f$ with noise $\boldsymbol{\epsilon}_t$ at each time-step $t$ independently. We view this as a transformation on $\boldsymbol{z}_{1:T}$ via a factored $\mathcal{F}$ with noise $\mathcal{E}$.

**Definition 3.3.** *Given a family of factored transformations $\mathbb{F}$, for $\mathcal{F} \in \mathbb{F}$ the prior $p \in \mathcal{M}^T(\Pi_{\mathcal{A}}, \mathcal{P}_{\mathcal{A}})$ is said to be identifiable up to affine transformations, when for any $\mathcal{F}' \in \mathbb{F}$ and $p' \in \mathcal{M}^T(\Pi_{\mathcal{A}}, \mathcal{P}_{\mathcal{A}})$ s.t. $\mathcal{F}_{\#}p = \mathcal{F}'_{\#}p'$, there exists an invertible factored affine mapping $\mathcal{H} : \mathbb{R}^{mT} \to \mathbb{R}^{mT}$ composed by $h : \mathbb{R}^m \to \mathbb{R}^m$, where $p \equiv \mathcal{H}_{\#}p'$.*

**Remark 3.1.** *For the factored $\mathcal{F}$, identifiability up to affine transformations extends from Def. 2.2.1 in Kivva et al. (2022), which is defined on $f$. In this case, the fact that we have some mappings $f, f'$ such that $f = (f' \circ h)$, where $h : \mathbb{R}^m \to \mathbb{R}^m$ is an invertible affine transformation, implies $\mathcal{F} = (\mathcal{F}' \circ \mathcal{H})$, where the factored mappings $\mathcal{F}, \mathcal{F}'$, and $\mathcal{H}$ are composed by $f, f'$, and $h$ respectively.*

Now we state the following identifiability result for (non-linear) SDS and prove it in Appendix D.

**Theorem 3.2.** *Assume there exists a latent variable model where observations are generated following Eq. (10), the prior distribution $p(\boldsymbol{z}_{1:T})$ belongs to the family $\mathcal{M}_{NL}^T = \mathcal{M}^T(\mathcal{I}_{\mathcal{A}}, \mathcal{G}_{\mathcal{A}})$ and satisfies (a1 - a2), and $f$ is a piece-wise linear mapping. Then:*

> *(i) If $f$, which composes $\mathcal{F}$, is weakly injective (Def. 2.1), the prior $p(\boldsymbol{z}_{1:T})$ is identifiable up to affine transformations as defined in Def. 3.3.*
> *(ii) If $f$, which composes $\mathcal{F}$, is continuous and injective (Def. 2.2), both prior $p(\boldsymbol{z}_{1:T})$ and $f$ are identifiable up to affine transformations, as defined in Def. 3.3 and Rem. 3.1.*

This result allows us to design identifiable SDSs parametrised using e.g., SoftPlus networks for $p_{\boldsymbol{\theta}}(\boldsymbol{z}_t | \boldsymbol{z}_{t-1}, s_t)$, and (Leaky) ReLU networks for $p_{\boldsymbol{\theta}}(\boldsymbol{x}_t | \boldsymbol{z}_t)$. Again, for neural networks identifiability refers only to their functional form.

**Remark 3.2.** *For an invertible factored affine transformation $\mathcal{H}$, we can show that $p_1 = \mathcal{H}_{\#}p_2$ with $p_1, p_2 \in \mathcal{M}_{NL}^T$. Then, we can obtain the following relation between transition parameters.*

$$\boldsymbol{m}_1(\boldsymbol{z}, a) = A\boldsymbol{m}_2\left(A^{-1}(\boldsymbol{z} - \mathbf{b}), \sigma(a)\right) + \mathbf{b}, \quad \boldsymbol{z} \in \mathbb{R}^m, a \in \mathcal{A} \tag{11}$$

$$\boldsymbol{\Sigma}_1(\boldsymbol{z}, a) = A\boldsymbol{\Sigma}_2\left(A^{-1}(\boldsymbol{z} - \mathbf{b}), \sigma(a)\right)A^T, \quad \boldsymbol{z} \in \mathbb{R}^m, a \in \mathcal{A} \tag{12}$$

*where $\sigma(\cdot)$ is a permutation over the set $\mathcal{A}$, and we use the subscript to refer to functions of $p_1$ or $p_2$. To see this, in Prop. C.1 we show that the MSM family is closed under factored affine transformations. Furthermore, $\mathcal{H}$ is composed by $h$, which is affine, i.e $h(\boldsymbol{z}) = A\boldsymbol{z} + \mathbf{b}, \boldsymbol{z} \in \mathbb{R}^m$. Finally, we can leverage the previous identifiability up to permutations result (Thm. 3.1) to establish the above.*

## 4 ESTIMATION

We consider two modelling choices for $N$ sequences $\mathcal{D} = \{\boldsymbol{x}_{1:T}\}$ of length $T$: (a) the MSM model (Eq. (3) with $\boldsymbol{z}_{1:T}$ replaced by $\boldsymbol{x}_{1:T}$) and (b) the SDS model with MSM latent prior (Eqs. (2),(3)). Although our theory imposes no restrictions on the discrete latent state distribution, the methods are implemented with a first-order stationary Markov chain, i.e., $p_{\boldsymbol{\theta}}(\mathbf{s}_{1:T}) = p_{\boldsymbol{\theta}}(s_1)\prod_{t=2}^T p_{\boldsymbol{\theta}}(s_t|s_{t-1})$.

**Markov Switching Models** We use expectation maximisation (EM) for efficient estimation of mixture models (Bishop, 2006). Below we discuss the M-step update of the transition distribution parametrised by neural networks; more details (including polynomial parametrisations) can be found in Appendix E. When considering analytic neural networks, we take a Generalised EM (GEM) (Dempster et al., 1977) approach where a gradient ascent step is performed

$$\boldsymbol{\theta}^{\text{new}} \leftarrow \boldsymbol{\theta}^{\text{old}} + \eta \sum_{t=2}^T \sum_{k=1}^K \gamma_{t,k} \nabla_{\boldsymbol{\theta}} \log p_{\boldsymbol{\theta}}(\boldsymbol{x}_t|\boldsymbol{x}_{t-1}, s_t = k), \tag{13}$$

where $\gamma_{t,k} = p_{\boldsymbol{\theta}}(s_t = k|\boldsymbol{x}_{1:T})$ and the update rule can be computed using back-propagation. In the equation we indicate the update rule for a single sample, but note that we use mini-batch stochastic gradient ascent when $N$ is large. Convergence is guaranteed for this approach to a local maximum of the likelihood (Bengio & Frasconi, 1994; Hälvä & Hyvarinen, 2020).

**Switching Dynamical Systems** We adopt variational inference as presented in Ansari et al. (2021). The parameters are learned by maximising the evidence lower bound (ELBO) (Kingma & Welling, 2014), and the proposed approximate posterior over the latent variables $\{\boldsymbol{z}_{1:T}, \mathbf{s}_{1:T}\}$ incorporates the exact posterior of the discrete latent states given the continuous latent variables:

$$q_{\boldsymbol{\phi}, \boldsymbol{\theta}}(\boldsymbol{z}_{1:T}, \mathbf{s}_{1:T}|\boldsymbol{x}_{1:T}) = q_{\boldsymbol{\phi}}(\boldsymbol{z}_1|\boldsymbol{x}_{1:T})\prod_{t=2}^T q_{\boldsymbol{\phi}}(\boldsymbol{z}_t|\boldsymbol{z}_{1:t-1}, \boldsymbol{x}_{1:T})p_{\boldsymbol{\theta}}(\mathbf{s}_{1:T}|\boldsymbol{z}_{1:T}). \tag{14}$$

As in Ansari et al. (2021) and Dong et al. (2020), the variational posterior over the continuous variables $q_{\boldsymbol{\phi}}(\boldsymbol{z}_{1:T}|\boldsymbol{x}_{1:T})$ simulates a smoothing process by first using a bi-directional RNN on $\boldsymbol{x}_{1:T}$, and then a forward RNN on the resulting embeddings. By introducing an exact posterior, the discrete latent variables can be marginalised from the ELBO objective (see Appendix E.2 for details),

$$p_{\boldsymbol{\theta}}(\boldsymbol{x}_{1:T}) \geq \mathbb{E}_{q_{\boldsymbol{\phi}}(\boldsymbol{z}_{1:T}|\boldsymbol{x}_{1:T})}\left[\log p_{\boldsymbol{\theta}}(\boldsymbol{x}_{1:T}|\boldsymbol{z}_{1:T})\right] - KL\left(q_{\boldsymbol{\phi}}(\boldsymbol{z}_{1:T}|\boldsymbol{x}_{1:T})||p_{\boldsymbol{\theta}}(\boldsymbol{z}_{1:T})\right). \tag{15}$$

We use Monte Carlo estimation with samples $\boldsymbol{z}_{1:T} \sim q_{\boldsymbol{\phi}}(\boldsymbol{z}_{1:T}|\boldsymbol{x}_{1:T})$ as well as the reparametrization trick for back-propagation (Kingma & Welling, 2014), and jointly learn the parameters using stochastic gradient ascent on the ELBO objective. The prior distribution $p_{\boldsymbol{\theta}}(\boldsymbol{z}_{1:T})$ can be computed exactly using the forward algorithm (Bishop, 2006) with messages $\{\alpha_{t,k}(\boldsymbol{z}_{1:t}) = p_{\boldsymbol{\theta}}(\boldsymbol{z}_{1:t}, s_t = k)\}$ by marginalising out $\mathbf{s}_{1:T}$:

$$p_{\boldsymbol{\theta}}(\boldsymbol{z}_{1:T}) = \sum_{k=1}^K \alpha_{T,k}(\boldsymbol{z}_{1:T}), \quad \alpha_{1,k}(\boldsymbol{z}_1) = p_{\boldsymbol{\theta}}(\boldsymbol{z}_1|s_1 = k)p_{\boldsymbol{\theta}}(s_1 = k),$$

$$\alpha_{t,k}(\boldsymbol{z}_{1:t}) = \sum_{k'=1}^K p_{\boldsymbol{\theta}}(\boldsymbol{z}_t|\boldsymbol{z}_{t-1}, s_t = k)p_{\boldsymbol{\theta}}(s_t = k|s_{t-1} = k')\alpha_{t-1,k'}(\boldsymbol{z}_{1:t-1}). \tag{16}$$

An alternative approach for SDS posterior inference is presented in Dong et al. (2020), although Ansari et al. (2021) outlines some disadvantages (see Appendix E.2 for a discussion). Note that estimating parameters with ELBO maximisation has no consistency guarantees in general, and we leave additional analyses regarding consistency for future work.

## 5 RELATED WORK

Sequential generative models based on non-linear Switching Dynamical Systems have gained interest over the recent years as they consider an expressive prior distribution that allows successful time series segmentation and forecasting. Some works include soft-switching Kalman Filters (Fraccaro et al., 2017), or recurrent SDSs (Dong et al., 2020) which can include explicit duration models on the switches (Ansari et al., 2021). However, identifiability in such highly non-linear scenarios is rarely studied. A similar situation is found for Markov Switching Models, which were first introduced by Poritz (1982) as switching linear auto-regressive processes. This family of state-space models re-use the forward-backward recursions (Rabiner, 1989) for tractable posterior estimation and have been studied decades ago for speech analysis (Poritz, 1982; Ephraim & Roberts, 2005) and economics (Hamilton, 1989). Frühwirth-Schnatter & Frèuhwirth-Schnatter (2006) review standard estimation approaches and applications of the MSM family. Although the majority of the proposed approaches estimate the parameters using tractable MLE solutions for their asymptotic properties, identifiability for general high-order autoregressive MSMs has not been proved. The main complication arises from the explicit dependency on the observed variables, which poses a great challenge to prove linear independence of the joint distribution given the states under relaxed assumptions. To the best of our knowledge, identifiability in MSMs has been explicitly studied in few occasions. An et al. (2013), establishes results in the discrete case from the joint probability of four consecutive observations.

Regarding non-linear ICA for time series, identifiability results extend from Khemakhem et al. (2020) where past values can be used as auxiliary information. Hyvarinen & Morioka (2017), Hyvarinen et al. (2019), Klindt et al. (2021), Hälvä & Hyvarinen (2020), and Morioka et al. (2021) assume mutually independent sources where the latter two introduce latent regime-switching sources via identifiable HMMs (Gassiat et al., 2016). Similar to our work, Hälvä et al. (2021) allows SDSs by imposing restrictions on the temporal correlations, despite having no identifiability of the latent transitions. For latent causal processes (Yao et al., 2022a;b) recover time-delayed latent causal variables and identify their dependencies. Their framework allows non-stationarity via distribution shifts or time-dependent effects across regimes given auxiliary information. Other works identify latent causal variables from multi-dimensional observations using intervention targets (Lippe et al., 2023b) or unknown binary interactions with auxiliary regime information (Lippe et al., 2023a).

## 6 EXPERIMENTS

We evaluate the identifiable MSMs and SDSs with three experiments: (1) simulation studies with ground truth available for verification of the identifiability results; (2) regime-dependent causal discovery in climate data with identifiable MSMs; and (3) segmentation of high-dimensional sequences of salsa dancing using MSMs and SDSs. Additional results are also presented in Appendix F.

### 6.1 SYNTHETIC EXPERIMENTS

**Markov Switching Models**    We generate data using ground-truth MSMs and evaluate the estimated functions upon them. We use fixed covariances and parametrise the transition means using random cubic polynomials or networks with cosine/SoftPlus activations. When increasing dimensions, we use locally connected networks (Zheng et al., 2018) to construct regime-dependent causal structures for the grouth-truth MSMs, which encourage sparsity and stable data generation. The estimation error is computed with $L_2$ distance between grouth-truth and estimated transition mean functions, after accounting for permutation equivalence and averaged over $K$ components. We also estimate the causal structure via thresholding the Jacobian of the estimated transition function, and compute the $F_1$ score (again after accounting for permutation equivalence) to evaluate structure estimation accuracy. See Appendices F.1 to F.4 for details regarding the experiment settings and metric computations.

As expected, Figure 3a shows increasing the sequence length generally reduces the $L_2$ estimation error. The polynomials are estimated with higher error for short sequences, which could be caused by the high-frequency components from the cubic terms. Meanwhile the smoothness of the softplus networks allows the MSM to achieve consistently better parameter estimates. Regarding scalability, Figure 3b shows low estimation errors when increasing the number of dimensions and components. For structure estimation acurracy, Figure 3c shows that the MSM with non-linear transitions is able to maintain high $F_1$ scores, despite the differences in $L_2$ distance when increasing dimensions and states (3b). Although the approach is restricted by first-order Markov assumptions, the synthetic setting shows promising directions for high-dimensional regime-dependent causal discovery.

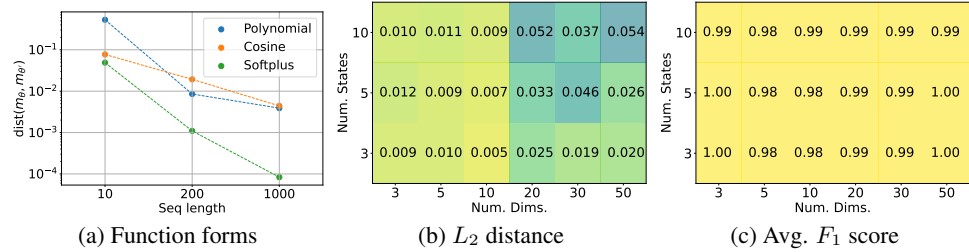

(a) Function forms      (b) $L_2$ distance      (c) Avg. $F_1$ score

Figure 3: Synthetic experiment results on MSMs. (a) $L_2$ distance error using different transition functions with varying $T$. (b) $L_2$ distance error and (c) averaged $F_1$ score of non-linear data (cosine activations) with increasing states and dimensions.

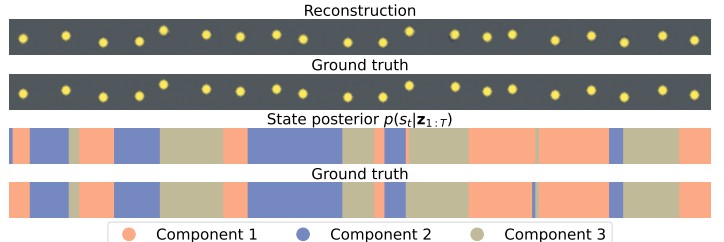

Figure 4: Reconstruction and segmentation (with ground truth) of a video generated from 2D latent variables sampled from a MSM.

Table 1: Synthetic experiment results on SDSs for increasing $x_t$ dimensions.

| Dims | $F_1$ score | $L_2$ dist. |
|---|---|---|
| 2 | 0.997 | 0.015 |
| 5 | 0.996 | 0.006 |
| 10 | 0.997 | 0.020 |
| 50 | 0.998 | 0.031 |
| 100 | 0.997 | 0.005 |
| (32,32,3) | 0.988 | 0.309 |

**Switching Dynamical Systems** We use data from 2D MSMs with cosine activations and $K = 3$ components, and a random Leaky ReLU network to generate observations (with no additive noise). For evaluation, we generate 1000 test samples and compute the $F_1$ score of the state posterior with respect the ground truth component. We also compute the $L_2$ distance of $\boldsymbol{m}(\boldsymbol{z}, \cdot)$ using Eq. (11). Again both metrics are computed after accounting for permutation and affine transformation equivalence (see Appendix F.1). We report the results in Table 1. The method is able to maintain high $F_1$ scores as well as a low $L_2$ distance between the estimated and ground-truth transition mean functions (modulo equivalences). To motivate the use of identifiable SDSs for real-world data, we generate videos with frame size $32 \times 32$ by treating the 2D latents as positions of a moving ball, and show a reconstruction with the corresponding segmentation in Figure 4. This high-dimensional setting increases the difficulty of accurate estimation as the reconstruction term of the ELBO out-weights the KL term for learning the latent MSM (Appendix F.3). This results in an increased $L_2$ distance from the ground truth MSM. Still, the estimated model achieves high-fidelity reconstructions (with an averaged pixel MSE of $8.89 \cdot 10^{-5}$), and accurate structure estimation as indicated by the $F_1$ score.

## 6.2 REGIME-DEPENDENT CAUSAL DISCOVERY

We explore regime-dependent causal discovery using climate data from Saggioro et al. (2020). The data consists on monthly observations of El Niño Southern Oscillation (ENSO) and All India Rainfall (AIR) from 1871 to 2016. We follow Saggioro et al. (2020) and train identifiable MSMs with linear and non-linear (softplus networks) transitions. Figure 5

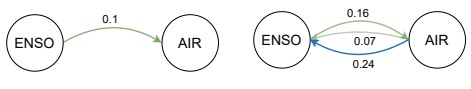

(a) linear effects      (b) non-linear effects

Figure 5: Regime-dependent graphs generated assuming (a) linear and (b) non-linear effects. Green and blue lines indicate effects in summer and winter months respectively.

shows the regime-dependent graphs extracted from both models, where the edge weights denote the absolute value of the corresponding entries in the estimated transition function's Jacobian (we keep edges with weights $\geq 0.05$). The MSMs capture regimes based on seasonality, as one component is assigned to summer months (May - September), and the other is assigned to winter months (October - April). In the linear case, the MSM discovers an effect from ENSO to AIR which occurs only during Summer. This result is consistent with Saggioro et al. (2020) and Webster & Palmer (1997), which suggests that ENSO has a direct effect on AIR during summer, but not in winter. For non-linear transitions (Figure 5b), additional links are discovered which are yet to be supported by scientific evidence in climate science. But because the flexibility of non-linear MSMs allows capturing correlations as causal effects in disguise, finding additional links may imply the presence of confounders that have an influence on both variables, which is often the case in scenarios with few observations.

## 6.3 SEGMENTATION OF DANCING PATTERNS

We consider *salsa dancing* sequences from the CMU mocap data to demonstrate our models' ability in segmenting high-dimensional time-series. See Appendix for details on the data (F.6), training methods (F.3), and additional datasets (F.7). We start from key-point tracking data and present in Figure 6 the segmentation results using both the identifiable MSM (softplus networks) and a baseline KVAE (Fraccaro et al., 2017). The iMSM assigns different dancing patterns to different states, e.g., the major pattern (forward-backward movements) is assigned to state 1. The iMSM also identifies this pattern at the end, which KVAE fails to recognise. Additionally, KVAE assigns a turning pattern into component 2, while iMSM treats turning as in state 1 patterns, but then jumps to component 2 consistently after observing

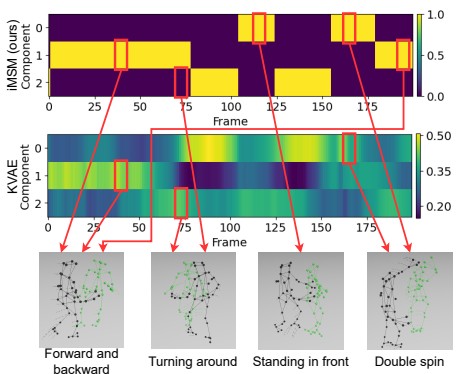

Figure 6: Posterior probability of a *salsa dancing* sequence of our approach (iMSM) and KVAE (Fraccaro et al., 2017) along with several patterns distinguished in the example.

it. The iMSM also classifies other dancing patterns into state 0. For limitations, the soft-switching mechanism restricts KVAE from confident component assignments. The iMSM's first-order Markov conditional transitions make it difficult to learn e.g., acceleration that would provide richer features for better segmentation.

Instead of evaluating the identifiable SDS on key-point trajectories, we generate videos by rendering 3D meshes from them using the renderer from Mahmood et al. (2019) to demonstrate identifiable SDS's applicability to videos. Figure 7 shows the corresponding video reconstructions and segmentation results, indicating similar interpretations: one component is more prominent and used for the majority of the forward and backward patterns; and the other components are used to model spinning and other dancing patterns. Quantitatively, our approach successfully reconstructs the sequences with an averaged pixel MSE of $2.26 \cdot 10^{-4}$, computed from a held out dataset of 560 sequences.

## 7 CONCLUSIONS

We present identifiability analysis regarding Markov Switching Models and Switching Dynamical Systems. Key to our contribution is the use of Gaussian transitions with analytic functions, which helps establish identifiability of MSMs, independently of the dynamic prior for the discrete states. We further extend the results to develop identifiable SDSs fully parameterised by neural networks. We empirically verify our theoretical results with synthetic experiments, and motivate our approach for regime-dependent causal discovery and high-dimensional time series segmentation with real data.

While our work focuses on identifiability analysis, in practice accurate estimation is also key to the success of causal discovery/representation learning from real data. Specifically, the current estimation methods are highly sensitive to hyper-parameter tuning, especially in modelling high-dimensional data where many estimation approaches are prone to state collapse. Also variational learning for sequential LVMs has no consistency guarantees unless assuming universal approximations of the $q$ posterior (Gong et al., 2023), which disagrees with the popular choices of Gaussian encoders. Future work should address these challenges in order to further scale SDSs for real-world causal

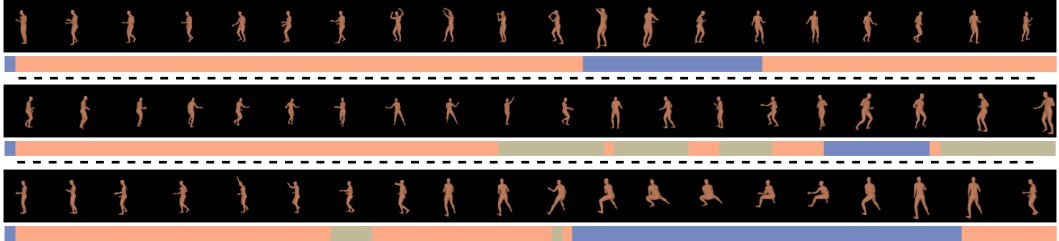

Figure 7: Reconstructions and segmentations of salsa dancing videos, where different colours indicate different components (brown: moving forward-backward, green: spinning, blue: others).

discovery. Other future directions include extending the identifiability results to higher-order MSMs, and designing efficient estimation methods for non-stationary discrete latent state priors.

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

## A  NON-PARAMETRIC FINITE MIXTURE MODELS

We use the following existing result on identifying finite mixtures (Yakowitz & Spragins, 1968), which introduces the concept of linear independence to the identification of finite mixtures. Specifically, consider a distribution family that contains functions defined on $\boldsymbol{x} \in \mathbb{R}^d$:

$$\mathcal{F}_{\mathcal{A}} := \{F_a(\boldsymbol{x}) | a \in \mathcal{A}\} \tag{17}$$

where $F_a(\boldsymbol{x})$ is an $d$-dimensional CDF and $\mathcal{A}$ is a measurable index set such that $F_a(\boldsymbol{x})$ as a function of $(\boldsymbol{x}, a)$ is measurable on $\mathbb{R}^d \times \mathcal{A}$. In this paper, we assume this measure theoretic assumption on $\mathcal{A}$ is satisfied. Now consider the following finite mixture distribution family by linearly combining the CDFs in $\mathcal{F}$:

$$\mathcal{H}_{\mathcal{A}} := \{H(\boldsymbol{x}) = \sum_{i=1}^{N} c_i F_{a_i}(\boldsymbol{x}) | N < +\infty, a_i \in \mathcal{A}, a_i \neq a_j, \forall i \neq j, \sum_{i=1}^{N} c_i = 1\}. \tag{18}$$

Then we specify the definition of *identifiable finite mixture family* as follows:

**Definition A.1.** *The finite mixture family $\mathcal{H}$ is said to be identifiable up to permutations, when for any two finite mixtures $H_1(x) = \sum_{i=1}^{M} c_i F_{a_i}(\boldsymbol{x})$ and $H_2(x) = \sum_{i=1}^{M} \hat{c}_i F_{\hat{a}_i}(\boldsymbol{x})$, $H_1(\boldsymbol{x}) = H_2(\boldsymbol{x})$ for all $\boldsymbol{x} \in \mathbb{R}^d$, if and only if $M = N$ and for each $1 \leq i \leq N$ there is some $1 \leq j \leq M$ such that $c_i = \hat{c}_j$ and $F_{a_i}(\boldsymbol{x}) = F_{\hat{a}_j}(\boldsymbol{x})$ for all $\boldsymbol{x} \in \mathbb{R}^d$.*

Then Yakowitz & Spragins (1968) proved the identifiability results for finite mixtures. To see this, we first introduce the concept of linearly independent functions under finite mixtures as follows.

**Definition A.2.** *A family of functions $\mathcal{F} = \{f_a(\boldsymbol{x}) | a \in \mathcal{A}\}$ is said to contain linearly independent functions under finite mixtures, if for any $\mathcal{A}_0 \subset \mathcal{A}$ such that $|\mathcal{A}_0| < +\infty$, the functions in $\{f_a(\boldsymbol{x}) | a \in \mathcal{A}_0\}$ are linearly independent.*

This is a weaker requirement of linear independence on function classes as it allows linear dependency by representing one function as the linear combination of *infinitely many* other functions. With this relaxed definition of linear independence we state the identifiability result of finite mixture models as follows.

**Proposition A.1.** *(Yakowitz & Spragins, 1968) The finite mixture distribution family $\mathcal{H}$ is identifiable up to permutations, iff. functions in $\mathcal{F}$ are linearly independent under the finite mixture model.*

## B  PROOF OF THEOREM 3.1

We follow the strategy described in the main text.

### B.1  IDENTIFIABILITY VIA LINEAR INDEPENDENCE

Proposition A.1 can be directly generalised to CDFs defined on $\boldsymbol{z}_{1:T} \in \mathbb{R}^{Tm}$. Furthermore, if we have a family of PDFs[3], e.g. $\mathcal{P}_{\mathcal{A}}^T := \Pi_{\mathcal{A}} \otimes (\otimes_{t=2}^T \mathcal{P}_{\mathcal{A}})$, with linearly independent components, then their corresponding $Tm$-dimensional CDFs are also linearly independent (and vice versa). Therefore we have the following result as a direct extension of Proposition A.1.

**Proposition B.1.** *Consider the distribution family given by Eq. 5. Then the joint distribution in $\mathcal{M}^T(\Pi_{\mathcal{A}}, \mathcal{P}_{\mathcal{A}})$ is identifiable up to permutations if and only if functions in $\mathcal{P}_{\mathcal{A}}^T$ are linearly independent under finite mixtures.*

The above assumption of linear independence under finite mixtures over the joint distribution implies the following identifiability result.

**Theorem B.1.** *Assume the functions in $\mathcal{P}_{\mathcal{A}}^T := \Pi_{\mathcal{A}} \otimes (\otimes_{t=2}^T \mathcal{P}_{\mathcal{A}})$ are linearly independent under finite mixtures, then the distribution family $\mathcal{M}^T(\Pi_{\mathcal{A}}, \mathcal{P}_{\mathcal{A}})$ is identifiable as defined in Definition 3.1.*

---

[3]In this case we assume that the probability measures are dominated by the Lebesgue measure on $\mathbb{R}^{Tm}$ and the CDFs are differentiable.

*Proof.* From proposition B.1 we see that, $\mathcal{P}_{\mathcal{A}}^T$ being linearly independent implies identifiability up to permutation for $\mathcal{M}^T(\Pi_{\mathcal{A}}, \mathcal{P}_{\mathcal{A}})$ in the finite mixture sense (Definition A.1). This means for $p_1(\boldsymbol{z}_{1:T})$ and $p_2(\boldsymbol{z}_{1:T})$ defined in Definition 3.1, we have $K = \hat{K}$ and for every $1 \leq i \leq K^T$, there exists $1 \leq j \leq \hat{K}^T$ such that $c_i = \hat{c}_j$ and

$$p_{a_1^i}(\boldsymbol{z}_1) \prod_{t=2}^T p_{a_t^i}(\boldsymbol{z}_t|\boldsymbol{z}_{t-1}) = p_{\hat{a}_1^j}(\boldsymbol{z}_1) \prod_{t=2}^T p_{\hat{a}_t^j}(\boldsymbol{z}_t|\boldsymbol{z}_{t-1}), \quad \forall \boldsymbol{z}_{1:T} \in \mathbb{R}^{Tm}.$$

This also indicates that $p_{a_t^i}(\boldsymbol{z}_t|\boldsymbol{z}_{t-1}) = p_{\hat{a}_t^j}(\boldsymbol{z}_t|\boldsymbol{z}_{t-1})$ for all $t \geq 2$, $\boldsymbol{z}_t, \boldsymbol{z}_{t-1} \in \mathbb{R}^m$, which can be proved by noticing that $p_a(\boldsymbol{z}_t|\boldsymbol{z}_{t-1})$ are conditional PDFs. To see this, notice that as the joint distributions on $\boldsymbol{z}_{1:T}$ are equal, then the marginal distributions on $\boldsymbol{z}_{1:T-1}$ are also equal:

$$p_{a_1^i}(\boldsymbol{z}_1) \prod_{t=2}^{T-1} p_{a_t^i}(\boldsymbol{z}_t|\boldsymbol{z}_{t-1}) = p_{\hat{a}_1^j}(\boldsymbol{z}_1) \prod_{t=2}^{T-1} p_{\hat{a}_t^j}(\boldsymbol{z}_t|\boldsymbol{z}_{t-1}), \quad \forall \boldsymbol{z}_{1:T-1} \in \mathbb{R}^{(T-1)m},$$

which immediately implies $p_{a_T^i}(\boldsymbol{z}_T|\boldsymbol{z}_{T-1}) = p_{\hat{a}_T^j}(\boldsymbol{z}_T|\boldsymbol{z}_{T-1}), \forall \boldsymbol{z}_{T-1}, \boldsymbol{x}_T \in \mathbb{R}^m$. Similar logic applies to the other time indices $t \geq 1$, which also implies $p_{a_1^i}(\boldsymbol{z}_1) = p_{\hat{a}_1^j}(\boldsymbol{z}_1)$ for all $\boldsymbol{x}_1 \in \mathbb{R}^m$.

Lastly, if there exists $t_1 \neq t_2$ such that $a_{t_1}^i = a_{t_2}^i$ but $\hat{a}_{t_1}^j \neq \hat{a}_{t_2}^j$, then the proved fact that, for any $\boldsymbol{\alpha}, \boldsymbol{\beta} \in \mathbb{R}^m$,

$$p_{\hat{a}_{t_1}^j}(\boldsymbol{z}_{t_1} = \boldsymbol{\beta}|\boldsymbol{z}_{t_1-1} = \boldsymbol{\alpha}) = p_{a_{t_1}^i}(\boldsymbol{z}_{t_1} = \boldsymbol{\beta}|\boldsymbol{z}_{t_1-1} = \boldsymbol{\alpha})$$
$$= p_{a_{t_2}^i}(\boldsymbol{z}_{t_2} = \boldsymbol{\beta}|\boldsymbol{z}_{t_2-1} = \boldsymbol{\alpha})$$
$$= p_{\hat{a}_{t_2}^j}(\boldsymbol{z}_{t_2} = \boldsymbol{\beta}|\boldsymbol{z}_{t_2-1} = \boldsymbol{\alpha}),$$

implies linear dependence of $\mathcal{P}_{\mathcal{A}}$, which contradicts to the assumption that $\mathcal{P}_{\mathcal{A}}^T$ are linearly independent under finite mixtures.

We show the contradiction by assuming the case where $\mathcal{P}_{\mathcal{A}}^{t-1}$ is linearly independent for some $t > 1$, and then we consider the linear independence on $\mathcal{P}_{\mathcal{A}}^t$. We should have

$$\sum_{i,j} \gamma_{ij} p_{a_{1:t-1}^i}(\boldsymbol{z}_{1:t-1}) p_{a_t^j}(\boldsymbol{z}_t|\boldsymbol{z}_{t-1}) = 0, \quad \forall \boldsymbol{z}_{1:t} \in \mathbb{R}^{(t-1)m} \times \mathbb{R}^m,$$

with $\gamma_{ij} = 0, \forall i, j$. We can swap the summations to observe that from linear dependence of $\mathcal{P}_{\mathcal{A}}$, we can get $\gamma_{ij} \neq 0, \forall i$ and some $j$ such that $\sum_j \gamma_{ij} p_{a_t^j}(\boldsymbol{z}_t|\boldsymbol{z}_{t-1}) = 0$.

$$\sum_i \left( \sum_j \gamma_{ij} p_{a_t^j}(\boldsymbol{z}_t|\boldsymbol{z}_{t-1}) \right) p_{a_{1:t-1}^i}(\boldsymbol{z}_{1:t-1}) = 0, \quad \forall \boldsymbol{z}_{1:t} \in \mathbb{R}^{(t-1)m} \times \mathbb{R}^m,$$

which satisfies the equation with $\gamma_{ij} \neq 0$ for some $i$ and $j$ and thus contradicts with the linear independence of $\mathcal{P}_{\mathcal{A}}^t$.

$\square$

## B.2 Linear independence for $T = 2$

Following the strategy as described in the main text, the next step requires us to start from linear independence results for $T = 2$, and then extend to $T > 2$. We therefore prove the following linear independence result.

**Lemma B.1.** *Consider two families $\mathcal{U}_I := \{u_i(\mathbf{y}, \boldsymbol{x})|i \in I\}$ and $\mathcal{V}_J := \{v_j(\boldsymbol{z}, \mathbf{y})|j \in J\}$ with $\boldsymbol{x} \in \mathcal{X}, \mathbf{y} \in \mathbb{R}^{d_y}$ and $\boldsymbol{z} \in \mathbb{R}^{d_z}$. We further assume the following assumptions:*

- *(b1) Positive function values: $u_i(\mathbf{y}, \boldsymbol{x}) > 0$ for all $i \in I, (\mathbf{y}, \boldsymbol{x}) \in \mathbb{R}^{d_y} \times \mathcal{X}$. Similar positive function values assumption applies to $\mathcal{V}_J$: $v_j(\boldsymbol{z}, \mathbf{y}) > 0$ for all $j \in J, (\boldsymbol{z}, \mathbf{y}) \in \mathbb{R}^{d_z} \times \mathbb{R}^{d_y}$.*

- *(b2) Unique indexing: for $\mathcal{U}_I$, $i \neq i' \in I \Leftrightarrow \exists \boldsymbol{x}, \mathbf{y}$ s.t. $u_i(\boldsymbol{x}, \mathbf{y}) \neq u_{i'}(\boldsymbol{x}, \mathbf{y})$. Similar unique indexing assumption applies to $\mathcal{V}_J$;*

(b3) *Linear independence under finite mixtures on specific non-zero measure subsets for $\mathcal{U}_I$: for any non-zero measure subset $\mathcal{Y} \subset \mathbb{R}^{d_y}$, $\mathcal{U}_I$ contains linearly independent functions under finite mixtures on $(\mathbf{y}, \boldsymbol{x}) \in \mathcal{Y} \times \mathcal{X}$.*

(b4) *Linear independence under finite mixtures on specific non-zero measure subsets for $\mathcal{V}_J$: there exists a non-zero measure subset $\mathcal{Y} \subset \mathbb{R}^{d_y}$, such that for any non-zero measure subsets $\mathcal{Y}' \subset \mathcal{Y}$ and $\mathcal{Z} \subset \mathbb{R}^{d_z}$, $\mathcal{V}_J$ contains linearly independent functions under finite mixtures on $(\boldsymbol{z}, \mathbf{y}) \in \mathcal{Z} \times \mathcal{Y}'$;*

(b5) *Linear dependence under finite mixtures for subsets of functions in $\mathcal{V}_J$ implies repeating functions: for any $\boldsymbol{\beta} \in \mathbb{R}^{d_y}$, any non-zero measure subset $\mathcal{Z} \subset \mathbb{R}^{d_z}$ and any subset $J_0 \subset J$ such that $|J_0| < +\infty$, $\{v_j(\boldsymbol{z}, \mathbf{y} = \boldsymbol{\beta})|j \in J_0\}$ contains linearly dependent functions on $\boldsymbol{z} \in \mathcal{Z}$ only if $\exists \, j \neq j' \in J_0$ such that $v_j(\boldsymbol{z}, \boldsymbol{\beta}) = v_{j'}(\boldsymbol{z}, \boldsymbol{\beta})$ for all $\boldsymbol{z} \in \mathbb{R}^{d_z}$.*

(b6) *Continuity for $\mathcal{V}_J$: for any $j \in J$, $v_j(\boldsymbol{z}, \mathbf{y})$ is continuous in $\mathbf{y} \in \mathbb{R}^{d_y}$.*

*Then for any non-zero measure subset $\mathcal{Z} \subset \mathbb{R}^{d_z}$, $\mathcal{U}_I \otimes \mathcal{V}_J := \{v_j(\boldsymbol{z}, \mathbf{y})u_i(\mathbf{y}, \boldsymbol{x})|i \in I, j \in J\}$ contains linear independent functions defined on $(\boldsymbol{x}, \mathbf{y}, \boldsymbol{z}) \in \mathcal{X} \times \mathbb{R}^{d_y} \times \mathcal{Z}$.*

*Proof.* Assume this sufficiency statement is false, then there exist a non-zero measure subset $\mathcal{Z} \subset \mathbb{R}^{d_z}$, $S_0 \subset I \times J$ with $|S_0| < +\infty$ and a set of non-zero values $\{\gamma_{ij} \in \mathbb{R}|(i,j) \in S_0\}$, such that

$$\sum_{(i,j) \in S_0} \gamma_{ij}v_j(\boldsymbol{z}, \mathbf{y})u_i(\mathbf{y}, \boldsymbol{x}) = 0, \quad \forall (\boldsymbol{x}, \mathbf{y}, \boldsymbol{z}) \in \mathcal{X} \times \mathbb{R}^{d_y} \times \mathcal{Z}. \tag{19}$$

Note that the choices of $S_0$ and $\gamma_{ij}$ are independent of any $\boldsymbol{x}, \mathbf{y}, \boldsymbol{z}$ values, but might be dependent on $\mathcal{Z}$. By assumptions (b1), the index set $S_0$ contains at least 2 different indices $(i, j)$ and $(i', j')$. In particular, $S_0$ contains at least 2 different indices $(i, j)$ and $(i', j')$ with $j \neq j'$, otherwise we can extract the common term $v_j(\boldsymbol{z}, \mathbf{y})$ out:

$$\sum_{(i,j) \in S_0} \gamma_{ij}v_j(\boldsymbol{z}, \mathbf{y})u_i(\mathbf{y}, \boldsymbol{x}) = v_j(\boldsymbol{z}, \mathbf{y})\left(\sum_{i:(i,j) \in S_0} \gamma_{ij}u_i(\mathbf{y}, \boldsymbol{x})\right) = 0, \quad \forall (\boldsymbol{x}, \mathbf{y}, \boldsymbol{z}) \in \mathcal{X} \times \mathbb{R}^{d_y} \times \mathcal{Z},$$

and as there exist at least 2 different indices $(i', j)$ and $(i, j)$ in $S_0$, we have at least one $i' \neq i$, and the above equation contradicts to assumptions (b1) - (b3).

Now define $J_0 = \{j \in A|\exists (i, j) \in S_0\}$ the set of all possible $j$ indices that appear in $S_0$, and from $|S_0| < +\infty$ we have $|J_0| < +\infty$ as well. We rewrite the linear combination equation (Eq. (19)) for any $\boldsymbol{\beta} \in \mathbb{R}^{d_y}$ as

$$\sum_{j \in J_0}\left(\sum_{i:(i,j) \in S_0} \gamma_{ij}u_i(\mathbf{y} = \boldsymbol{\beta}, \boldsymbol{x})\right)v_j(\boldsymbol{z}, \mathbf{y} = \boldsymbol{\beta}) = 0, \quad \forall (\boldsymbol{x}, \boldsymbol{z}) \in \mathcal{X} \times \mathcal{Z}. \tag{20}$$

From assumption (b3) we know that the set $\mathcal{Y}_0 := \{\boldsymbol{\beta} \in \mathbb{R}^{d_y}|\sum_{i:(i,j) \in S_0} \gamma_{ij}u_i(\mathbf{y} = \boldsymbol{\beta}, \boldsymbol{x}) = 0, \forall \boldsymbol{x} \in \mathcal{X}\}$ can only have zero measure in $\mathbb{R}^{d_y}$. Write $\mathcal{Y} \subset \mathbb{R}^{d_y}$ the non-zero measure subset defined by assumption (b4), we have $\mathcal{Y}_1 := \mathcal{Y}\backslash\mathcal{Y}_0 \subset \mathcal{Y}$ also has non-zero measure and satisfies assumption (b4). Combined with assumption (b1), we have for each $\boldsymbol{\beta} \in \mathcal{Y}_1$, there exists $\boldsymbol{x} \in \mathcal{X}$ such that $\sum_{i:(i,j) \in S_0} \gamma_{ij}u_i(\mathbf{y} = \boldsymbol{\beta}, \boldsymbol{x}) \neq 0$ for at least two $j$ indices in $J_0$. This means for each $\boldsymbol{\beta} \in \mathcal{Y}_1$, $\{v_j(\boldsymbol{z}, \mathbf{y} = \boldsymbol{\beta})|j \in J_0\}$ contains linearly dependent functions on $\boldsymbol{z} \in \mathcal{Z}$. Now under assumption (b5), we can split the index set $J_0$ into subsets indexed by $k \in K(\boldsymbol{\beta})$ as follows, such that within each index subset $J_k(\boldsymbol{\beta})$ the functions with the corresponding indices are equal:

$$J_0 = \cup_{k \in K(\boldsymbol{\beta})}J_k(\boldsymbol{\beta}), \quad J_k(\boldsymbol{\beta}) \cap J_{k'}(\boldsymbol{\beta}) = \emptyset, \forall k \neq k' \in K(\boldsymbol{\beta}),$$
$$j \neq j' \in J_k(\boldsymbol{\beta}) \quad \Leftrightarrow \quad v_j(\boldsymbol{z}, \mathbf{y} = \boldsymbol{\beta}) = v_{j'}(\boldsymbol{z}, \mathbf{y} = \boldsymbol{\beta}), \quad \forall \boldsymbol{z} \in \mathcal{Z}. \tag{21}$$

Then we can rewrite Eq. (20) for any $\boldsymbol{\beta} \in \mathcal{Y}_1$ as

$$\sum_{k \in K(\boldsymbol{\beta})}\left(\sum_{j \in J_k(\boldsymbol{\beta})}\sum_{i:(i,j) \in S_0} \gamma_{ij}u_i(\mathbf{y} = \boldsymbol{\beta}, \boldsymbol{x})v_j(\boldsymbol{z}, \mathbf{y} = \boldsymbol{\beta})\right) = 0, \quad \forall (\boldsymbol{x}, \boldsymbol{z}) \in \mathcal{X} \times \mathcal{Z}. \tag{22}$$

Recall from Eq. (21) that $v_j(\boldsymbol{z}, \boldsymbol{y} = \boldsymbol{\beta})$ and $v_{j'}(\boldsymbol{z}, \boldsymbol{y} = \boldsymbol{\beta})$ are the same functions on $\boldsymbol{z} \in \mathcal{Z}$ iff. $j \neq j'$ are in the same index set $J_k(\boldsymbol{\beta})$. This means if Eq. (19) holds, then for any $\boldsymbol{\beta} \in \mathcal{Y}_1$, under assumptions (b1) and (b5),

$$\sum_{j \in J_k(\boldsymbol{\beta})} \sum_{i:(i,j) \in S_0} \gamma_{ij} u_i(\mathbf{y} = \boldsymbol{\beta}, \boldsymbol{x}) = 0, \quad \forall \boldsymbol{x} \in \mathbb{R}^d, \quad k \in K(\boldsymbol{\beta}). \tag{23}$$

Define $C(\boldsymbol{\beta}) = \min_k |J_k(\boldsymbol{\beta})|$ the minimum cardinality count for $j$ indices in the $J_k(\boldsymbol{\beta})$ subsets. Choose $\boldsymbol{\beta}^* \in \arg\min_{\boldsymbol{\beta} \in \mathcal{Y}_1} C(\boldsymbol{\beta})$:

1. We have $C(\boldsymbol{\beta}^*) < |J_0|$ and $|K(\boldsymbol{\beta}^*)| \geq 2$. Otherwise for all $j \neq j' \in J_0$ we have $v_j(\boldsymbol{z}, \mathbf{y} = \boldsymbol{\beta}) = v_{j'}(\boldsymbol{z}, \mathbf{y} = \boldsymbol{\beta})$ for all $\boldsymbol{z} \in \mathcal{Z}$ and $\boldsymbol{\beta} \in \mathcal{Y}_1$, so that they are linearly dependent on $(\boldsymbol{z}, \mathbf{y}) \in \mathcal{Z} \times \mathcal{Y}_1$, a contradiction to assumption (b4) by setting $\mathcal{Y}' = \mathcal{Y}_1$.

2. Now assume $|J_1(\boldsymbol{\beta}^*)| = C(\boldsymbol{\beta}^*)$ w.l.o.g.. From assumption (b5), we know that for any $j \in J_1(\boldsymbol{\beta}^*)$ and $j' \in J_0 \backslash J_1(\boldsymbol{\beta}^*)$, $v_j(\boldsymbol{z}, \mathbf{y} = \boldsymbol{\beta}) = v_{j'}(\boldsymbol{z}, \mathbf{y} = \boldsymbol{\beta})$ only on zero measure subset of $\mathcal{Z}$ at most. Then as $|J_0| < +\infty$ and $\mathcal{Z} \subset \mathbb{R}^{d_z}$ has non-zero measure, there exist $\boldsymbol{z}_0 \in \mathcal{Z}$ and $\delta > 0$ such that

$$|v_j(\boldsymbol{z} = \boldsymbol{z}_0, \mathbf{y} = \boldsymbol{\beta}^*) - v_{j'}(\boldsymbol{z} = \boldsymbol{z}_0, \mathbf{y} = \boldsymbol{\beta}^*)| \geq \delta, \quad \forall j \in J_1(\boldsymbol{\beta}^*), \forall j' \in J_0 \backslash J_1(\boldsymbol{\beta}^*).$$

   Under assumption (b6), there exists $\epsilon(j) > 0$ such that we can construct an $\epsilon$-ball $B_{\epsilon(j)}(\boldsymbol{\beta}^*)$ using $\ell_2$-norm, such that

$$|v_j(z = \boldsymbol{z}_0, \mathbf{y} = \boldsymbol{\beta}^*) - v_j(z = \boldsymbol{z}_0, \mathbf{y} = \boldsymbol{\beta})| \leq \delta/3, \quad \forall \boldsymbol{\beta} \in B_{\epsilon(j)}(\boldsymbol{\beta}^*).$$

   Choosing a suitable $0 < \epsilon \leq \min_{j \in J_0} \epsilon(j)$ (note that $\min_{j \in J_0} \epsilon(j) > 0$ as $|J_0| < +\infty$) and constructing an $\ell_2$-norm-based $\epsilon$-ball $B_\epsilon(\boldsymbol{\beta}^*) \subset \mathcal{Y}_1$, we have for all $j \in J_1(\boldsymbol{\beta}^*), j' \in J_0 \backslash J_1(\boldsymbol{\beta}^*), j' \notin J_1(\boldsymbol{\beta})$ for all $\boldsymbol{\beta} \in B_\epsilon(\boldsymbol{\beta}^*)$ due to

$$|v_j(\boldsymbol{z} = \boldsymbol{z}_0, \mathbf{y} = \boldsymbol{\beta}) - v_{j'}(\boldsymbol{z} = \boldsymbol{z}_0, \mathbf{y} = \boldsymbol{\beta})| \geq \delta/3, \quad \forall \boldsymbol{\beta} \in B_\epsilon(\boldsymbol{\beta}^*).$$

   So this means for the split $\{J_k(\boldsymbol{\beta})\}$ of any $\boldsymbol{\beta} \in B_\epsilon(\boldsymbol{\beta}^*)$, we have $J_1(\boldsymbol{\beta}) \subset J_1(\boldsymbol{\beta}^*)$ and therefore $|J_1(\boldsymbol{\beta})| \leq |J_1(\boldsymbol{\beta}^*)|$. Now by definition of $\boldsymbol{\beta}^* \in \arg\min_{\boldsymbol{\beta} \in \mathcal{Y}} C(\boldsymbol{\beta})$ and $|J_1(\boldsymbol{\beta}^*)| = C(\boldsymbol{\beta}^*)$, we have $J_1(\boldsymbol{\beta}) = J_1(\boldsymbol{\beta}^*)$ for all $\boldsymbol{\beta} \in B_\epsilon(\boldsymbol{\beta}^*)$.

3. One can show that $|J_1(\boldsymbol{\beta}^*)| = 1$, otherwise by definition of the split (Eq. (21)) and the above point, there exists $j \neq j' \in J_1(\boldsymbol{\beta}^*)$ such that $v_j(\boldsymbol{z}, \mathbf{y} = \boldsymbol{\beta}) = v_{j'}(\boldsymbol{z}, \mathbf{y} = \boldsymbol{\beta})$ for all $\boldsymbol{z} \in \mathcal{Z}$ and $\boldsymbol{\beta} \in B_\epsilon(\boldsymbol{\beta}^*)$, a contradiction to assumption (b4) by setting $\mathcal{Y}' = B_\epsilon(\boldsymbol{\beta}^*)$. Now assume that $j \in J_1(\boldsymbol{\beta}^*)$ is the only index in the subset, then the fact proved in the above point that $J_1(\boldsymbol{\beta}) = J_1(\boldsymbol{\beta}^*)$ for all $\boldsymbol{\beta} \in B_\epsilon(\boldsymbol{\beta}^*)$ means

$$\sum_{i:(i,j) \in S_0} \gamma_{ij} u_i(\mathbf{y} = \boldsymbol{\beta}, \boldsymbol{x}) = 0, \quad \forall \boldsymbol{x} \in \mathcal{X}, \quad \forall \boldsymbol{\beta} \in B_\epsilon(\boldsymbol{\beta}^*),$$

   again a contradiction to assumption (b3) by setting $\mathcal{Y} = B_\epsilon(\boldsymbol{\beta}^*)$.

The above 3 points indicate that Eq. (23) cannot hold for all $\boldsymbol{\beta} \in \mathcal{Y}_1$ (and therefore for all $\boldsymbol{\beta} \in \mathcal{Y}$) under assumptions (b3) - (b6), therefore a contradiction is reached. □

### B.3 LINEAR INDEPENDENCE IN THE NON-PARAMETRIC CASE

The previous result can be used to show conditions for the linear independence of the joint distribution family $\mathcal{P}_{\mathcal{A}}^T$ in the non-parametric case.

**Theorem B.2.** *Define the following joint distribution family*

$$\left\{ p_{a_1, a_{2:T}}(\boldsymbol{z}_{1:T}) = p_{a_1}(\boldsymbol{z}_1) \prod_{t=2}^{T} p_{a_t}(\boldsymbol{z}_t | \boldsymbol{z}_{t-1}), \quad p_{a_1} \in \Pi_{\mathcal{A}}, \quad p_{a_t} \in \mathcal{P}_{\mathcal{A}}, t = 2, ..., T \right\},$$

*and assume $\Pi_{\mathcal{A}}$ and $\mathcal{P}_{\mathcal{A}}$ satisfy assumptions (b1)-(b6) as follows,*

*(c1) $\Pi_{\mathcal{A}}$ and $\mathcal{P}_{\mathcal{A}}$ satisfy (b1) and (b2): positive function values and unique indexing,*

*(c2)* $\Pi_{\mathcal{A}}$ *satisfies (b3), and*

*(c3)* $\mathcal{P}_{\mathcal{A}}$ *satisfies (b4)-(b6).*

*Then the following statement holds: For any $T \geq 2$ and any subset $\mathcal{Z} \subset \mathbb{R}^m$ The joint distribution family contains linearly independent distributions for $(\boldsymbol{z}_{1:T-1}, \boldsymbol{z}_T) \in \mathbb{R}^{(T-1)m} \times \mathcal{Z}$.*

*Proof.* We proceed to prove the statement by induction as follows. Here we set $I = J = \mathcal{A}$.

(1) $T = 2$: The result can be proved using Lemma B.1 by setting in the proof, $u_i(\mathbf{y} = \boldsymbol{z}_1, \boldsymbol{x} = \boldsymbol{z}_0) = \pi_i(\boldsymbol{z}_1), i \in \mathcal{A}$ and $v_j(\boldsymbol{z} = \boldsymbol{z}_2, \mathbf{y} = \boldsymbol{z}_1) = p_j(\boldsymbol{z}_2|\boldsymbol{z}_1), j \in \mathcal{A}$.

(2) $T > 2$: Assume the statement holds for the joint distribution family when $T = \tau - 1$. Note that we can write $p_{a_{1:\tau}}(\boldsymbol{z}_{1:\tau})$ as

$$p_{a_{1:\tau}}(\boldsymbol{z}_{1:\tau}) = p_{a_1, a_{2:\tau-1}}(\boldsymbol{x}_{1:\tau-1}) p_{a_\tau}(\boldsymbol{z}_\tau | \boldsymbol{z}_{\tau-1}).$$

Then the statement for $T = \tau$ can be proved using Lemma B.1 by setting $u_i(\mathbf{y} = \boldsymbol{z}_{\tau-1}, \boldsymbol{x} = \boldsymbol{z}_{1:\tau-2}) = p_{a_{1:\tau-1}}(\boldsymbol{z}_{1:\tau-1}), i = a_{1:\tau-1}$, and $v_j(\boldsymbol{z} = \boldsymbol{z}_\tau, \mathbf{y} = \boldsymbol{z}_{\tau-1}) = p_{a_\tau}(\boldsymbol{z}_\tau | \boldsymbol{z}_{\tau-1}), j = a_\tau$. Note that the family spanned with $p_{a_{1:\tau-1}}(\boldsymbol{z}_{1:\tau-1}), i = a_{1:\tau-1}$ satisfies (b1) and (b2) from $\Pi_{\mathcal{A}}$ and $\mathcal{P}_{\mathcal{A}}$ directly, and (b3) from the induction hypothesis. $\square$

With the result above, one can construct identifiable Markov Switching Models as long as the initial and transition distributions are consistent with assumptions (c1)-(c3).

## B.4 LINEAR INDEPENDENCE IN THE NON-LINEAR GAUSSIAN CASE

As described, in the final step of the proof we explore properties of the Gaussian transition and initial distribution families (Eqs. (6) and (8) respectively). The unique indexing assumption of the Gaussian transition family (Eq. (7)) implies linear independence as shown below.

**Proposition B.2.** *Functions in $\mathcal{G}_A$ are linearly independent on variables $(\boldsymbol{z}_t, \boldsymbol{z}_{t-1})$ if the unique indexing assumption (Eq. (7)) holds.*

*Proof.* Assume the statement is false, then there exists $\mathcal{A}_0 \subset \mathcal{A}$ and a set of non-zero values $\{\gamma_a | a \in \mathcal{A}_0\}$, such that

$$\sum_{a \in \mathcal{A}_0} \gamma_a \mathcal{N}(\boldsymbol{z}_t; \boldsymbol{m}(\boldsymbol{z}_{t-1}, a), \boldsymbol{\Sigma}(\boldsymbol{z}_{t-1}, a)) = 0, \quad \forall \boldsymbol{z}_t, \boldsymbol{z}_{t-1} \in \mathbb{R}^m.$$

In particular, this equality holds for any $\boldsymbol{z}_{t-1} \in \mathbb{R}^m$, meaning that a weighted sum of Gaussian distributions (defined on $\boldsymbol{z}_t$) equals to zero. Note that Yakowitz & Spragins (1968) proved that multivariate Gaussian distributions with different means and/or covariances are linearly independent. Therefore the equality above implies for any $\boldsymbol{z}_{t-1}$

$$\boldsymbol{m}(\boldsymbol{z}_{t-1}, a) = \boldsymbol{m}(\boldsymbol{z}_{t-1}, a') \quad \text{and} \quad \boldsymbol{\Sigma}(\boldsymbol{z}_{t-1}, a) = \boldsymbol{\Sigma}(\boldsymbol{z}_{t-1}, a') \quad \forall a, a' \in A_0, a \neq a',$$

a contradiction to the unique indexing assumption. $\square$

We now draw some connections from the previous Gaussian families to assumptions (b1-b6) in Lemma B.1.

**Proposition B.3.** *The conditional Gaussian distribution family $\mathcal{G}_A$ (Eq. (6), under the unique indexing assumption (Eq. (7), satisfies assumptions (b1), (b2) and (b5) in Lemma B.1, if we define $\mathcal{V}_J := \mathcal{G}_A, \boldsymbol{z} := \boldsymbol{z}_t$ and $\mathbf{y} := \boldsymbol{z}_{t-1}$.*

**Proposition B.4.** *The initial Gaussian distribution family $\mathcal{I}_A$ (Eq. (8), under the unique indexing assumption (Eq. (9), satisfies assumptions (b1), (b2) and (b3) in Lemma B.1, if we define $\mathcal{U}_I := \mathcal{I}_A, \mathbf{y} := \boldsymbol{z}_1$ and $\boldsymbol{x} = \mathcal{X} = \emptyset$.*

To see why $\mathcal{G}_A$ satisfies (b5), notice Gaussian densities are analytic in $\boldsymbol{z}_t$. Similar ideas apply to show that $\mathcal{I}_A$ satisfies (b3). With the previous results, we can rewrite the previous result for the non-linear Gaussian case.

**Theorem B.3.** *Define the following joint distribution family under the non-linear Gaussian model*

$$\mathcal{P}_{\mathcal{A}}^T = \left\{ p_{a_1, a_{2:T}}(\boldsymbol{z}_{1:T}) = p_{a_1}(\boldsymbol{z}_1) \prod_{t=2}^T p_{a_t}(\boldsymbol{z}_t | \boldsymbol{z}_{t-1}), \right.$$

$$\left. a_t \in \mathcal{A}, \quad p_{a_1} \in \mathcal{I}_{\mathcal{A}}, \quad p_{a_t} \in \mathcal{G}_{\mathcal{A}}, \quad t = 2, ..., T \right\}, \quad (24)$$

*with $\mathcal{G}_{\mathcal{A}}$, $\mathcal{I}_{\mathcal{A}}$ defined by Eqs. (6), (8) respectively. Assume:*

(d1) *Unique indexing for $\mathcal{G}_{\mathcal{A}}$ and $\mathcal{I}_{\mathcal{A}}$: Eqs. (7), (9) hold;*

(d2) *Continuity for the conditioning input: distributions in $\mathcal{G}_{\mathcal{A}}$ are continuous w.r.t. $\boldsymbol{z}_{t-1} \in \mathbb{R}^m$;*

(d3) *Zero-measure intersection in certain region: there exists a non-zero measure set $\mathcal{X}_0 \subset \mathbb{R}^m$ s.t. $\{\boldsymbol{z}_{t-1} \in \mathcal{X}_0 | \boldsymbol{m}(\boldsymbol{z}_{t-1}, a) = \boldsymbol{m}(\boldsymbol{z}_{t-1}, a'), \boldsymbol{\Sigma}(\boldsymbol{z}_{t-1}, a) = \boldsymbol{\Sigma}(\boldsymbol{z}_{t-1}, a')\}$ has zero measure, for any $a \neq a'$;*

*Then, the joint distribution family contains linearly independent distributions for $(\boldsymbol{z}_{1:T-1}, \boldsymbol{z}_T) \in \mathbb{R}^{(T-1)m} \times \mathbb{R}^m$.*

*Proof.* Note that assumptions (b1) - (b3) and (b5) are satisfied due to Propositions B.3 and B.4, and assumptions (b6) and (d2) are equivalent, and assumption (b4) holds due to assumption (d3). To show (d3) $\implies$ (b4), We first define $\mathcal{V}_J := \mathcal{G}_{\mathcal{A}}$, $\boldsymbol{z} := \boldsymbol{z}_t$, and $\mathbf{y} := \boldsymbol{z}_{t-1}$ from Prop. B.3. From (d3), $\mathcal{Y} := \mathcal{X}_0$ and note that $\mathcal{V}_J$ contains linear independent functions on $(\boldsymbol{z}, \mathbf{y}) \in \mathcal{M} \subset \mathcal{Z} \times \mathcal{Y}$ if $\mathcal{M} \neq \mathcal{Z} \times \mathcal{D}$, where $\mathcal{D}$ denotes the set where intersection of moments happen within $\mathcal{Y}$. Also by (d3), $\mathcal{D}$ has measure zero and thus, (b4) holds since $\mathcal{Y}'$ is a non-zero measure set.

Then, the statement holds by Theorem B.2. $\qquad\square$

### B.5 Concluding the proof

Below we formally state the proof for Theorem 3.1 by further assuming parametrisations of the Gaussian moments via analytic functions, i.e. assumption (a2).

*Proof.* (a1) and (d1) are equivalent. Following (a2), let $\boldsymbol{m}(\cdot, a) : \mathbb{R}^m \to \mathbb{R}^m$ be a multivariate analytic function, which allows a multivariate Taylor expansion. The corresponding Taylor expansion of $\boldsymbol{m}(\cdot, a)$ implies (d2). Similar logic applies to $\boldsymbol{\Sigma}(\cdot, a)$. To show (d3), we note for any $a \neq a'$ the set of intersection of moments, i.e. $\{\boldsymbol{z} \in \mathbb{R}^m | \boldsymbol{m}(\boldsymbol{z}, a) = \boldsymbol{m}(\boldsymbol{z}, a'), \boldsymbol{\Sigma}(\boldsymbol{z}_{t-1}, a) = \boldsymbol{\Sigma}(\boldsymbol{z}_{t-1}, a')\}$ can be separated as the intersection of the sets $\{\boldsymbol{z} \in \mathbb{R}^m | \boldsymbol{m}(\boldsymbol{x}, a) = \boldsymbol{m}(\boldsymbol{z}, a')\}$ and $\{\boldsymbol{z} \in \mathbb{R}^m | \boldsymbol{\Sigma}(\boldsymbol{z}_{t-1}, a) = \boldsymbol{\Sigma}(\boldsymbol{z}_{t-1}, a')\}$. Wlog, the set $\{\boldsymbol{z} \in \mathbb{R}^m | \boldsymbol{m}(\boldsymbol{z}, a) = \boldsymbol{m}(\boldsymbol{z}, a')\}$ is the zero set of an analytic function $\boldsymbol{f} := \boldsymbol{m}(\cdot, a) - \boldsymbol{m}(\cdot, a')$. Proposition 0 in Mityagin (2015) shows that the zero set of a real analytic function on $\mathbb{R}^m$ has zero measure unless $\boldsymbol{f}$ is identically zero. Hence, the intersection of moments has zero measure from our premise of unique indexing.

Since (d1-d3) are satisfied, by Theorem B.3 we have linear independence of the joint distribution family, which by Theorem B.1 implies identifiability of the MSM in the sense of Def. 3.1. $\qquad\square$

## C Properties of the MSM

In this section, we present some results involving MSMs for convenience. First, we start with the result on first-order stationary Markov chains presented in section 3.1.

**Corollary C.1.** *Consider an identifiable MSM from Def. 3.1, where the prior distribution of the states $p(\mathbf{s}_{1:T})$ follows a first-order stationary Markov chain, i.e $p(\mathbf{s}_{1:T}) = \pi_{s_1} Q_{s_1, s_2} \ldots Q_{s_{T-1}, s_T}$, where $\boldsymbol{\pi}$ denotes the initial distribution: $p(s_1 = k) = \pi_k$, and $Q$ denotes the transition matrix: $p(s_t = k | s_{t-1} = l) = Q_{l,k}$. Then, $\boldsymbol{\pi}$ and $Q$ are identifiable up to permutations.*

*Proof.* From Def. 3.1, we have $K = \hat{K}$ and for every $1 \leq i \leq K^T$ there is some $1 \leq j \leq \hat{K}^T$ such that $c_i = \hat{c}_j$. Now writing $\mathbf{s}_{1:T} = (s_1^i, ..., s_T^i) = \varphi(i)$ and $\hat{\mathbf{s}}_{1:T} = (\hat{s}_1^j, ..., \hat{s}_T^j) = \varphi(j)$, we have

$$c_i = \pi_{s_1^i} Q_{s_1^i, s_2^i} \ldots Q_{s_{T-1}^i, s_T^i} = \hat{\pi}_{\hat{s}_1^j} \hat{Q}_{\hat{s}_1^j, \hat{s}_2^j} \ldots \hat{Q}_{\hat{s}_{T-1}^j, \hat{s}_T^j} = \hat{c}_j.$$

Since the joint distributions are equal on $\mathbf{s}_{1:T}$, they must also be equal on $\mathbf{s}_{1:T-1}$. Therefore, we have $Q_{s_{T-1}^i, s_T^i} = \hat{Q}_{\hat{s}_{T-1}^j, \hat{s}_T^j}$. Similar logic applies to $t \geq 1$, which also implies $\pi_{s_1^i} = \hat{\pi}_{\hat{s}_1^j}$.

For $t = 1$, the above implies that for each $i \in \{1, ..., K\}$, there exists some $j \in \{1, ..., K\}$, such that $\pi_i = \hat{\pi}_j$. This indicates permutation equivalence. We denote $\sigma(\cdot)$ as such permutation function, so that for all $i \in \{1, ..., K\}$ and the corresponding $j$, $\pi_i = \tilde{\pi}_j = \pi_{\sigma(j)}$.

For $t > 1$, the previous implication gives us that for $i, j \in \{1, \ldots, K\}, \exists k, l \in \{1, \ldots, K\}$ such that $Q_{i,j} = \hat{Q}_{k,l}$. Following the previous logic, we can define permutations that match $Q$ and $\hat{Q}$: $i = \sigma_1(k), j = \sigma_2(l)$. We observe from the second requirement in Def. 3.1 that if $i = j$, then $k = l$ and since $\sigma_1(k) = \sigma_2(l)$, we have that the permutations $\sigma_1(\cdot)$ and $\sigma_2(\cdot)$ must be equal. Therefore, we have $Q_{i,j} = \hat{Q}_{k,l} = Q_{\sigma_1(k), \sigma_1(l)}$.

Finally, we can use the second requirement in Def. 3.1 to see that $\sigma(\cdot)$ and $\sigma_1(\cdot)$ must be equal. $\qquad \square$

**Proposition C.1.** *The joint distribution of the Markov Switching Model with Gaussian analytic transitions and Gaussian initial distributions is closed under factored invertible affine transformations, $\mathbf{z}'_{1:T} = \mathcal{H}(\mathbf{z}_{1:T})$: $\mathbf{z}'_t = A\mathbf{z}_t + \mathbf{b}$, $1 \leq t \leq T$.*

*Proof.* Consider the following affine transformation $\mathbf{z}'_t = A\mathbf{z}_t + \mathbf{b}$ for $1 \leq t \leq T$, and the joint distribution of a Markov Switching Model with $T$ timesteps

$$p(\mathbf{z}_{1:T}) = \sum_{\mathbf{s}_{1:T}} p(\mathbf{s}_{1:T}) p(\mathbf{z}_1 | s_1) \prod_{t=2}^{T} p(\mathbf{z}_t | \mathbf{z}_{t-1}, s_t),$$

where we denote the initial distribution as $p(\mathbf{z}_1 | s_1 = i) = \mathcal{N}(\mathbf{z}_1; \boldsymbol{\mu}(i), \boldsymbol{\Sigma}_1(i))$ and the transition distribution as $p(\mathbf{z}_t | \mathbf{z}_{t-1}, s_t = i) = \mathcal{N}(\mathbf{z}_t; \mathbf{m}(\mathbf{z}_{t-1}, i), \boldsymbol{\Sigma}(\mathbf{z}_{t-1}, i))$. We need to show that the distribution still consists of Gaussian initial distributions and Gaussian analytic transitions. Let us consider the change of variables rule, which we apply to $p(\mathbf{z}_{1:T})$

$$p_{\mathbf{z}'_{1:T}}(\mathbf{z}'_{1:T}) = \frac{p_{\mathbf{z}_{1:T}} \left( A_{1:T}^{-1} \left( \mathbf{z}'_{1:T} - \mathbf{b}_{1:T} \right) \right)}{\det(A_{1:T})},$$

where we use the subscript $\mathbf{z}'_{1:T}$ to indicate the probability distribution in terms of $\mathbf{z}'_{1:T}$, but we drop it for simplicity. Note that the inverse of a block diagonal matrix can be computed as the inverse of each block, and we use similar properties for the determinant, i.e. $\det(A_{1:T}) = \det(A) \cdots \det(A)$.

The distribution in terms of the transformed variable is expressed as follows:

$$p(\boldsymbol{z}'_{1:T}) = \sum_{s_{1:T}} p(s_{1:T}) \frac{p\left(A^{-1}\left(\boldsymbol{z}'_1 - \mathbf{b}\right)|s_1\right)}{\det(A)} \prod_{t=2}^{T} \frac{p\left(A^{-1}\left(\boldsymbol{z}'_t - \mathbf{b}\right)\mid\left(A^{-1}\left(\boldsymbol{z}'_{t-1} - \mathbf{b}\right)\right), s_t\right)}{\det(A)}$$

$$= \sum_{i_1,\ldots,i_T} p\left(s_{1:T} = \{i_1, \ldots, i_T\}\right) \mathcal{N}\left(\boldsymbol{z}'_1; A\boldsymbol{\mu}(i_1) + \mathbf{b}, A\boldsymbol{\Sigma}_1(i_1)A^T\right)$$

$$\prod_{t=2}^{T} \frac{1}{\sqrt{(2\pi)^m \det\left(A\boldsymbol{\Sigma}\left(A^{-1}\left(\boldsymbol{z}'_{t-1} - \mathbf{b}\right), i_t\right)A^T\right)}}$$

$$\exp\left(-\frac{1}{2}\left(A^{-1}\left(\boldsymbol{z}'_t - \mathbf{b}\right) - \mathbf{m}\left(A^{-1}\left(\boldsymbol{z}'_{t-1} - \mathbf{b}\right), i_t\right)\right)^T\right.$$

$$\left.\boldsymbol{\Sigma}\left(A^{-1}\left(\boldsymbol{z}'_{t-1} - \mathbf{b}\right), i_t\right)^{-1}\left(A^{-1}\left(\boldsymbol{z}'_t - \mathbf{b}\right) - \mathbf{m}\left(A^{-1}\left(\boldsymbol{z}'_{t-1} - \mathbf{b}\right), i_t\right)\right)\right)$$

$$= \sum_{i_1,\ldots,i_T} p\left(s_{1:T} = \{i_1, \ldots, i_T\}\right) \mathcal{N}\left(\boldsymbol{z}'_1; A\boldsymbol{\mu}(i_1) + \mathbf{b}, A\boldsymbol{\Sigma}_1(i_1)A^T\right)$$

$$\prod_{t=2}^{T} \frac{1}{\sqrt{(2\pi)^m \det\left(A\boldsymbol{\Sigma}\left(A^{-1}\left(\boldsymbol{z}'_{t-1} - \mathbf{b}\right), i_t\right)A^T\right)}}$$

$$\exp\left(-\frac{1}{2}\left(\boldsymbol{z}'_t - A\mathbf{m}\left(A^{-1}\left(\boldsymbol{z}'_{t-1} - \mathbf{b}\right), i_t\right) - \mathbf{b}\right)^T\right.$$

$$\left.A^{-1}\boldsymbol{\Sigma}\left(A^{-1}\left(\boldsymbol{z}'_{t-1} - \mathbf{b}\right), i_t\right)^{-1}A^{-T}\left(\boldsymbol{z}'_t - A\mathbf{m}\left(A^{-1}\left(\boldsymbol{z}'_{t-1} - \mathbf{b}\right), i_t\right) - \mathbf{b}\right)\right)$$

$$= \sum_{i_1,\ldots,i_T} p\left(s_{1:T} = \{i_1, \ldots, i_T\}\right) \mathcal{N}\left(\boldsymbol{z}'_1; A\boldsymbol{\mu}(i_1) + \mathbf{b}, A\boldsymbol{\Sigma}_1(i_1)A^T\right)$$

$$\prod_{t=2}^{T} \mathcal{N}\left(\boldsymbol{z}'_t; A\mathbf{m}\left(A^{-1}\left(\boldsymbol{z}'_{t-1} - \mathbf{b}\right), i_t\right) + \mathbf{b}, A\boldsymbol{\Sigma}\left(A^{-1}\left(\boldsymbol{z}'_{t-1} - \mathbf{b}\right), i_t\right)A^T\right)$$

We observe that the resulting distribution is a Markov Switching Model with changes in the Gaussian initial and transition distributions, where the analytic transitions are transformed as follows: $m'(\boldsymbol{z}'_{t-1}, i_t) = A\mathbf{m}\left(A^{-1}\left(\boldsymbol{z}'_{t-1} - \mathbf{b}\right), i_t\right) + \mathbf{b}$, and $\boldsymbol{\Sigma}'(\boldsymbol{z}'_{t-1}, i_t) = A\boldsymbol{\Sigma}\left(A^{-1}\left(\boldsymbol{z}'_{t-1} - \mathbf{b}\right), i_t\right)A^T$ for any $i_t \in \{1, \ldots, K\}$. $\square$

## D    PROOF OF SDS IDENTIFIABILITY

### D.1    PRELIMINARIES

We need to introduce some definitions and results that will be used in the proof. These have been previously defined in Kivva et al. (2022).

**Definition D.1.** *Let $D_0 \subseteq D \subseteq \mathbb{R}^n$ be open sets. Let $f_0 : D_0 \to \mathbb{R}$. We say that an analytic function $f : D \to \mathbb{R}$ is an analytic continuation of $f_0$ onto $D$ if $f(\boldsymbol{x}) = f_0(\boldsymbol{x})$ for every $\boldsymbol{x} \in D_0$.*

**Definition D.2.** *Let $\boldsymbol{x}_0 \in \mathbb{R}^m$ and $\delta > 0$. Let $p : B(\boldsymbol{x}_0, \delta) \to \mathbb{R}$. Define*

$$Ext(p) : \mathbb{R}^m \to \mathbb{R}$$

*to be the unique analytic continuation of $p$ on the entire space $\mathbb{R}^m$ if such a continuation exists, and to be 0 otherwise.*

**Definition D.3.** *Let $D_0 \subset D$ and $p : D \to \mathbb{R}$ be a function. We define $p|_{D_0} : D \to \mathbb{R}$ to be a restriction of $p$ to $D_0$, namely a function that satisfies $p|_{D_0}(\boldsymbol{x}) = p(\boldsymbol{x})$ for every $\boldsymbol{x} \in D_0$.*

**Definition D.4.** *Let $f : \mathbb{R}^m \to \mathbb{R}^n$ be a piece-wise affine function. We say that a point $\boldsymbol{x} \in f(\mathbb{R}^m) \subseteq \mathbb{R}^n$ is generic with respect to $f$ if the pre-image $f^{-1}(\{\boldsymbol{x}\})$ is finite and there exists $\delta > 0$, such that $f : B(\boldsymbol{z}, \delta) \to \mathbb{R}^n$ is affine for every $\boldsymbol{z} \in f^{-1}(\{\boldsymbol{x}\})$.*

**Lemma D.1.** *If $f : \mathbb{R}^m \to \mathbb{R}^n$ is a piece-wise affine function such that $\{\boldsymbol{x} \in \mathbb{R}^n : |f^{-1}(\{\boldsymbol{x}\})| > 1\} \subseteq f(\mathbb{R}^m)$ has measure zero with respect to the Lebesgue measure on $f(\mathbb{R}^m)$, then $\dim(f(\mathbb{R}^m)) = m$ and almost every point in $f(\mathbb{R}^m)$ is generic with respect to $f$.*

## D.2   PROOF OF THEOREM 3.2.(I)

We extend the results from Kivva et al. (2022) to using our MSM family as prior distribution for $\boldsymbol{z}_{1:T}$. The strategy requires finding some open set where the transformations $\mathcal{F}$ and $\mathcal{G}$ from two equally distributed SDSs are invertible, and then use analytic function properties to establish the identifiability result. First, we need to show that the points in the pre-image of a piece-wise factored mapping $\mathcal{F}$ can be computed using the MSM prior.

**Lemma D.2.** *Consider a random variable $\boldsymbol{z}_{1:T}$ which follows a Markov Switching Model distribution. Let us consider $f : \mathbb{R}^m \to \mathbb{R}^m$, a piece-wise affine mapping which generates the random variable $\boldsymbol{x}_{1:T} = \mathcal{F}(\boldsymbol{z}_{1:T})$ as $\boldsymbol{x}_t = f(\boldsymbol{z}_t), 1 \leq t \leq T$. Also, consider $\boldsymbol{x}^{(0)} \in \mathbb{R}^m$ a generic point with respect to $f$. Then, $\boldsymbol{x}_{1:T}^{(0)} = \{\boldsymbol{x}^{(0)}, \ldots, \boldsymbol{x}^{(0)}\} \in \mathbb{R}^{Tm}$ is also a generic point with respect to $\mathcal{F}$ and the number of points in the pre-image $\mathcal{F}^{-1}(\{\boldsymbol{x}_{1:T}^{(0)}\})$ can be computed as*

$$\left| \mathcal{F}^{-1} \left( \left\{ \boldsymbol{x}_{1:T}^{(0)} \right\} \right) \right| = \lim_{\delta \to 0} \int_{\boldsymbol{x}_{1:T} \in \mathbb{R}^{Tm}} Ext \left( p|_{B(\boldsymbol{x}_{1:T}^{(0)}, \delta)} \right) d\boldsymbol{x}_{1:T}$$

*Proof.* If $\boldsymbol{x}^{(0)} \in \mathbb{R}^m$ is a generic point with respect to $f$, $\boldsymbol{x}_{1:T}^{(0)}$ is also a generic point with respect to $\mathcal{F}$ since the pre-image is $\mathcal{F}(\{\boldsymbol{x}_{1:T}^{(0)}\})$ now larger but still finite. In other words, $\mathcal{F}(\{\boldsymbol{x}_{1:T}^{(0)}\})$ is the Cartesian product $\mathcal{Z} \times \mathcal{Z} \times \cdots \times \mathcal{Z}$, where $\mathcal{Z} = \{\boldsymbol{z}_1, \boldsymbol{z}_2, \ldots, \boldsymbol{z}_n\}$ are the points in the pre-image $f(\{\boldsymbol{x}^{(0)}\})$. Considering this, we have well defined affine mappings $\mathcal{G}_{i_1, \ldots, i_T} : B(\{\boldsymbol{z}_{i_1}, \ldots, \boldsymbol{z}_{i_T}\}, \epsilon) \to \mathbb{R}^m$, $i_t \in \{1, \ldots, n\}$ for $1 \leq t \leq T$, such that $\mathcal{G}_{i_1, \ldots, i_T} = \mathcal{F}(\boldsymbol{z}_{1:T}), \forall \boldsymbol{z}_{1:T} \in B(\{\boldsymbol{z}_{i_1}, \ldots, \boldsymbol{z}_{i_T}\}, \epsilon)$. This affine mapping $\mathcal{G}_{i_1, \ldots, i_T}$ is factored as follows:

$$g_{i_t}(\boldsymbol{z}_t) = f(\boldsymbol{z}_t), \quad \forall \boldsymbol{z}_t \in B(\boldsymbol{z}_i, \epsilon)$$

$$\mathcal{G}_{i_1, \ldots, i_T} = \begin{pmatrix} A_{i_1} & \ldots & \boldsymbol{0} \\ \vdots & \ddots & \vdots \\ \boldsymbol{0} & \ldots & A_{i_T} \end{pmatrix} \begin{pmatrix} \boldsymbol{z}_1 \\ \vdots \\ \boldsymbol{z}_T \end{pmatrix} + \begin{pmatrix} b_{i_1} \\ \vdots \\ b_{i_T} \end{pmatrix}$$

Let $\delta_0 > 0$ such that

$$B(\boldsymbol{x}_{1:T}^{(0)}, \delta_0) \subseteq \bigcap_{i_1, \ldots, i_T}^{n} \mathcal{G}_{i_1, \ldots, i_T}(B(\{\boldsymbol{z}_{i_1}, \ldots, \boldsymbol{z}_{i_T}\}, \epsilon))$$

we can compute the likelihood for every $\boldsymbol{x}_{1:T} \in B(\boldsymbol{x}_{1:T}^{(0)}, \delta')$ with $0 < \delta' < \delta_0$ using Prop. C.1 where the MSM is closed under factored affine transformations.

$$p|_{B(\boldsymbol{x}_{1:T}^{(0)}, \delta)} = \sum_{i_1, \ldots, i_T} \sum_{j_1, \ldots, j_T}^{K} p\left(s_{1:T} = \{j_1, \ldots, j_T\}\right) \mathcal{N}\left(\boldsymbol{x}_1; A_{i_1}\boldsymbol{\mu}(j_1) + \mathbf{b}_{i_1}, A_{i_1}\boldsymbol{\Sigma}_1(j_1)A_{i_1}^T\right)$$

$$\prod_{t=2}^{T} \mathcal{N}\left(\boldsymbol{x}_t; A_{i_t}\mathbf{m}\left(A_{i_{t-1}}^{-1}\left(\boldsymbol{x}_{t-1} - \mathbf{b}_{i_{t-1}}\right), j_t\right) + \mathbf{b}_{i_t}, A_{i_t}\boldsymbol{\Sigma}\left(A_{i_t}^{-1}\left(\boldsymbol{x}_{t-1} - \mathbf{b}_{i_{t-1}}\right), j_t\right)A_{i_t}^T\right)$$

Where the previous density is an analytic function which is defined on an open neighbourhood of $\boldsymbol{x}_{1:T}^{(0)}$. Then from the identity theorem of analytic functions the resulting density defines the analytic

extension of $p|_{B(\boldsymbol{x}_{1:T}^{(0)}, \delta)}$ on $\mathbb{R}^m$. Then, we have

$$
\int_{\boldsymbol{x}_{1:T} \in \mathbb{R}^{Tm}} \operatorname{Ext}\left(p|_{B(\boldsymbol{x}_{1:T}^{(0)}, \delta)}\right) d\boldsymbol{x}_{1:T}
$$

$$
= \sum_{i_1, \ldots, i_T}^{s} \sum_{j_1, \ldots, j_T}^{K} p\left(s_{1:T} = \{j_1, \ldots, j_T\}\right) \mathcal{N}\left(\boldsymbol{x}_1; A_{i_1}\boldsymbol{\mu}(j_1) + \mathbf{b}_{i_1}, A_{i_1}\boldsymbol{\Sigma}_1(j_1)A_{i_1}^T\right)
$$

$$
\prod_{t=2}^{T} \mathcal{N}\left(\boldsymbol{x}_t; A_{i_t}\mathbf{m}\left(A_{i_{t-1}}^{-1}\left(\boldsymbol{x}_{t-1} - \mathbf{b}_{i_{t-1}}\right), j_t\right) + \mathbf{b}_{i_t}, A_{i_t}\boldsymbol{\Sigma}\left(A_{i_{t-1}}^{-1}\left(\boldsymbol{x}_{t-1} - \mathbf{b}_{i_{t-1}}\right), j_t\right)A_{i_t}^T\right)
$$

$$
= \sum_{i_1, \ldots, i_T}^{n} \int_{\boldsymbol{x}_{1:T} \in \mathbb{R}^{Tm}} \sum_{j_1, \ldots, j_T}^{K} p\left(s_{1:T} = \{j_1, \ldots, j_T\}\right) \mathcal{N}\left(\boldsymbol{x}_1; A_{i_1}\boldsymbol{\mu}(j_1) + \mathbf{b}_{i_1}, A_{i_1}\boldsymbol{\Sigma}_1(j_1)A_{i_1}^T\right)
$$

$$
\prod_{t=2}^{T} \mathcal{N}\left(\boldsymbol{x}_t; A_{i_t}\mathbf{m}\left(A_{i_{t-1}}^{-1}\left(\boldsymbol{x}_{t-1} - \mathbf{b}_{i_{t-1}}\right), j_t\right) + \mathbf{b}_{i_t},\right.
$$

$$
\left. A_{i_t}\boldsymbol{\Sigma}\left(A_{i_{t-1}}^{-1}\left(\boldsymbol{x}_{t-1} - \mathbf{b}_{i_{t-1}}\right), j_t\right)A_{i_t}^T\right) d\boldsymbol{x}_{1:T}
$$

$$
= \sum_{i_1, \ldots, i_T}^{n} 1 = n^T = \left|\mathcal{F}^{-1}(\{\boldsymbol{x}_{1:T}^{(0)}\})\right|
$$

$\square$

We can deduce the following corollary as in Kivva et al. (2022).

**Corollary D.1.** *Let $\mathcal{F}, \mathcal{G} : \mathbb{R}^{Tm} \to \mathbb{R}^{Tn}$ be factored piece-wise affine mappings, with $\boldsymbol{x}_t := f(\boldsymbol{z}_t)$ and $\boldsymbol{x}_t' := g(\boldsymbol{z}_t')$, for $1 \leq t \leq T$. Assume $f$ and $g$ are weakly-injective (Def. 2.1). Let $\boldsymbol{z}_{1:T}$ and $\boldsymbol{z}_{1:T}'$ be distributed according to the identifiable MSM family. Assume $\mathcal{F}(\boldsymbol{z}_{1:T})$ and $\mathcal{G}(\boldsymbol{z}_{1:T}')$ are equally distributed. Assume that for $\boldsymbol{x}_0 \in \mathbb{R}^n$ and $\delta > 0$, $f$ is invertible on $B(\boldsymbol{x}_0, 2\delta) \cap f(\mathbb{R}^m)$.*

*Then, for $\boldsymbol{x}_{1:T}^{(0)} = \{\boldsymbol{x}_0, \ldots, \boldsymbol{x}_0\} \in \mathbb{R}^{Tn}$ there exists $\boldsymbol{x}_{1:T}^{(1)} \in B(\boldsymbol{x}_{1:T}^{(0)}, \delta)$ and $\delta_1 > 0$ such that both $\mathcal{F}$ and $\mathcal{G}$ are invertible on $B(\boldsymbol{x}_{1:T}^{(1)}, \delta_1) \cap \mathcal{F}(\mathbb{R}^{Tm})$.*

*Proof.* First, we observe that since $\mathcal{F}$ is a factored mapping, if $f$ is invertible on $B(\boldsymbol{x}_0, 2\delta) \cap f(\mathbb{R}^m)$, we can compute the inverse of $\mathcal{F}$ on $B(\boldsymbol{x}_{1:T}^{(0)}, 2\delta) \cap \mathcal{F}(\mathbb{R}^m)$ for $\boldsymbol{x}_{1:T}^{(0)} = \{\boldsymbol{x}_0, \ldots, \boldsymbol{x}_0\} \in \mathbb{R}^{Tn}$ as $\mathcal{F}^{-1}(\boldsymbol{x}_{1:T}) = \{f^{-1}(\boldsymbol{x}_1), \ldots, f^{-1}(\boldsymbol{x}_T)\} \in \mathbb{R}^{Tm}$, for $\boldsymbol{x}_t \in B(\boldsymbol{x}_0, 2\delta) \cap f(\mathbb{R}^m)$, $1 \leq t \leq T$. Then, $\mathcal{F}$ is invertible on $B(\boldsymbol{x}_{1:T}^{(0)}, 2\delta) \cap \mathcal{F}(\mathbb{R}^m)$.

By Lemma D.1, almost every point $\boldsymbol{x} \in B(\boldsymbol{x}^{(0)}, \delta) \cap f(\mathbb{R}^m)$ is generic with respect to $f$ and $g$, as both mappings are weakly injective. As discussed previously, if $\boldsymbol{x}^{(0)} \in f(\mathbb{R}^m)$ is a generic point with respect to $f$, the point $\boldsymbol{x}_{1:T}^{(0)} = \{\boldsymbol{x}^{(0)}, \ldots, \boldsymbol{x}^{(0)}\} \in \mathcal{F}(\mathbb{R}^{Tm})$ is also generic with respect to $\mathcal{F}$, as the finite points in the preimage $f^{-1}(\{\boldsymbol{x}^{(0)}\})$ extend to finite points in the preimage $\mathcal{F}^{-1}(\{\boldsymbol{x}_{1:T}^{(0)}\})$. Then, almost every point $\boldsymbol{x}_{1:T} \in B(\boldsymbol{x}_{1:T}^{(0)}, \delta) \cap \mathcal{F}(\mathbb{R}^{Tm})$ is generic with respect to $\mathcal{F}$ and $\mathcal{G}$.

Consider now $\boldsymbol{x}_{1:T}^{(1)} = \{\boldsymbol{x}^{(1)}, \ldots, \boldsymbol{x}^{(1)}\} \in B(\boldsymbol{x}_{1:T}^{(0)}, \delta)$ such a generic point. From the invertibility of $\mathcal{F}$ on $B(\boldsymbol{x}_{1:T}^{(1)}, \delta)$, we have $|\mathcal{F}^{-1}(\{\boldsymbol{x}_{1:T}^{(1)}\})| = 1$. By Lemma D.2, we have that $|\mathcal{G}^{-1}(\{\boldsymbol{x}_{1:T}^{(1)}\})| = 1$, as $\boldsymbol{x}_{1:T}^{(1)}$ is generic with respect to $\mathcal{F}$ and $\mathcal{G}$. Then, there exists, $\delta > \delta_1 > 0$ such that on $\left(B(\boldsymbol{x}_{1:T}^{(1)}, \delta_1) \cap \mathcal{F}(\mathbb{R}^{Tm})\right) \subset \left(B(\boldsymbol{x}_{1:T}^{(0)}, 2\delta) \cap \mathcal{F}(\mathbb{R}^{Tm})\right)$ the function $\mathcal{G}$ is invertible. $\square$

We need an additional result to prepare the proof for Theorem 3.2.(i).

**Theorem D.1.** *Let $\mathcal{F}, \mathcal{G} : \mathbb{R}^{Tm} \to \mathbb{R}^{Tn}$ be factored piece-wise affine mappings, with $\boldsymbol{x}_t := f(\boldsymbol{z}_t)$ and $\boldsymbol{x}_t' := g(\boldsymbol{z}_t')$, for $1 \leq t \leq T$. Let $\boldsymbol{z}_{1:T}$ and $\boldsymbol{z}_{1:T}'$ be distributed according to the identifiable MSM family. Assume $\mathcal{F}(\boldsymbol{z}_{1:T})$ and $\mathcal{G}(\boldsymbol{z}_{1:T}')$ are equally distributed, and that there exists $\boldsymbol{x}_{1:T}^{(0)} \in \mathbb{R}^{Tn}$ and*

$\delta > 0$ *such that $\mathcal{F}$ and $\mathcal{G}$ are invertible on $B(\boldsymbol{x}_{1:T}^{(0)}, \delta) \cap f(\mathbb{R}^{Tm})$. Then there exists an invertible factored affine transformation $\mathcal{H}$ such that $\mathcal{H}(\boldsymbol{z}_{1:T}) = \boldsymbol{z}'_{1:T}$.*

*Proof.* From the invertibility of $\mathcal{F}$ and $\mathcal{G}$ in $B(\boldsymbol{x}_{1:T}^{(0)}, \delta) \cap f(\mathbb{R}^{Tm})$ we can find a $Tm$-dimensional affine subspace $B(\boldsymbol{x}_{1:T}^{(1)}, \delta_1) \cap L$, where $\delta_1 > 0$, $B(\boldsymbol{x}_{1:T}^{(1)}, \delta_1) \subseteq B(\boldsymbol{x}_{1:T}^{(0)}, \delta)$, and $L \subseteq \mathbb{R}^{Tn}$ such that $\mathcal{H}_{\mathcal{F}}, \mathcal{H}_{\mathcal{G}} : \mathbb{R}^{Tm} \to L$ are a pair of invertible affine functions where $\mathcal{H}_{\mathcal{F}}^{-1}$ and $\mathcal{H}_{\mathcal{G}}^{-1}$ coincide with $\mathcal{F}^{-1}$ and $\mathcal{G}^{-1}$ on $B(\boldsymbol{x}_1, \delta_1) \cap L$ respectively. The fact that $\mathcal{F}$ and $\mathcal{G}$ are factored implies that $\mathcal{H}_{\mathcal{F}}, \mathcal{H}_{\mathcal{G}}$ are also factored. To see this, we observe that the inverse of a block diagonal matrix is the inverse of each block, as an example for $\mathcal{F}$, we first have that $\mathcal{H}_{\mathcal{F}}^{-1}$ must be forcibly factored since it needs to coincide with $\mathcal{F}^{-1}$.

$$
\mathcal{H}_{\mathcal{F}}^{-1}(\boldsymbol{x}_{1:T}) = \begin{pmatrix} \tilde{A}_f & \dots & \mathbf{0} \\ \vdots & \ddots & \vdots \\ \mathbf{0} & \dots & \tilde{A}_f \end{pmatrix} \begin{pmatrix} \boldsymbol{x}_1 \\ \vdots \\ \boldsymbol{x}_T \end{pmatrix} + \begin{pmatrix} \tilde{b}_f \\ \vdots \\ \tilde{b}_f \end{pmatrix} = \begin{pmatrix} f^{-1}(\boldsymbol{x}_1) \\ \vdots \\ f^{-1}(\boldsymbol{x}_T) \end{pmatrix}
$$

then we can take the inverse to obtain the factored $\mathcal{H}_{\mathcal{F}}$.

$$
\begin{aligned}
\mathcal{H}_{\mathcal{F}} &= \begin{pmatrix} \tilde{A}_f^{-1} & \dots & \mathbf{0} \\ \vdots & \ddots & \vdots \\ \mathbf{0} & \dots & \tilde{A}_f^{-1} \end{pmatrix} \begin{pmatrix} \boldsymbol{z}_1 \\ \vdots \\ \boldsymbol{z}_T \end{pmatrix} - \begin{pmatrix} \tilde{A}_f^{-1} & \dots & \mathbf{0} \\ \vdots & \ddots & \vdots \\ \mathbf{0} & \dots & \tilde{A}_f^{-1} \end{pmatrix} \begin{pmatrix} \tilde{b}_f \\ \vdots \\ \tilde{b}_f \end{pmatrix} \\
&= \begin{pmatrix} A_f & \dots & \mathbf{0} \\ \vdots & \ddots & \vdots \\ \mathbf{0} & \dots & A_f \end{pmatrix} \begin{pmatrix} \boldsymbol{z}_1 \\ \vdots \\ \boldsymbol{z}_T \end{pmatrix} + \begin{pmatrix} b_f \\ \vdots \\ b_f \end{pmatrix}, \quad \text{where } A_f = \tilde{A}_f^{-1}, \text{ and } b_f = -\tilde{A}_f^{-1}\tilde{b}_f
\end{aligned}
$$

Since $\mathcal{F}(\boldsymbol{z}_{1:T})$ and $\mathcal{G}(\boldsymbol{z}'_{1:T})$ are equally distributed, we have that $\mathcal{H}_{\mathcal{F}}(\boldsymbol{z}_{1:T})$ and $\mathcal{H}_{\mathcal{G}}(\boldsymbol{z}'_{1:T})$ are equally distributed on $B(\boldsymbol{x}_{1:T}^{(1)}, \delta_1) \cap L$. From Prop C.1, we know that $\mathcal{H}_{\mathcal{F}}(\boldsymbol{z}_{1:T})$ and $\mathcal{H}_{\mathcal{G}}(\boldsymbol{z}'_{1:T})$ are distributed according to the identifiable MSM family, which implies $\mathcal{H}_{\mathcal{F}}(\boldsymbol{z}_{1:T}) = \mathcal{H}_{\mathcal{G}}(\boldsymbol{z}'_{1:T})$, and also $\mathcal{H}_{\mathcal{G}}^{-1}(\mathcal{H}_{\mathcal{F}}(\boldsymbol{z}_{1:T})) = \boldsymbol{z}'_{1:T}$, where $\mathcal{H} = \mathcal{H}_{\mathcal{G}}^{-1} \circ \mathcal{H}_{\mathcal{F}}$ is an affine transformation. $\square$

From the previous result and Theorem 3.1, there exists a permutation $\sigma(\cdot)$, such that $\mathbf{m}_{fg}(\boldsymbol{z}', k) = \mathbf{m}'(\boldsymbol{z}', \sigma(k))$ for $1 \le k \le K$.

$$
\begin{aligned}
\mathbf{m}'(\boldsymbol{z}', \sigma(k)) = \mathbf{m}_{fg}(\boldsymbol{z}', k) &= A_g^{-1}\mathbf{m}_f\left(A_g \boldsymbol{z}' + \mathbf{b}_g, k\right) - A_g^{-1}\mathbf{b}_g \\
&= A_g^{-1}A_f \mathbf{m}\left(A_f^{-1}A_g \boldsymbol{z}' + A_f^{-1}\left(\mathbf{b}_g - \mathbf{b}_f\right), k\right) + A_g^{-1}\left(\mathbf{b}_f - \mathbf{b}_g\right) \\
&= A\mathbf{m}\left(A^{-1}(\boldsymbol{z}' - \mathbf{b}), k\right) + \mathbf{b},
\end{aligned}
$$

where $A = A_g^{-1}A_f$ and $\mathbf{b} = A_g^{-1}\left(\mathbf{b}_f - \mathbf{b}_g\right)$. Similar implications apply for $\Sigma(\boldsymbol{z}, a), a \in \mathcal{A}$, which we indicate in Rem. 3.2.

Now we have all the elements to prove Theorem 3.2.(i).

*Proof.* We assume there exists another model that generates the same distribution from Eq.(10), whith a prior $p' \in \mathcal{M}_{NL}^T$ under assumptions (a1-a2), and non-linear emmision $\mathcal{F}'$, composed by $f$ which is weakly injective and piece-wise linear: i.e. $(\mathcal{F}\#p)(\boldsymbol{x}_{1:T}) = (\mathcal{F}'\#p')(\boldsymbol{x}_{1:T})$.

From weakly-injectiveness, at least for some $\boldsymbol{x}_0 \in \mathbb{R}^n$ and $\delta > 0$, $f$ is invertible on $B(\boldsymbol{x}_0, 2\delta) \cap f(\mathbb{R}^m)$. This satisfies the preconditions from Corollary D.1, which implies there exists $\boldsymbol{x}_{1:T}^{(1)} \in B(\boldsymbol{x}_{1:T}^{(0)}, \delta)$ and $\delta_1 > 0$ such that both $\mathcal{F}$ and $\mathcal{F}'$ are invertible on $B(\boldsymbol{x}_{1:T}^{(1)}, \delta_1) \cap \mathcal{F}(\mathbb{R}^{Tm})$. Thus, by Theorem D.1, there exists an affine transformation $\mathcal{H}$ such that $\mathcal{H}(\boldsymbol{z}_{1:T}) = \boldsymbol{z}'_{1:T}$, which means that $p \in \mathcal{M}_{NL}^T$ is identifiable up to affine transformations. $\square$

### D.3 PROOF OF THEOREM 3.2.(II)

So far we have proved the identifiability of the transition function on the latent MSM distribution up to affine transformations. By further assuming injectivity of the piece-wise mapping $\mathcal{F}$, we can

prove identifiability of $\mathcal{F}$ up to affine transformations by re-using results from Kivva et al. (2022). We begin by stating the following known result.

**Lemma D.3.** *Let $Z \sim \sum_{j=1}^{J} \lambda_j \mathcal{N}(\boldsymbol{\mu}_j, \Sigma_j)$ where $Z$ is a GMM (in reduced form). Assume that $f : \mathbb{R}^m \to \mathbb{R}^m$ is a continuous piecewise affine function such that $f(Z) \sim Z$. Then $f$ is affine.*

We can state the identification of $\mathcal{F}$.

**Theorem D.2.** *Let $\mathcal{F}, \mathcal{G} : \mathbb{R}^{mT} \to \mathbb{R}^{nT}$ be continuous invertible factored piecewise affine functions. Let $\boldsymbol{z}_{1:T}, \boldsymbol{z}'_{1:T}$ be random variables distributed according to MSMs. Suppose that $\mathcal{F}(\boldsymbol{z}_{1:T})$ and $\mathcal{G}(\boldsymbol{z}'_{1:T})$ are equally distributed.*

*Then there exists a factored affine transformation $\mathcal{H} : \mathbb{R}^{mT} \to \mathbb{R}^{mT}$ such that $\mathcal{H}(\boldsymbol{z}_{1:T}) = \boldsymbol{z}'_{1:T}$ and $\mathcal{G} = \mathcal{F} \circ \mathcal{H}^{-1}$.*

*Proof.* From Theorem D.1, there exists an invertible affine transformation $\mathcal{H}_1 : \mathbb{R}^{mT} \to \mathbb{R}^{mT}$ such that $\mathcal{H}_1(\boldsymbol{z}_{1:T}) = \boldsymbol{z}'_{1:T}$. Then, $\mathcal{F}(\boldsymbol{z}_{1:T}) \sim \mathcal{G}(\mathcal{H}_1(\boldsymbol{z}_{1:T}))$. From the invertibility of $\mathcal{G}$, we have $\boldsymbol{z}_{1:T} \sim (\mathcal{H}_1^{-1} \circ \mathcal{G}^{-1} \circ \mathcal{F})(\boldsymbol{z}_{1:T})$. We note that $\mathcal{H}_1, \mathcal{G}, \mathcal{F}$ are factored mappings, and structured as follows

$$\left(\mathcal{H}_1^{-1} \circ \mathcal{G}^{-1} \circ \mathcal{F}\right)(\boldsymbol{z}_{1:T}) = \begin{pmatrix} \left(h_1^{-1} \circ g^{-1} \circ f\right)(\boldsymbol{z}_1) \\ \vdots \\ \left(h_1^{-1} \circ g^{-1} \circ f\right)(\boldsymbol{z}_T) \end{pmatrix} \sim \begin{pmatrix} (\boldsymbol{z}_1) \\ \vdots \\ (\boldsymbol{z}_T) \end{pmatrix},$$

where the inverse of $\mathcal{H}_1$ is also factored, as observed from previous results. Since the transformation is equal for $1 \leq t \leq T$, we can proceed for $t = 1$ considering $\boldsymbol{z}_1$ is distributed as a GMM (in reduced form), as it corresponds to the initial distribution of the MSM. Then, by Lemma D.3, there exists an affine mapping $h_2 : \mathbb{R}^m \to \mathbb{R}^m$ such that $h_1^{-1} \circ g^{-1} \circ f = h_2$. Then,

$$\mathcal{F} = \begin{pmatrix} f \\ \vdots \\ f \end{pmatrix} = \begin{pmatrix} g \circ h_1 \circ h_2 \\ \vdots \\ g \circ h_1 \circ h_2 \end{pmatrix} = (\mathcal{G} \circ \mathcal{H}),$$

where $h = h_1 \circ h_2$. Considering the invertibility of $\mathcal{G}$ and the fact that $\mathcal{F}(\boldsymbol{z}_{1:T})$ and $\mathcal{G}(\boldsymbol{z}'_{1:T})$ are equally distributed, we also have $\mathcal{H}(\boldsymbol{z}_{1:T}) = \boldsymbol{z}'_{1:T}$. □

We use the previous result to prove Theorem 3.2.(ii).

*Proof.* We assume there exists another model that generates the same distribution from Eq.(10), whith a prior $p' \in \mathcal{M}_{NL}^T$ under assumptions (a1-a2), and non-linear emmision $\mathcal{F}'$, composed by $f$ which is continuous, injective and piece-wise linear: i.e. $(\mathcal{F}\#p)(\boldsymbol{x}_{1:T}) = (\mathcal{F}'\#p')(\boldsymbol{x}_{1:T})$.

These are the preconditions to satisfy Theorem D.2, which implies there exists an affine transformation $\mathcal{H}$ such that $\mathcal{H}(\boldsymbol{z}_{1:T}) = \boldsymbol{z}'_{1:T}$ and $\mathcal{F} = \mathcal{F}' \circ \mathcal{H}$. In other words, the prior $p \in \mathcal{M}_{NL}^T$, and $f$ which composes $\mathcal{F}$ are identifiable up to affine transformations. □

# E ESTIMATION DETAILS

We provide additional details from the descriptions in the main text.

## E.1 EXPECTATION MAXIMISATION ON MSMS

For convenience, the expressions below are computed from samples $\{\boldsymbol{z}_{1:T}^b\}_{b=1}^B$ for a batch of size $B$. Recall we formulate our method in terms of the expectation maximisation (EM) algorithm. Given some arrangement of the parameter values $(\theta')$, the E-step computes the posterior distribution of the latent variables $p_{\theta'}(\mathbf{s}_{1:T}|\boldsymbol{z}_{1:T})$. This can then be used to compute the expected log-likelihood of the complete data (latent variables and observations),

$$\mathcal{L}(\boldsymbol{\theta}, \boldsymbol{\theta}') := \frac{1}{B} \sum_{b=1}^{B} \mathbb{E}_{p_{\boldsymbol{\theta}'}(\mathbf{s}_{1:T}^b|\boldsymbol{z}_{1:T}^b)} \left[ \log p_{\boldsymbol{\theta}}(\boldsymbol{z}_{1:T}^b, \mathbf{s}_{1:T}^b) \right]. \tag{25}$$

Given a first-order stationary Markov chain, we denote the posterior probability $p_{\boldsymbol{\theta}}(s_t^b = k | \boldsymbol{z}_{1:T}^b)$ as $\gamma_{t,k}^b$, and the joint posterior of two consecutive states $p_{\boldsymbol{\theta}}(s_t^b = k, s_{t-1}^b = l | \boldsymbol{z}_{1:T}^b)$ as $\xi_{t,k,l}^b$. For this case, the result is equivalent to the HMM case and can be found in the literature, e.g. Bishop (2006). We can then compute a more explicit form of Eq. (25),

$$\mathcal{L}(\boldsymbol{\theta}, \boldsymbol{\theta}') = \frac{1}{B} \sum_{b=1}^{B} \sum_{k=1}^{K} \gamma_{1,k}^b \log \pi_k + \frac{1}{B} \sum_{b=1}^{B} \sum_{t=2}^{T} \sum_{k=1}^{K} \sum_{l=1}^{K} \xi_{t,k,l}^b \log Q_{lk} +$$

$$\frac{1}{B} \sum_{b=1}^{B} \sum_{k=1}^{K} \gamma_{1,k}^b \log p_{\boldsymbol{\theta}}(\boldsymbol{z}_1^b |, s_1^b = k) + \frac{1}{B} \sum_{b=1}^{B} \sum_{t=2}^{T} \sum_{k=1}^{K} \gamma_{t,k}^b \log p_{\boldsymbol{\theta}}(\boldsymbol{z}_t^b | \boldsymbol{z}_{t-1}^b, s_t^b = k), \quad (26)$$

where $\pi$ and $Q$ denote the initial and transition distribution of the Markov chain. In the M-step, the previous expression is maximised to calculate the update rules for the parameters, i.e. $\boldsymbol{\theta}^{\text{new}} = \arg\max_{\boldsymbol{\theta}} \mathcal{L}(\boldsymbol{\theta}, \boldsymbol{\theta}')$. The updates for $\pi$ and $Q$ are also obtained using standard results for HMM inference (again see Bishop (2006)). Assuming Gaussian initial and transition densities, we can also use standard literature results for updating the initial mean and covariance. For the transition densities, we consider a family with fixed covariance matrices, and only the means $\boldsymbol{m}_{\boldsymbol{\theta}}(\cdot, k)$ are dependent on the previous observation. In this case, the standard results can also be used to update the covariances of the transition distributions. We drop the subscript $\boldsymbol{\theta}$ for convenience.

The updates for the mean parameters are dependent on the functions we choose. For multivariate polynomials of degree $P$, we can recover an exact M-step by transforming the mapping into a matrix-vector operation:

$$\boldsymbol{m}(\boldsymbol{z}_{t-1}, k) = \sum_{c=1}^{C} A_{k,c} \hat{\boldsymbol{z}}_{c,t-1}, \quad \hat{\boldsymbol{z}}_{t-1}^T = \begin{pmatrix} 1 & z_{t-1,1} & \cdots & z_{t-1,d} & z_{t-1,1}^2 & z_{t-1,1} z_{t-1,2} & \cdots \end{pmatrix},$$

$$(27)$$

where $\hat{z}_{t-1} \in \mathbb{R}^C$ denotes the polynomial features of $\boldsymbol{z}_{t-1}$ up to degree $P$. The total number of features is $C = \binom{P+d}{d}$ and the exact update for $A_k$ is

$$A_k \leftarrow \left( \sum_{b=1}^{B} \sum_{t=2}^{T} \gamma_{t,k}^b \boldsymbol{z}_t^b (\hat{\boldsymbol{z}}_{t-1}^b)^T \right) \left( \sum_{b=1}^{B} \sum_{t=2}^{T} \gamma_{t,k}^b \hat{\boldsymbol{z}}_{t-1}^b (\hat{\boldsymbol{z}}_{t-1}^b)^T \right)^{-1}. \quad (28)$$

For exact updates such as the one above, we require $B$ to be sufficiently large to ensure consistent updates during training. In the main text, we already discussed the case where the transition means are parametrised by neural networks.

## E.2 Variational Inference for SDSs

We provide more details on the ELBO objective for SDSs.

$$\log p_{\boldsymbol{\theta}}(\boldsymbol{x}_{1:T}) = \log \int \sum_{\mathbf{s}_{1:T}} p_{\boldsymbol{\theta}}(\boldsymbol{x}_{1:T}, \boldsymbol{z}_{1:T}, \mathbf{s}_{1:T}) d\boldsymbol{z}_{1:T} \quad (29)$$

$$\geq \mathbb{E}_{q_{\phi,\theta}(\boldsymbol{z}_{1:T}, \mathbf{s}_{1:T} | \boldsymbol{x}_{1:T})} \left[ \log \frac{p_{\boldsymbol{\theta}}(\boldsymbol{x}_{1:T}, \boldsymbol{z}_{1:T} | \mathbf{s}_{1:T}) p_{\boldsymbol{\theta}}(\mathbf{s}_{1:T})}{q_{\phi}(\boldsymbol{z}_{1:T}, \mathbf{s}_{1:T} | \boldsymbol{x}_{1:T})} \right] \quad (30)$$

$$\geq \mathbb{E}_{q_{\phi}(\boldsymbol{z}_{1:T} | \boldsymbol{x}_{1:T})} \left[ \log \frac{p_{\boldsymbol{\theta}}(\boldsymbol{x}_{1:T} | \boldsymbol{z}_{1:T})}{q_{\phi}(\boldsymbol{z}_{1:T} | \boldsymbol{x}_{1:T})} + \mathbb{E}_{p_{\boldsymbol{\theta}}(\mathbf{s}_{1:T} | \boldsymbol{z}_{1:T})} \left[ \log \frac{p_{\boldsymbol{\theta}}(\boldsymbol{z}_{1:T} | \mathbf{s}_{1:T}) p_{\boldsymbol{\theta}}(\mathbf{s}_{1:T})}{p_{\boldsymbol{\theta}}(\mathbf{s}_{1:T} | \boldsymbol{z}_{1:T})} \right] \right]$$

$$(31)$$

$$\geq \mathbb{E}_{q_{\phi}(\boldsymbol{z}_{1:T} | \boldsymbol{x}_{1:T})} \left[ \log \frac{p_{\boldsymbol{\theta}}(\boldsymbol{x}_{1:T} | \boldsymbol{z}_{1:T})}{q_{\phi}(\boldsymbol{z}_{1:T} | \boldsymbol{x}_{1:T})} + \log p_{\boldsymbol{\theta}}(\boldsymbol{z}_{1:T}) \right] \quad (32)$$

$$\approx \log p_{\boldsymbol{\theta}}(\boldsymbol{x}_{1:T} | \boldsymbol{z}_{1:T}) + \log p_{\boldsymbol{\theta}}(\boldsymbol{z}_{1:T}) - \log q_{\phi}(\boldsymbol{z}_{1:T} | \boldsymbol{x}_{1:T}), \quad \boldsymbol{z}_{1:T} \sim q_{\phi} \quad (33)$$

where as as mentioned, we compute the ELBO objective using Monte Carlo integration with samples $\boldsymbol{z}_{1:T}$ from $q_{\phi}$, and $p_{\boldsymbol{\theta}}(\boldsymbol{z}_{1:T})$ is computed using Eq. (16). Alternatively, Dong et al. (2020) proposes computing the gradients of the latent MSM using the following rule.

$$\nabla \log p_{\boldsymbol{\theta}}(\boldsymbol{x}_{1:T}, \boldsymbol{z}_{1:T}) = \mathbb{E}_{p_{\boldsymbol{\theta}}(\mathbf{s}_{1:T} | \boldsymbol{z}_{1:T})} [\nabla \log p_{\boldsymbol{\theta}}(\boldsymbol{x}_{1:T}, \boldsymbol{z}_{1:T}, \mathbf{s}_{1:T}] \quad (34)$$

where the objective is similar to Eq.(13). Below we reflect on the main aspects of each method.

- Dong et al. (2020) computes the parameters of the latent MSM using a loss term similar to Eq. (13). Although we need to compute the exact posteriors explicitly, we only take the gradient with respect to $\log p_{\boldsymbol{\theta}}(\boldsymbol{z}_t|\boldsymbol{z}_{t-1}, s_t = k)$ which is relatively efficient. Unfortunately, the approach is prone to state collapse and additional loss terms with annealing schedules need to be implemented.

- Ansari et al. (2021) does not require computing exact posteriors as the parameters of the latent MSM are optimized using the forward algorithm. The main disadvantage is that we require back-propagation to flow through the forward computations, which is more inefficient. Despite this, the objective used is less prone to state collapse and optimisation becomes simpler.

Although both approaches show good performance empirically, we observed that training becomes a difficult task and requires careful hyper-parameter tuning and multiple initialisations. Note that the methods are not (a priori) theoretically consistent with the previous identifiability results. Since exact inference is not tractable in SDSs, one cannot design a consistent estimator such as MLE. Future developments should focus on combining the presented methods with tighter variational bounds (Maddison et al., 2017) to design consistent estimators for such generative models.

## F EXPERIMENT DETAILS

### F.1 METRICS

**Markov Switching Models** Consider $K$ components, where as described the evaluation is performed by computing the averaged sum of the distances between the estimated function components. Since we have identifiability of the function forms up to permutations, we need to compute distances with all the permutation configurations to resolve this indeterminacy. Therefore, we can quantify the estimation error as follows

$$\text{err} := \min_{\mathbf{k}=\text{perm}(\{1,\ldots,K\})} \frac{1}{K} \sum_{i=1}^{K} d(\boldsymbol{m}(\cdot,i), \hat{\boldsymbol{m}}(\cdot,k_i)), \tag{35}$$

where $d(\cdot, \cdot)$ denotes the L2 distance between functions. We compute an approximate L2 distance by evaluating the functions on points sampled from a random region of $\mathbb{R}^m$ and averaging the Euclidean distance, more specifically we sample $10^5$ in the $[-1,1]^d$ interval for each evaluation.

$$d(f,g) := \int_{\boldsymbol{x} \in [-1,1]^d} \sqrt{||f(\boldsymbol{x}) - g(\boldsymbol{x})||^2} m\boldsymbol{x} \tag{36}$$

$$\approx \frac{1}{10^5} \sum_{i=1}^{10^5} \sqrt{||f(\boldsymbol{x}^{(i)}) - g(\boldsymbol{x}^{(i)})||^2}, \quad \boldsymbol{x}^{(i)} \sim Uniform([-1,1]^m) \tag{37}$$

Note that resolving the permutation indeterminacy has a cost of $\mathcal{O}(K!)$, which for $K > 5$ already poses some problems in both monitoring performance during training and testing. To alleviate this computational cost, we take a greedy approach, where for each estimated function component we pair it with the ground truth function with the lowest L2 distance. Note that this can return a suboptimal result when the functions are not estimated accurately, but the computational cost is reduced to $\mathcal{O}(K^2)$.

**Switching Dynamical Systems** To compute the $L_2$ distance for the transitions means in SDSs, we first need to resolve the linear transformation in Eq.(11). Thus, we compute the following

$$\arg \min_{h} \left\{ d\Big(f, (f' \circ h)\Big) \right\} \tag{38}$$

where $f, f'$ compose the groundtruth $\mathcal{F}$ and estimated $\mathcal{F}'$ non-linear emissions respectively, and $h$ denotes the affine transformation. We compute the above $L_2$ norm using 1000 generated observations

from a held out test dataset. Finally, we compute the error as in Eq. (35), and with the following $L_2$ norm.

$$d\bigg(\boldsymbol{m}(\cdot, i), A\hat{\boldsymbol{m}}\big(A^{-1}(\boldsymbol{z} - \mathbf{b}), \sigma(i)\big) + \mathbf{b}\bigg) \tag{39}$$

which is taken from Eq.(11). Similarly, we compute the norm using samples from the ground truth latent variables, generated from the held out test dataset. To resolve the permutation $\sigma(\cdot)$, we first compute the $F_1$ score on the segmentation task as indicated, by counting the total true positives, false negatives and false positives. Then, $\sigma(\cdot)$ is determined from the permutation with highest $F_1$ score.

### F.2  Averaged Jacobian and causal structure computation

Regarding regime-dependent causal discovery, our approach can be considered as a functional causal model-based method (see Glymour et al. (2019) for the complete taxonomy). In such methods, the causal structure is usually estimated by inspecting the parameters that encode the dependencies between data, rather than performing independence tests (Tank et al., 2021). In the linear case, we can threshold the transition matrix to obtain an estimate of the causal structure (Pamfil et al., 2020). The non-linear case is a bit more complex since the transition functions are not separable among variables, and the Jacobian can differ considerably for different input values. With the help of locally connected networks, Zheng et al. (2018) aim to encode the variable dependencies in the first layer, and perform similar thresholding as in the linear case. To encourage that the causal structure is captured in the first layer and prevent it from happening in the next ones, the weights in the first layer are regularised with L1 loss to encourage sparsity, and all the weights in the network are regularised with L2 loss.

In our experiments, we observe this approach requires enormous finetuning with the potential to sacrifice the flexibility of the network. Instead, we estimate the causal structure by thresholding the averaged absolute-valued Jacobian with respect to a set of samples. We denote the Jacobian of $\hat{\boldsymbol{m}}(\boldsymbol{z}, k)$ as $\boldsymbol{J}_{\hat{\boldsymbol{m}}(\cdot, k)}(\boldsymbol{z})$. To ensure that the Jacobian captures the effects of the regime of interest, we use samples from the data set and classify them with the posterior distribution. In other words, we will create $K$ sets of variables, where each set $\mathcal{Z}_k$ with size $N_K = |\mathcal{Z}_k|$ contains variables that have been selected using the posterior, i.e. $\boldsymbol{z}^{(i)} \in \mathcal{Z}_k$ if $k = \arg\max p_{\boldsymbol{\theta}}(\mathbf{s}^{(i)}|\boldsymbol{z}_{1:T})$, where we use the index $i$ to denote that $\boldsymbol{z}^{(i)}$ is associated with $\mathbf{s}^{(i)}$. Then, for a given regime $k$, the matrix that encodes the causal structure $\hat{G}_k$ is expressed as

$$\hat{G}_k := \mathbf{1}\left(\frac{1}{N_k}\sum_{i=1}^{N_k}\left|\boldsymbol{J}_{\hat{\boldsymbol{m}}(\cdot, k)}\left(\boldsymbol{z}^{(i)}\right)\right| > \tau\right), \quad \boldsymbol{z}^{(i)} \in \mathcal{Z}_k, \tag{40}$$

where $\mathbf{1}(\cdot)$ is an indicator function which equals to 1 if the argument is true and 0 otherwise. We $\tau = 0.05$ in our experiments. Finally, we evaluate the estimated $K$ regime-dependent causal graphs can be evaluated in terms of the average F1-score over components.

### F.3  Training specifications

All the experiments are implemented in Pytorch (Paszke et al., 2019) and carried out on NVIDIA RTX 2080Ti GPUs, except for the experiments with videos (synthetic and salsa), where we used NVIDIA RTX A6000 GPUs.

**Markov Switching Models**  When training polynomials (including the linear case), we use the exact batched M-step updates with batch size 500 and train for a maximum of 100 epochs, and stop when the likelihood plateaus. When considering updates in the form of Eq. (13), e.g. neural networks, we use ADAM optimiser (Kingma & Ba, 2015) with an initial learning rate $7 \cdot 10^{-3}$ and decrease it by a factor of $0.5$ on likelihood plateau up to 2 times. We vary the batch size and maximum training time depending on the number of states and dimensions. For instance, for $K = 3$ and $d = 3$, we use a batch size of 256 and train for a maximum of 25 epochs. For other configurations, we decrease the batch size and increase the maximum training time to meet GPU memory requirements. Similar to related approaches (Hälvä & Hyvarinen, 2020), we use random restarts to achieve better parameter estimates.

**Switching Dynamical Systems** Since training SDSs requires careful hyperparameter tuning for each setting, we provide details for each setting separately.

- For the synthetic experiments, we use batch size $64$, and we train for $100$ epochs. We use ADAM optimiser (Kingma & Ba, 2015) with an initial learning rate $5 \cdot 10^{-4}$, and decrease it by a factor of $0.5$ every $30$ epochs. To avoid state collapse, we perform an initial warm-up phase for 5 epochs, where we train with fixed discrete state parameters $\pi$ and $Q$, which we fix to uniform distributions. We run multiple seeds and select the best model on the ELBO objective. Regarding the network architecture, we estimate the transition means using two-layer networks with cosine activations and 16 hidden dimensions, and the non-linear emission using two-layer networks with Leaky ReLU activations with 64 hidden dimensions. For the inference network, the bi-directional RNN has 2 hidden layers and 64 hidden dimensions, and the forward RNN has an additional 2 layers with 64 hidden dimensions. We use LSTMs for the RNN updates.

- For the synthetic videos, we vary some of the above configurations. We use batch size $64$ and train for $200$ epochs with the same optimiser and learning rate, but we now decrease it by a factor of $0.8$ every $80$ epochs. Instead of running an initial warm-up phase, we devise a three-stage training. First, we pre-train the encoder (emission) and decoder networks, where for 10 epochs the objective ignores the terms from the MSM prior. The second phase is inspired by Ansari et al. (2021), where we use softmax with temperature on the logits of $\pi$ and $Q$. To illustrate, we use softmax with temperature as follows

$$Q_{k,:} = p(s_t | s_{t-1} = k) = Softmax(\mathbf{o}_k / \tau), \quad t \in \{2, \dots, T\} \tag{41}$$

where $\mathbf{o}_k$ are the logits of $p(s_t | s_{t-1} = k)$ and $\tau$ is the temperature. We start with $\tau = 10$, and decay it exponentially every 50 iterations by a factor of $0.99$ after the pre-training stage. The third stage begins when $\tau = 1$, where we train the model as usual. Again, we run multiple seeds and select the best model on the ELBO. The network architecture is similar, except that we use additional CNNs for inference and generation. For inference, we use a 5-layer CNN with 64 channels, kernel size 3, and padding 1, Leaky ReLU activations, and we alternate between using stride 2 and 1. We then run a 2-layer MLP with Leaky ReLU activations and 64 hidden dimensions and forward the embedding to the same RNN inference network we described before. For generation, we use a similar network, starting with a 2-layer MLP with Leaky ReLU activations and 64 hidden dimensions, and use transposed convolutions instead of convolutions (with the same configuration as before).

- For the salsa dancing videos, we use batch size 8 and train for a maximum of $400$ epochs. We use the same optimiser and an initial learning rate of $10^{-4}$ and stop on ELBO plateau. We use a similar three-stage training as before, where we pre-train the encoder-decoder networks for 10 epochs. For the second stage, we start with $\tau = 100$ and decay it by a factor of $0.975$ every 50 iterations after the pre-training phase. As always, we run multiple seeds and select the best model on ELBO. For the network architectures, we use the same as in the previous synthetic video experiment, but we use 7-layer CNNs, we increase all the network sizes to 128, and we use a latent MSM of 256 dimensions with $K = 3$ components. The transitions of the continous latent variables are parametrised with 2-layer MLPs with SofPlus activations and 256 hidden dimensions.

### F.4 SYNTHETIC EXPERIMENTS

For data generation, we sample $N = 10000$ sequences of length $T = 200$ in terms of a stationary first-order Markov chain with $K$ states. The transition matrix $Q$ is set to maintain the same state with probability $90\%$ and switch to the next state with probability $10\%$, and the initial distribution $\pi$ is the stationary distribution of $Q$. The initial distributions are Gaussian components with means sampled from $\mathcal{N}(0, 0.7^2\mathbf{I})$ and the covariance matrix is $0.1^2\mathbf{I}$. The covariance matrices of the transition distributions are fixed to $0.05^2\mathbf{I}$, and the mean transitions $m(\mathbf{z}, k), k = 1, \dots, K$ are parametrised using polynomials of degree $P = 3$ with random weights, random networks with cosine activations, or random networks with softplus activations. For the locally connected networks (Zheng et al., 2018), we use cosine activation networks, and the sparsity is set to allow 3 interactions per element on average. All neural networks consist of two-layer MLPs with 16 hidden units.

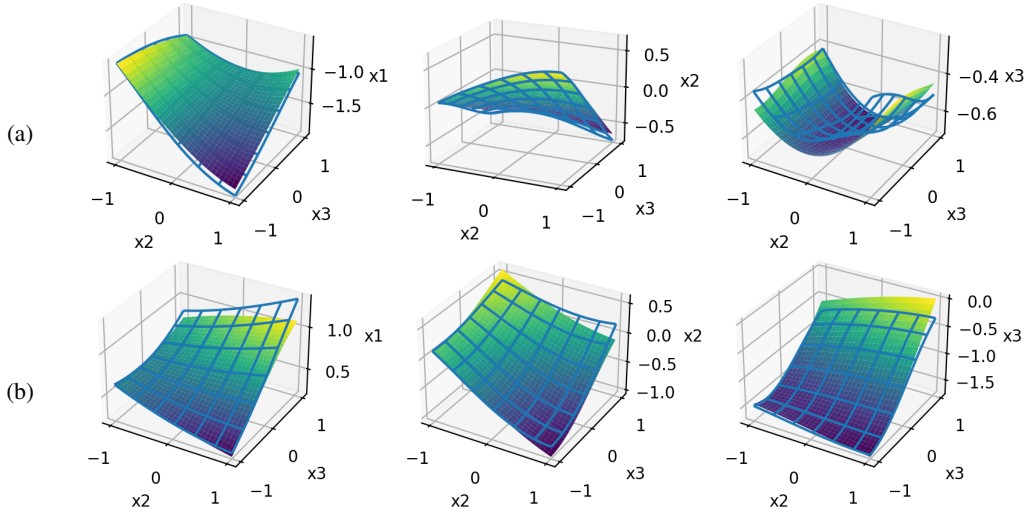

Figure 8: Function responses using 3 dimensions where we vary $x_2$ and $x_3$. Column $i$ shows the response with respect to the $i$-th dimension. The blue grid shows the ground truth function for (a) 3 states showing component 1, and (b) 10 states showing component 6.

When experimenting with SDSs, we use $K = 3$ and a 2D latent MSM. We then further generate observations using two-layer Leaky ReLU netwoks with 8 hidden units. For synthetic videos, we render a ball on $32 \times 32$ coloured images from the 2D coordinates of the latent variables. When rendering images, the MSM trajectories are scaled to ensure the ball is always contained in the image canvas.

In Figure 8, we show visualisations of some function responses for the experiment considering increasing variables and states (figs. 3b and 3c). Recall that, for $K = 3$ states and $d = 3$ dimensions, we achieve $9 \cdot 10^{-3}$ L2 distance error and the responses in Figure 8a show low discrepancies with respect to the ground truth. Similar observations can be made with $K = 10$ states in Figure 8b, where the $L_2$ distance error is $10^{-2}$.

### F.5 ENSO-AIR EXPERIMENT

The data consists of monthly observations of El Niño Southern Oscillation (ENSO) and All India Rainfall (AIR), starting from 1871 to 2016. Following the setting in Saggioro et al. (2020), we more specifically use the indicators *Niño 3.4 SST Index*[4] and *All-India Rainfall precipitation*[5] respectively. In total, we have $N = 1$ samples with $T = 1752$ time steps and consider $K = 2$ components.

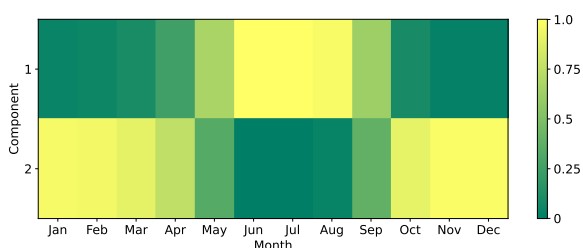

Figure 9: Posterior distribution grouped by month.

In the main text, we claim that our approach captures regimes based on seasonality. To visualise this, We group the posterior distribution by month (fig. 9), where similar groupings arise from both models, and observe that one component is assigned to Summer months (from May to September), and the other is assigned to Winter months (from October to April). To better illustrate the seasonal dependence present in this data. We show the function responses assuming linear and non-linear (softplus networks) transitions in Figures 10a and 10b respectively. In

---

[4]Extracted from `https://psl.noaa.gov/gcos_wgsp/Timeseries/Nino34/`.
[5]Extracted from `https://climexp.knmi.nl/getindices.cgi?STATION=All-India_Rainfall&TYPE=p&WMO=IITMData/ALLIN`.

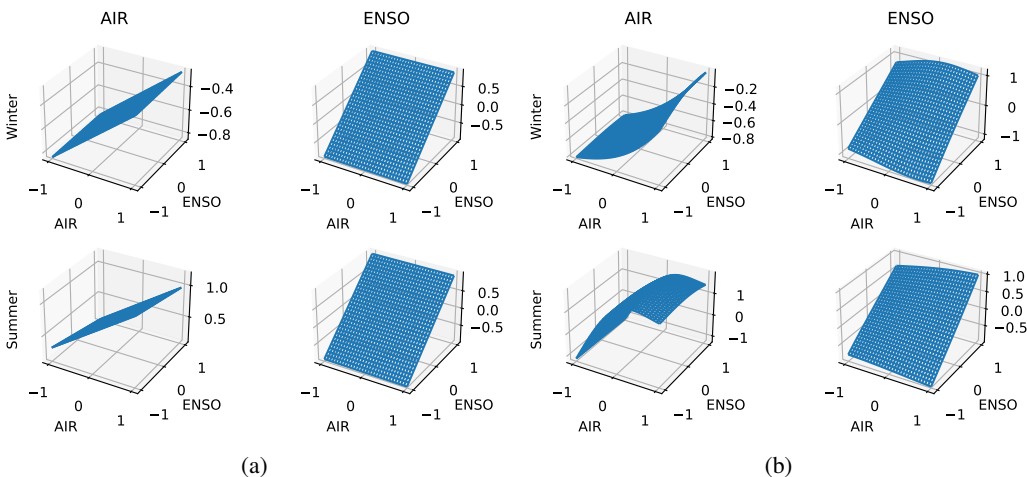

(a)                                   (b)

Figure 10: Function responses of the ENSO-AIR experiment assuming (a) linear and (b) non-linear softplus networks. Each row shows the function responses for Winter and Summer respectively.

the linear case, we observe that the function responses on the ENSO variable are invariant across regimes. However, the response on the AIR variable varies across regimes, as we observe that the slope with respect to the ENSO input is zero in Winter, and increases slightly in Summer. This visualisation is consistent with the results reported in the regime-dependent graph (fig. 5). In the non-linear case, we now observe that the responses of the ENSO variable are slightly different, but the slope differences in the responses of the AIR variable with respect to the ENSO input are harder to visualise. The noticeable difference is that the self-dependency of the AIR variable changes non-linearly across regimes, contrary to the linear case where the slope with respect to AIR input was constant.

### F.6  SALSA EXPERIMENT

**Mocap sequences**  The data we consider for this experiment consists on 28 *salsa dancing* sequences from the CMU mocap data. Each trial consists of a sequence with varying length, where the observations represent 3D positions of 41 joints of both participants. Following related approaches (Dong et al., 2020), we use information of one of the participants, which should be sufficient for capturing dynamics, with a total of $41 \times 3$ observations per frame. Then, we subsample the data by a factor of 4, normalise the data, and clip each sequence to $T = 200$.

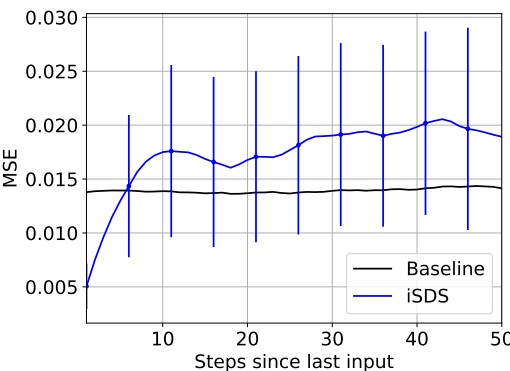

Figure 11: Forecasting averaged pixel MSE and standard deviation (vertical lines) of our iSDS using test data sequences.

**Video sequences**  To train SDSs, we generate $64 \times 64$ video sequences of length $T = 200$. The dancing sequences are originally obtained from the same CMU mocap data, but they are processed into human meshes using Mahmood et al. (2019) and available on the AMASS dataset. The total processed samples from CMU salsa dancing sequences are 14. To generate videos, we subsample the sequences by a factor of 8, and augment the data by rendering human meshes with rotated perspectives and offsetting the subsampled trajectories. To do so, we adapt the available code from Mahmood et al. (2019), and generate 10080 train samples and 560 test samples. In figure 12 we show examples of reconstructed salsa videos from the test dataset using our identifiable SDS, where we observe that the method achieves high-fidelity reconstructions. Additionally, in Figure 11 we provide forecasting results on the test data for 50 future frames from the last observation. As a reference, at each frame we compare the iSDS predictions

with a black image (indicated as baseline). As we observe, the prediction error increases rapidly. More specifically, the averaged pixel MSE for the first 20 predicted frames is $0.0148$, which is high compared to the reconstruction error ($2.26 \cdot 10^{-4}$). Nonetheless, this result is expected for the following reasons. First, the discrete transitions are independent of the observations, which can trigger dynamical changes that are not aligned with the groundtruth transitions. Second, the errors are accumulated over time, which combined with the previous point can rapidly cause disparities in the predictions. Finally, our formulation does not train the model based on prediction explicitly. Although the predicted frames are not aligned with the ground truth, in Figure 13 we show that our iSDS is able to produce reliable future sequences, despite some exceptions (see 6th forecast). In general, we observe that our proposed model can be used to generate reliable future dancing sequences. We note that despite the restrictions assumed to achieve identifiability guarantees (e.g. removed feedback from observations in comparison to Dong et al. (2020)), our iSDS serves as a generative model for high-dimensional sequences.

## F.7 REAL DANCING VIDEOS

To further motivate the applications of our identifiable SDS in challenging realistic domains, we consider real dancing sequences from the AIST Dance DB (Tsuchida et al., 2019). As in the previous semi-synthetic video experiment, we focus on segmenting dancing patterns from high-dimensional input. The data contains a total of $12670$ sequences of varying lengths, which include $10$ different dancing genres (with $1267$ sequences each), different actors, and camera orientations. We focus on segmenting sequences corresponding to the *Middle Hip Hop* genre, where we leave $100$ sequences for testing. We process each sequence as follows: (i) we subsample the video by 4, (ii) we crop each frame to center the dancer position, (iii) we resize each frame to $64 \times 64$, and (iv) we crop the length of the video to $T = 200$ frames.

For training, we adopt the same architecture and hyper-parameters as in the salsa dancing video experiment (See Appendix F.3; except we set a batch size of 16 this time). Here we also include a pre-training stage for the encoder-decoder networks, but in this case we include all the available dancing sequences (all genres, except the test samples). In the second stage where the transitions are learned, we use only the *Middle Hip Hop* sequences. We note that training the iSDS in this dataset is particularly challenging, as we find that the issue of state collapsed reported by related works (Dong et al., 2020; Ansari et al., 2021) is more prominent in this scenario. To mitigate this problem, we train our model using a combination of the KL annealing schedule proposed in Dong et al. (2020) and the temperature coefficient proposed in Ansari et al. (2021), which we already included previously. We start with a KL annealing term of $10^4$ and decay it by a factor of 0.95 every 50 iterations, and with $\tau = 10^3$, where we decay it by a factor of 0.975 every 100 iterations. We run this second phase for a maximum of 1000 epochs.

We show reconstruction and segmentation results in Figure 14, where we observe our iSDS learns different components from the dancing sequences. In general we observe that the sequences are segmented according to different dancing moves, except for some cases where only a prominent mode is present. We also note that different prominent modes are present depending on background information. For example, the *blue* mode is prominent for white background, and the *teal* mode is prominent for combined black and white backgrounds. Such findings indicate the possibility of having data artifacts, in which the model performs segmentation based on the background information as they could be correlated with the dancing dynamics. Furthermore, our iSDS reconstructs the video sequences with high fidelity, except for fine-grained details such as the hands. Quantitatively, our approach reconstructs the sequences with an averaged pixel MSE of $7.85 \cdot 10^3$. We consider this quantity is reasonable as it is an order of magnitude higher in comparison to the previous salsa videos, where the sequences were rendered on black backgrounds.

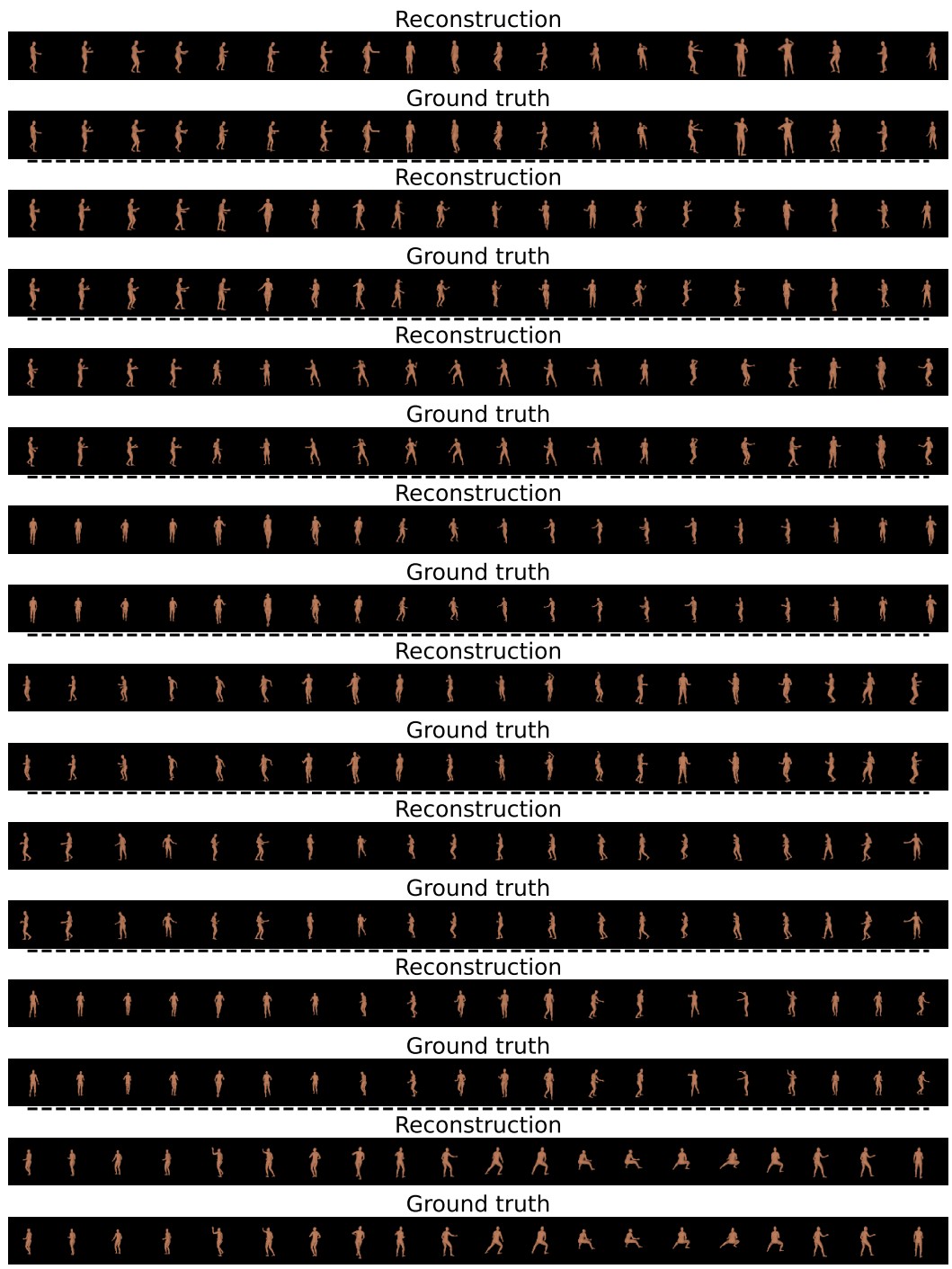

Figure 12: Reconstruction and ground truth of salsa dancing videos from the test dataset.

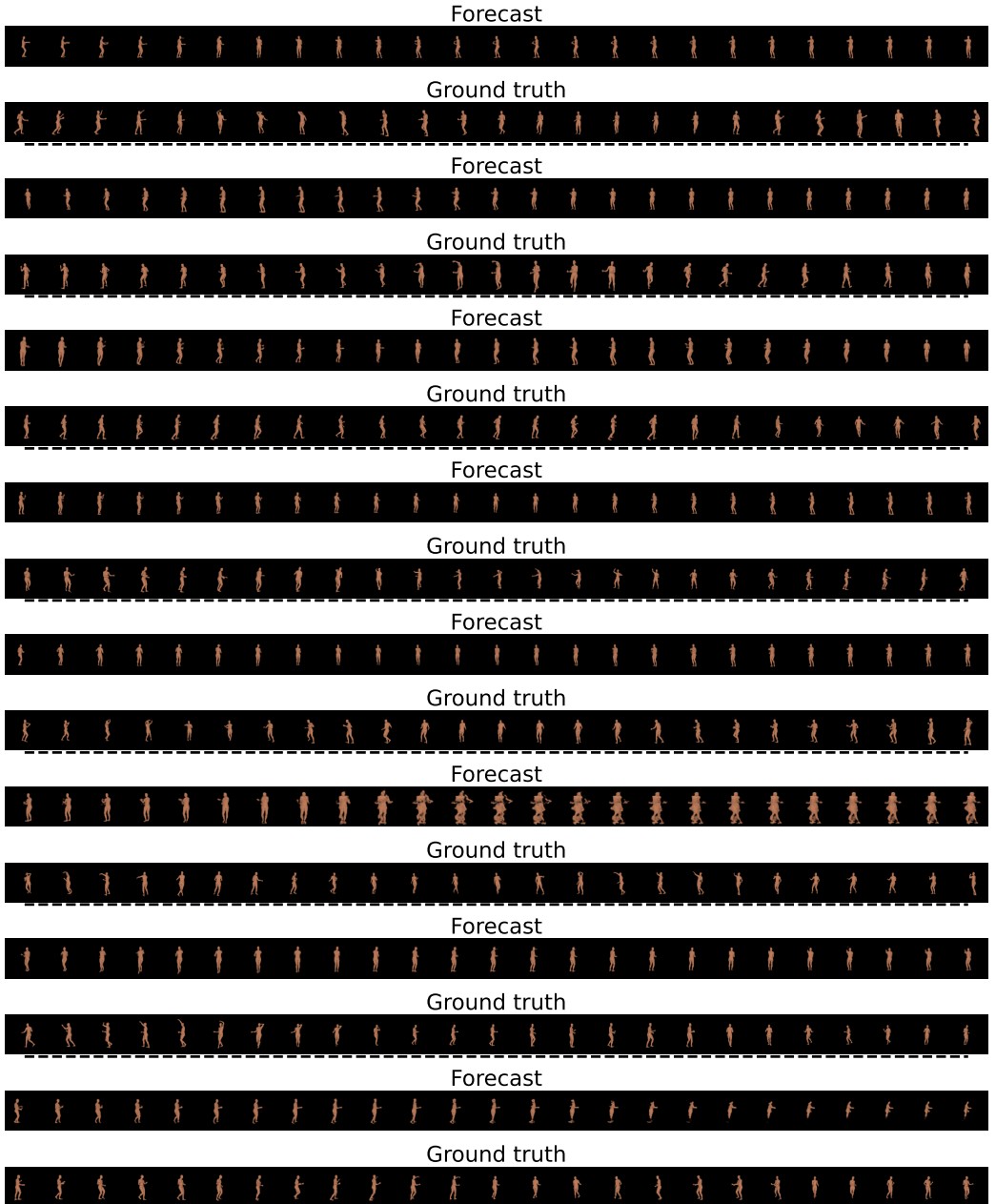

Figure 13: Forecasts and corresponding ground truth of future $T = 50$ frames from salsa test samples.

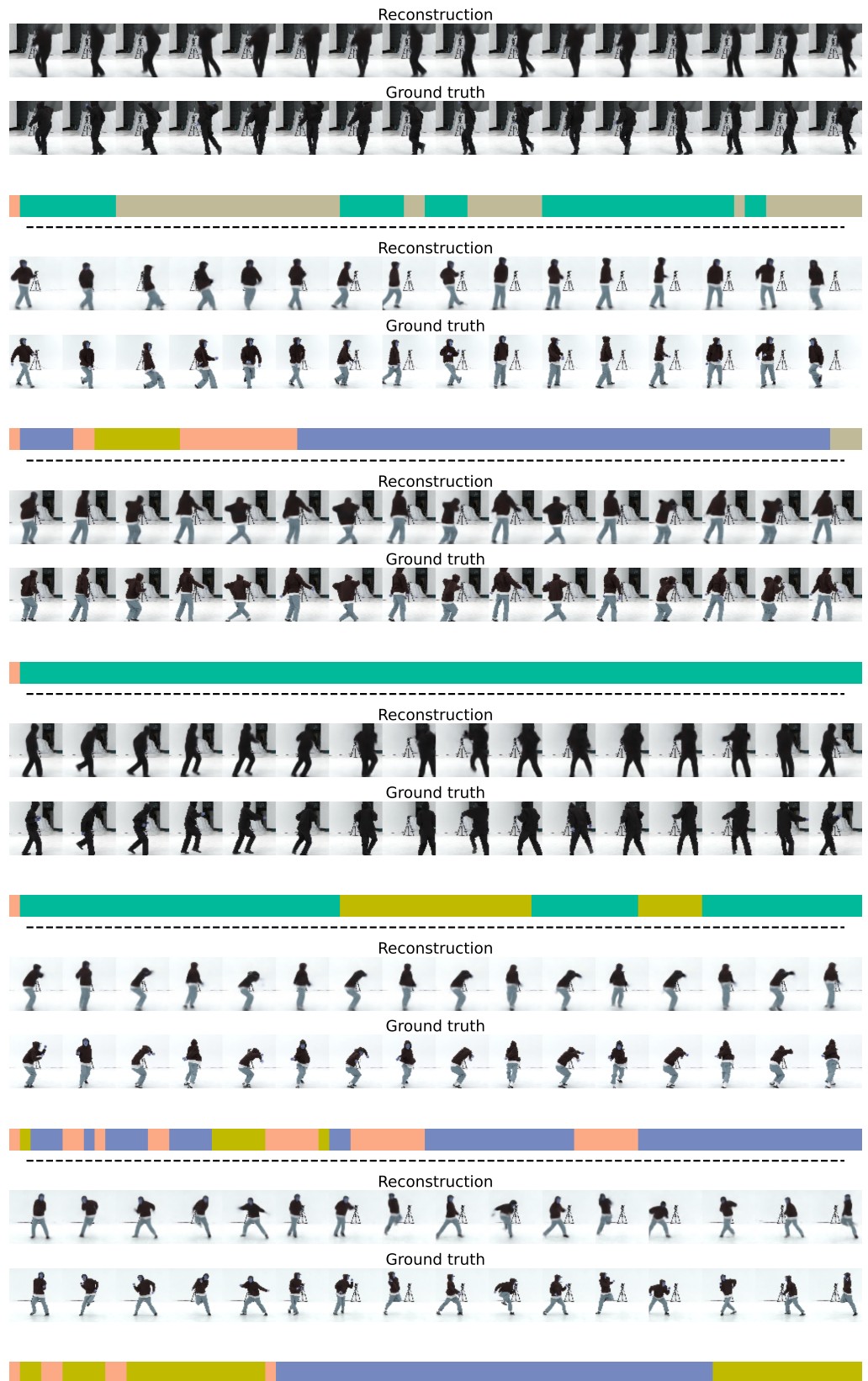

Figure 14: Reconstruction, ground truth, and segmentation of dancing videos from the AIST Dance DB test set.

