# OpenReview forum: "On the Identifiability of Switching Dynamical Systems"
_ICLR.cc/2024/Conference — Submitted to ICLR 2024_

### Official Review · Reviewer_QaMu · 2023-10-25

**Soundness:** 3 good
**Presentation:** 2 fair
**Contribution:** 2 fair
**Rating:** 6
**Confidence:** 3

**Summary:**

The authors present a paper on identifiability in MSMs and SDS, using Gaussian transitions and neural networks for parameterisation. The study's findings are empirically verified with synthetic experiments and suggest practical applications for causal discovery and time series segmentation in real and synthetic data.

**Strengths:**

The main review is concentrated to this section owing to the flow in which this review was conducted.


### Introduction

- Your definition of causal identifiability is wrong. You say: "In causal inference (Peters et al., 2017), identifiability refers to whether the underlying causal structure can be correctly pinpointed from infinite observational data." - that is not at all what identifiability means in causal inference. It means whether or not an intervention e.g. $P(Y \mid do(X=x))$ can be computed only from knowledge of the observational distribution $P(V)$ where $V$ is the set of all variables in the causal model, and the causal diagram $\mathcal{G}$. What you are referring to has more to do with _causal discovery_ which is the process of learning the true DAG from the observational data.

### Background

- Is it important that $K < +\infty$?
- It may help readability if you use different symbols, rather than $\theta$ for everything, for the different densities.

### Theoretical considerations

- This section may be better called 'Methods'. The current heading is a bit ambiguous - theoretical considerations of what?
- I do not understand, what does $\{1,\ldots, K^T\} \rightarrow \{1,\ldots,K\}^T$ mean?
- What are these mild measure-theoretic considerations? What does 'mild' mean in this context?
- You are interchangeably using | and \mid in your densities, pick one.
- Typo: 'indentifiability' under definition 3.2

### Related work

- Again, your definition of causal identifiability does not mean what you say it does.

### Experiments

- I am confused. In section 6.2 you are using the correct term 'causal discovery' and you are using the idea correctly as well, so why are you referring to the structure learning with 'identifiability' hitherto?

### Conclusion

- Same here; _causal discovery_ is being used rather than identifiability.

**Weaknesses:**

See Strength section.

**Questions:**

See Strength section.

---

> ### Author Response · Authors · 2023-11-16
> **Response to reviewer (1/2)**
>
> We thank the reviewer for their time and constructive feedback on our work. We hope to engage in discussions during this period to further improve the quality of this manuscript.
>
> **Summary**
>
> We first provide a summary of the issues outlined by the reviewer and the actions we will take accordingly.
>
> - The reviewer raises concerns in the terminology referring to identifiability in the context of causal discovery, as it can give the wrong impression that we are claiming structure identifiability.
> - **Action**: We update the document fixing the terminology, so that it becomes clear that although our results allow causal discovery in regime-dependent structures (thanks to having access to the transition derivatives), we are not claiming structure identifiability.
>
>
> **Response Details**
>
> Below we provide clarifications for each of the issues outlined in the reviewer assessment.
>
>
> > Your definition of causal identifiability is wrong.
>
> Considering our task regarding causality does not consider intervention distributions and we are working with observational data, we agree that the proper term would be causal discovery instead of causal inference. With this being said, the notion of identifiability holds true for causal discovery (whether the underlying causal graph can be determined from observational data).
>
> We thank the reviewer for spotting this and we will update the terminology regarding causal inference to causal discovery.
>
>
> > Is it important that $K < \infty$?
>
> It is crucial to set $K < \infty$ (and also $T < \infty$), as we are framing our Markov Switching Model as a finite mixture model. This notion is generally true for State-Space Models with discrete variables. In this case, it helps us understand that although the number of mixture components grows exponentially ($K^T$), we can still use finite mixture model results as $T$ and $K$ are finite.
>
> > It may help readability if you use different symbols, rather than $\theta$ for everything, for the different densities.
>
> We use $\theta$ to denote the parametrization of the distributions of our models. We used a single variable for simplification. We consider using different names for the decoders, discrete prior dynamics, and continuous prior dynamics (initial and transition distribution) could become cumbersome.
>
> It also simplifies the formulation we use in the Estimation section, as we denote the parameters of the generative model with $\theta$ and the approximate posterior with $\phi$.
>
>
> > This section may be better called 'Methods'. The current heading is a bit ambiguous - theoretical considerations of what?
>
> Experiments aside, note that our main contributions are purely theoretical, i.e. we are not proposing any novel probabilistic modeling technique/method. The generative models we study come from a family of methods presented in recent years: KVAE (Fraccaro et al. 2017), SNLDS (Dong et al. 2020), RED-SDS (Ansari et al. 2021), etc. Therefore, we consider that a heading like “Methods” would give the reader the incorrect feeling that we are proposing a novel model.
>
> We first present this notion in the “Background section”, and then present our theoretical results in the next section. We named the section “Theoretical Considerations” to highlight that the core contribution of our work is theoretical, meaning that we are proving identifiability for well-established generative models which have been used in the past with no guarantees. The first paragraph of the “Theoretical Considerations” section explains what are such considerations (identifiable MSM and identifiable SDS).
>
> Other previous papers have also used this heading, such as RHINO (Gong et al 2023), or similar ones such as “Identifiability Theory” by HMM-ICA (Halva et al. 2020) or iVAE (Khemakem et al 2020).

---

> ### Author Response · Authors · 2023-11-16
> **Response to reviewer (2/2)**
>
> > I do not understand, what does $1,\dots,K^T \rightarrow 1, \dots, K^T$ mean?
>
> We believe the reviewer is referring to the bijective *path indexing function* $\varphi: ${$1,\dots,K^T$}$ \rightarrow ${$1, \dots, K$}$^T$. This path indexing function is important because it allows us to translate the original notation used in Markov Switching Models to a notation which is suitable for a finite mixture model of $K^T$ components and vice versa. See below for further clarifications.
>
> - Consider a Markov Switching Model trajectory, i.e $p(z_{1:T}| s_{1:T})$. We can represent the state assignment for such trajectory using a tuple $(s_1=i_1, s_2=i_2, \dots, s_T=i_T)$, where the space of the tuple is the cartesian product {$1, \dots, K$}$^T$.
> - Using this notation for finite mixture models can be cumbersome. For example, the marginalization of $s_{1:T}$ requires T summations.
> - To simplify this, we use the *path indexing function*, which uniquely retrieves a set of states from a single index (in this case a number between $1$ and $K^T$). By presenting this bijection, we can change the original notation of MSMs to finite mixtures. Following the previous example, the marginalization now takes only a big sum.
> The resulting notation with indices instead of tuples allows the proofs to be relatively easier to follow.
>
> > What are these mild measure-theoretic considerations? What does 'mild' mean in this context?
>
> These conditions can be found in Appendix A, which is indicated in the main text. It basically means that the measure we use is well-defined.
>
> > Again, your definition of causal identifiability does not mean what you say it does.
>
> We are a bit confused with this sentence, as we do not refer to “causal identifiability” in this section. The only reference to causality we make is to summarize some contributions regarding latent causal models such as CITRIS and LEAP.
>
> > I am confused. In section 6.2 you are using the correct term 'causal discovery' and you are using the idea correctly as well, so why are you referring to the structure learning with 'identifiability' hitherto?
>
> We believe the confusion comes from the following erroneous claim found in the introduction: “This also leads to the first identifiability proof for regime-dependent causal discovery…” We acknowledge this is not accurate to our results, as we are providing identification in terms of the MSM transitions, rather than structure identifiability.
>
> Although we mention structure identifiability to provide context of the main goal for causal discovery, we will make sure the erroneous claims are modified.
>
> > Same here; causal discovery is being used rather than identifiability.
>
> We believe this is referred to the following sentences:
>
> “We empirically verify our theoretical results with synthetic experiments, and motivate our approach for regime-dependent causal discovery and high-dimensional time series segmentation with real data.”
>
> “While our work focuses on identifiability analysis, in practice accurate estimation is also key to the success of causal discovery/representation learning from real data.”
>
> We use the term causal discovery to denote the task of learning graphs from observational data, and we believe that the term is correctly used in this case.

---

> ### Author Response · Authors · 2023-11-21
> **Reminder for Reviewer QaMu**
>
> Dear Reviewer QaMu,
>
> Once again, we appreciate your time in reviewing our work. As the discussion period is finishing soon, we would be grateful if you could confirm whether our reply and modifications to the document addressed your concerns.

---

> > ### Comment · Reviewer_QaMu · 2023-11-22
> > **Acknowledgement**
> >
> > Thanks to the authors for a thorough response. I confirm that I have read their review and have no further questions or comments.

---

### Official Review · Reviewer_nPz5 · 2023-10-30

**Soundness:** 4 excellent
**Presentation:** 3 good
**Contribution:** 3 good
**Rating:** 6
**Confidence:** 4

**Summary:**

This work focuses on the identifiability of Markov Switching Models and extended it to Switching Dynamical Systems. The authors presented identification conditions for first-order Markov models that uses non-linear Gaussian transitions. They proved the identifiability of Switching Dynamical Systems up to affine transformations. They also developed corresponding estimation algorithms. The proposed method was demonstrated by empirical studies on time series such as videos and climate data.

**Strengths:**

Conditions in which first-order MSMs with non-linear Gaussian transitions are identifiable up to permutations.

Analysis of identifiability conditions for non-parametric first-order MSMs.

Conditions for SDSs identifiability up to affine transformations of the latent variables and non-linear emission.

Discovery of time-dependent causal structures in time-series.

**Weaknesses:**

The conditions of identifiability such as invertibility are strong and thus less nontrivial, and would limit the practicality.

The empirical studies lack demonstration of identifiability or the benefits of identifiability.

**Questions:**

Could the authors discuss how useful the identifiability would be in practice, especially for those latent variables? I'm not questioning the contribution. I have seen papers talking about the identifiability subject to certain transformations, but few showcase it in their examples.

The index $a$ and $s$ seem exchangeable. What's the difference?

---

> ### Author Response · Authors · 2023-11-16
> **Response to reviewer (1/2)**
>
> We thank the reviewer for their time and constructive feedback on our work. We hope to engage in discussions during this period to find ways to improve our manuscript.
>
> **Weaknesses**
>
>
> First, we address the **weaknesses** outlined by the reviewer.
>
> > The conditions of identifiability such as invertibility are strong and thus less nontrivial, and would limit the practicality.
>
> We agree that invertibility can be considered a strong assumption which can limit the practicality of our approach when extending it to realistic domains. However, the invertibility condition only applies to the emission of the Switching Dynamical System, and not the transitions of the Markov Switching Models where the requirement is smoothness (analytic functions).
>
> The invertibility condition of the emission function in the Switching Dynamical System can be weakened to weakly injective functions (injective functions on at least an open set), which in turn gives us weakened identification results (**Theorem 3.2.ii**: The prior MSM is identifiable up to affine transformations).
>
> For some extra context on the emission assumptions, other related works such as HMM-ICA (Halva et al. 2020), SNICA (Halva et al. 2021), LEAP (Yao et al. 2022),  and CITRIS (Lippe et al. 2022), also assume injectivity or even bijectivity (which is much stronger). In this case, the weakly-injectivity assumption which we include in our work is much weaker but rarely used in other related works.
>
> > The empirical studies lack demonstration of identifiability or the benefits of identifiability.
>
> We consider the empirical studies to showcase some of the benefits of identifiability. Therefore, we believe some discussion here could be very helpful in terms of improving this part.
> - For the case with Markov Switching Model, the direct benefit of identifiability we aim to depict in our work is identifying regime-dependent causal structures. In causal discovery, there are few works which consider non-stationary time series, and even fewer which allow time-dependent causal structures. In this case, identifiability results for regime-dependent dynamics are helpful to understand data with e.g. seasonality trends. As we observe in the case we show (Section 6.2). We would like to note that regime-dependent causal discovery has not been proven identifiable, and our work establishes a first result on this matter by directly identifying functions.
> - For Switching Dynamical Systems, the benefits of identifiability are a bit more complex to demonstrate at this stage. For example, in case we had a latent causal structure underlying the continuous latent variables, the affine transformation would prevent us from correctly depicting such structure. For this reason, we showcased some results with 2D MSM rendered as videos, where the learned latent space is an affine transformation of the original one. For the case with salsa data, the problem relies on estimation of a complex and high-dimensional setting, which results in the method being very prone to state collapse. This motivates progressing our line of work further with consistent estimation methods for SDSs. We continue this discussion on the **questions** section.
>
> Of course, all of the cases with real data come at the cost of assuming correct model specification, i.e. whether the assumptions are in fact aligned with the *true underlying distribution*. In this case, identifiability theory is very important to understand how far we can go in terms of making interpretations from the learned structures (e.g. in the case of causal discovery).

---

> ### Author Response · Authors · 2023-11-16
> **Response to reviewer (2/2)**
>
> **Questions**
>
> We address the **questions** below:
>
> > Could the authors discuss how useful the identifiability would be in practice, especially for those latent variables?
>
> This question is in line with the previous item. We would like to provide further motivation for the practicality of identifiability with the following previous result.
>
> Fraccaro et al. (2017) [1] perform an experiment using a switching dynamical system (KVAE) with video data of a bouncing ball. In their experiments, they show a visualization of the 2D latent space, which results in a linear transformation of the original box in which the ball was contained. This empirical result suggests one should be able to prove that indeed there exists a one to one correspondence between the latent space distribution and observed distribution up to an affine transformation. Unfortunately such results cannot be found in the literature.
>
> In our work, the distribution of the discrete states is independent of the observations, which means that the bouncing ball example is not embedded within our theoretical findings. However, we choose to depict an equivalent example using 2D MSMs rendered as image data to showcase that one can recover the true latents up to affine transformations purely from images (by using (Leaky)-ReLU CNNs) as decoders.
>
> We can find further motivation for identifiable Switching Dynamical Systems with another example on climate data. In this case, imagine we cannot quantify the climate variables (and model them as MSMs), but instead, we have access to some high-dimensional representation of them (e.g. precipitation maps, etc). Given the seasonal nature of climate variables, approaches such as the one we present could be a good fit for identifying causal structures from high-dimensional inputs. For a visualization, see [Figure 1 of Runge et al. (2020)](https://www.nature.com/articles/s41467-019-10105-3) [2]. However, since we only achieve affine transformation identifiability, we would need to impose further restrictions to our model to reduce the affine transformation to e.g. component-wise scaling and permutation.
>
> We believe our manuscript could benefit from including a similar discussion in the Appendix to motivate the importance of identifiable Switching Dynamical Systems.
>
> [1] Fraccaro, M., Kamronn, S., Paquet, U. and Winther, O., 2017. A disentangled recognition and nonlinear dynamics model for unsupervised learning. Advances in neural information processing systems, 30.
>
> [2] Runge, J., Bathiany, S., Bollt, E., Camps-Valls, G., Coumou, D., Deyle, E., Glymour, C., Kretschmer, M., Mahecha, M.D., Muñoz-Marí, J. and Van Nes, E.H., 2019. Inferring causation from time series in Earth system sciences. Nature communications, 10(1), p.2553.
>
> > The index a and  s seem exchangeable. What's the difference?
>
> The indices $a$ we use in this case are used to show how to understand the Markov Switching Model family as a finite mixture model (finite mixture of $K^T$ trajectories). The idea is to drop the conditioning discrete state variables so that we can **write the MSM as a linear combination of functions in $\mathbb{R}^{Tm}$.** We expand below for further clarification.
> - The notation using $s$ is the original used for indexing the state value at each time step. This notation becomes cumbersome when formulating the MSM as a finite mixture, as the state assignments in a trajectory are a tuple of length T, where the space of assignments becomes a cartesian product {$1, …, K$}$^T$.
> - To see why the above notation is cumbersome, we can imagine trying to marginalize the state variables from $p(z_{1:T},s_{1:T})$, where the we would have T summations, and the state assignment would be $s_{1:T} = (i_1, \dots, i_T)$.
> - To simplify this, we will just consider that our problem is indeed a mixture of $K^T$ trajectories. Then, we can use another indexing variable $i\in ${$1, \dots, K^T $}, which will be a number (and not a tuple). Now the trajectory indexing becomes much simpler as we  can use a single number. In this case, we introduce the indices $a^i_{1:T}$ for each $i$, that will retrieve the corresponding initial or transition functions for the family we defined.
> - The idea is that using $s$ and $a$ is interchangeable. Note that we indicate that they are equivalent notations. However, it facilitates work when framing MSMs as finite mixture models and thus proving identifiability, as we now have one big linear combination of functions.

---

> ### Comment · Reviewer_nPz5 · 2023-11-21
>
> Thank you for addressing my concerns.
> Since the identifiability is a crucial point of this work, it is worthwhile to elaborate on those advantages of identifiability in comparison to other non-identifiable methods in the empirical examples.
> I have no further questions.

---

> > ### Author Response · Authors · 2023-11-22
> > **Response to Comment**
> >
> > Dear Reviewer,
> >
> > We appreciate your response. We would like to give some additional views on the importance of identifiable models over non-identifiable ones.
> > - Assuming no model misspecification (i.e. assumptions are aligned with reality), identifiable models can be useful in terms of interpretability of the learned representations. We showcase this view in our causal discovery experiment, where we find that the non-linear model learns graphs which do not agree with expert knowledge, possibly due to confounding issues. Note that any other algorithm with no identifiability guarantees cannot perform the same analysis, as multiple graphs could be allowed in the solution that maximises likelihood.
> > -  Of course, if one cares about pure performance (e.g. in terms of forecasting), models that have less restrictions (and thus possibly losing identifiability guarantees) will probably have the edge. In our empirical analysis, we attempt to motivate that while identifiability guarantees restrict our models in terms of e.g. not being able to consider additional dependencies, they are still competitive in terms of segmenting real videos.
> > - Most importantly, without identifiability, it is not possible to prove consistency of any estimation algorithm, as multiple parameter arrangements could give the same minima in the loss function.

---

### Official Review · Reviewer_JVt8 · 2023-10-31

**Soundness:** 3 good
**Presentation:** 2 fair
**Contribution:** 3 good
**Rating:** 6
**Confidence:** 3

**Summary:**

This work proposes an identifiability results of switching dynamical systems. The results are a direct extension of Kivva et al. (2022). The main idea is to show that the distribution under the switch dynamic system can be represented through finite mixture distributes which are indeed identifiable.

**Strengths:**

- The identifiability results of switching dynamical systems are proposed.

**Weaknesses:**

- Does the number of states need to be specified as prior information?
- Whether the number of states is identifiable and how to select it practically.
- It would be better if there has a simple example to show the identifiability of the model in theory.
- Moreover, the definition of the identifiability in this work should be stated clearly.

**Questions:**

See the weaknesses above.

---

> ### Author Response · Authors · 2023-11-16
> **Response to reviewer (1/2)**
>
> We thank the reviewer for their time and feedback on our work, and we hope we can engage in further discussions.
>
> We first clarify some concerns raised in the **summary** statement.
>
> > The results are a direct extension of Kivva et al. (2022).
>
> We would like to clarify some **contributions** for the theory part, which we believe to be very important in terms of significance and novelty.
> - Although we use results from Kivva et al. (2022), they are only used for establishing the identifiability of the Switching Dynamical System given an identifiable Markov Switching Model prior.
> - Establishing results identifiability for non-linear Markov Switching Models is a non-trivial task even in the first-order case. To the best of our knowledge, our work is the first contribution to showcase results with non-linear Gaussian transitions (shown in the main text), and more importantly non-parametric transitions (Appendix B). Since results on finite mixture models cannot be directly extended to MSMs, we provide a novel technique to prove their identifiability (See Lemma B.1).
> - Finally, the identifiability of the transitions for such highly non-linear State-Space Models has not been addressed despite the variety of contributions in recent years. As these proofs are generally non-trivial, we consider our result to be very significant for the SSM community. Approaches like Halva et al. (2021) [1] include SDSs, but there are no identifiability statements on the transition functions.
>
> Therefore, we cannot qualify our work as a direct extension of Kivva et al. (2022), as proving identifiability for non-parametric non-linear MSMs is a non-trivial contribution with independent significance.
>
> [1] Hälvä, H., Le Corff, S., Lehéricy, L., So, J., Zhu, Y., Gassiat, E. and Hyvarinen, A., 2021. Disentangling identifiable features from noisy data with structured nonlinear ICA. Advances in Neural Information Processing Systems, 34, pp.1624-1633.
>
> >  The main idea is to show that the distribution under the switch dynamic system can be represented through finite mixture distributes which are indeed identifiable.
>
> From this sentence it seems like the reviewer considers that if one writes a Markov Switching Model as a finite mixture, the model is identifiable. This is true **only when the mixture trajectories are linear independent**. Proving such a linear independence result is generally non-trivial, and one of our key contributions.

---

> ### Author Response · Authors · 2023-11-16
> **Response to reviewer (2/2)**
>
> Below we address the **weaknesses** outlined by the reviewer:
>
> > Does the number of states need to be specified as prior information?
>
> TL;DR: No.
>
> Just as in the non-temporal finite mixture case (e.g. Gaussian Mixture Model), the number of states **is identifiable from observations**, both for the MSM and SDS. This information can be found in **Definition 3.1**, where the notion of an identifiable MSM requires the number of states to be identifiable.
>
> Note that other recent ICA works which use discrete latent variables (HMM-ICA [2], IIA-ICA [3], or SNICA [1]), rely on Gasiat et al. (2016) [4], which requires the number of states to be known a priori. Therefore, our result here is much more stronger.
>
> [2] Hälvä, H. and Hyvarinen, A., 2020, August. Hidden markov nonlinear ica: Unsupervised learning from nonstationary time series. In Conference on Uncertainty in Artificial Intelligence (pp. 939-948). PMLR.
>
> [3] Morioka, H., Hälvä, H. and Hyvarinen, A., 2021, March. Independent innovation analysis for nonlinear vector autoregressive process. In International Conference on Artificial Intelligence and Statistics (pp. 1549-1557). PMLR.
>
> [4] Gassiat, É., Cleynen, A. and Robin, S., 2016. Inference in finite state space non parametric hidden Markov models and applications. Statistics and Computing, 26, pp.61-71.
>
> > Whether the number of states is identifiable and how to select it practically.
>
> As mentioned, the number of states is identifiable. Selecting the number of states in practice is a different notion than identifiability, which refers to the estimation method.
>
> As an example, we can think about the equivalent non-temporal case where we need to find the number of mixture components in a clustering problem. One can then use the MLE objective with models initialized with different numbers of states. The identifiability conditions paired up with the general consistency of the MLE solution guarantee that the model with the correct number of states will have the highest data log-likelihood.
>
> Note that this only works when the number of states is identifiable.
>
> > It would be better if there has a simple example to show the identifiability of the model in theory.
>
> In the first paragraph of section 3, we refer to our contribution as the temporal extension of a finite mixture model. In section 3.1, we adapt the notation of an MSM to finite mixture models, making the analogy where we now have a mixture of K^T trajectories.
>
> “This notation shows that the MSM extends finite mixture models to temporal settings as a finite mixture of $K^T$ trajectories composed by (conditional) distributions … “
>
> With this in mind, the identifiability of the MSM is close to the identifiability of a finite mixture model. Direct application of finite mixture model theory shows that linear independence of the $K^T$ mixings is a sufficient condition for identifiability. This explanation is depicted in the Proof sketch of **Theorem 3.1**, where we give the reader an overall idea of the proof technique, and how to prove linear independence using induction.
>
>
> > Moreover, the definition of the identifiability in this work should be stated clearly.
>
> We thank the reviewer for the suggestion. However, we believe this notion is already stated in Definitions 3.1 and 3.3, where we state the definition of identifiable Markov Switching Models, and the identifiability of prior distributions up to affine transformations respectively.
>
> For functions parameterized with neural networks, we refer to the paragraph after **Theorem 3.1**, second sentence, where we mention using analytic neural networks: “... In the latter case, the identifiability result applies to the functional form only, since network weights do not uniquely index the functions that they parameterise …”. For further clarification, we will update the manuscript re-stating this latter notion of identifiability on neural network functions for the SDS as well.

---

### Official Review · Reviewer_SfP9 · 2023-11-12

**Soundness:** 3 good
**Presentation:** 3 good
**Contribution:** 2 fair
**Rating:** 5
**Confidence:** 4

**Summary:**

This paper provides identifiability theorems for general dynamical systems with nonlinear transitions and emission functions. This is done first by showing identifiability for Markov Switching Models and then for Switching Dynamical Systems. Variational inference algorithms are presented for learning and posterior inference.

**Strengths:**

The presentation is good and the paper reads well with nice explanations in many places.

It is in general important work to study and expand our understanding of identifiability in these models. Here the identifiability is shown for a general class of MSMs and SDSs which is a nice result. In particular, the identifiability does not assume independence of the latent components unlike like the works in the nonlinear ICA area.

It is also nice to see experiments, albeit somewhat limited (see below), on real data despite the theoretical focus of the paper.

**Weaknesses:**

1) **The identifiability approach/results follow quite directly from previous works with some results already known previously** -- one could have assumed the results here to be already implied based on earlier works: the approach for proving MSM using mixtures model presentation is well known from the general work in [1] (which could be cited). While this specific class of models covered in this work is not explicitly mentioned, the results therein could be seen as already implying the identifiability of this type of models -- more explicit acknowledgement that this approach has been taken before would probably be sufficient (for the MSM results). The identifiabiltiy of SDSs with nonlinear activation function is just an application of the Kivva paper on this specific model class. Further identifiability of SDSs (in nonlinear ICA form) has actually already been established -- below point covers this more.

2) **There are lacking citations and incorrect interpretations of previous works** -- if these are acknowledged appropriately the contributions of this paper would appear less significant. Most importantly, this paper claims that previous works have not considered identifiability in nonlinear dynamical systems with nonlinear emission function: specifically the authors state that "In contrast, Hälvä et al. (2021) assumes linear SDSs," -- this is not correct,  their identifiability results do not appear to make any such assumptions (based on my reading of their Theorems 1 and 2); their practical algorithm does seem to assume that but that has nothing to do with the identifiability. Identifiability of SDSs is thus achieved in their work. For identifiability of models with autoregressive transitions, see also [2], and the IIA-HMM model particularly -- I don't see this paper mentioned and discussed even though it has some similarity. In terms of estimation algorithm, the seminal work of [3] is not mentioned -- you need to at least explain why their work does not apply here or why the current approach is superior.

3) **The identifiability results are weaker than acknowledged. or at least not sufficiently discussed** By directly relying on the result of Kivva to prove the identifiability of the nonlinear emission function / latent components, the results unfortunately inherit its weaknesses. 1) The identifiability requires one to know the family of the noise distribution which is a cumbersome assumption (but see Q2 below). 2) The results relies on piecewise linear emission function and is therefore less general than many other works that allow e.g. almost any injective $f(x)$ -- consider for instance what happens in your case if $f(x)$ is a Gaussian Process. 3) As far as I understand, that while there is no assumption of independence here (c.f. nonlinear ICA), the results are also clearly weaker i.e. for latent vector $\mathbf{z}$ your results give essentially $f(\mathbf{z}) = Af'(\mathbf{z'}) + b$ -- contrast this to nonlinear ICA where one would identify the individual coordinates as per $z_i = h(z_i')$ for some invertible $h$ -- please correct if I have misunderstood.

4) ** Experimental evaluation lacking in places**: The evaluation of the models on the salsa data seems very qualitative and much more rigorous evaluation would be preferred with some quantitative evaluation metrics.(Q3 below)

misc.:
- "p(z) is identifiable up to affine transformations" -- This is imprecise language, explain whether you mean parameter identificaiton or identification of z or what.
- Definition 2.1 define what $B$ is
- grammar: "The generative model consider in this work"
- Equation (4) and elswhere you use $p_a$ but earlier $p_{\theta}$, please explain more clearly what the subscript $a$ means
- Figure 2 is poor quality -- please provide a better quality figure
- "Hälvä et al. (2021) introduces time-dependence between sources via linear SDSs" , this appears to be incorrect as mentioned above (this mistake is in two places in the paper)



[1] Allman et al.(2009) Identifiability of parameters in latent structure models with many observed variables

[2] Morioka et al. (2021) Independent Innovation Analysis for Nonlinear Vector Autoregressive Process

[3] Johnson et al. (2017) Composing graphical models with neural networks for structured representations and fast inference

**Questions:**

Q1: What is the justification for you estimation algorithm in light of the existence of the work [3] (see reference above) including benefits, disadvantages?

Q2: Why do use the results of Khemakhem, Kivva that assumes that we know the noise terms distribution? Why do you not instead apply e.g. the results from Halva et al (2021), Theorem 1, that allows any arbitrary noise distribution?

Q3: Why is it not possible to apply a more quantitative performance metric in the salsa experiment -?

---

> ### Author Response · Authors · 2023-11-16
> **Reply summary**
>
> We thank the reviewer for their time and constructive feedback on our work, and we hope we can engage in further discussions to improve the quality of our manuscript.
>
> We first provide a summary of our reply. For extra details, we refer to the detailed response.
>
> **Reply summary**
> - Our contribution on the MSM part is both **novel** and **significant**.
>   - Novelty + Significance: We obtain identifiability results of the MSM transitions up to permutations in **non-parametric** case. For clarity, we only presented a parametric example (non-linear Gaussian) in the main text as a special case.
>
>   - Novelty: our proof technique is new, which relies on **a brand new theoretical result (Lemma B.1)**. The direction from Allman et al. (2009) does not work for general non-parametric MSMs.
> - The existing work suggested by the reviewer (namely Halva et al. (2021)) is restrictive in terms of **assumptions** and **identifiability** results.
>   - Assumptions: Stationarity of the noiseless distribution $f(s_t)$ is already a very big restriction in our setup, where instead we could have a nonstationary latent MSM.
>   - Identifiability: Only demixing identifiability is achieved, i.e. identifiability of $f$ and $f(s_t)$. The identifiability of latent source transitions is not established. Therefore, Halva et al. (2021) do not prove the identifiability of SDSs.
> - The main reason for extending to SDSs using Kivva et al. (2022) relies on our assumptions of analytic priors and closedness under affine transformations.

---

> ### Author Response · Authors · 2023-11-16
> **Detailed response to reviewer (1/3)**
>
> **Response details**
>
> Below we address the **weaknesses** outlined by the reviewer
>
> > The identifiability approach/results follow quite directly from previous works with some results already known previously
>
> We are aware of Allman et al. (2009) results, which directly lead to HMM identifiability in Gassiat et al. (2016). In fact our initial attempts followed Allman et al. (2009), but in the end we found it not applicable to non-parametric MSMs due to the following reasons.
> - The Allman et al. (2009) technique is based on expressing the marginal distribution as a mixture of tensors, where each tensor is an outer product of 3 vectors, and each of these vectors is effectively $p(z_t | s_t)$. Therefore this technique can only be applied to HMMs where the emission probability is dependent on the discrete state only. On the other hand, MSMs introduce explicit dependencies on the data via the transition dynamics $p(z_t|z_{t-1},s_t)$.
> - The Allman et al. (2009) technique needs to have at least $T=3$. It won’t work for $T=2$ and it’s easy to give counterexamples here. In contrast, our technique for MSM works for $T=2$ under our assumptions.
> - The key restriction of Allman et al. (2009) technique (see its Section 4), as well as in many subsequent papers including Gassiat et al. (2016), comes from the use of Kruskal’s “Three-Way Arrays” [1].
>
> Observing the limitations of Allman et al. (2009), we decided to go back to the very first finite mixture model identifiability result (Yakowitz & Spragins, 1968) as the starting point. This leads to our key and novel theoretical result (Lemma B.1) which is crucial for our MSM identifiability proof in the non-parametric case.
>
> [1] Kruskal, J.B., 1977. Three-way arrays: rank and uniqueness of trilinear decompositions, with application to arithmetic complexity and statistics. Linear algebra and its applications, 18(2), pp.95-138.
>
> > There are lacking citations and incorrect interpretations of previous works.
>
> We thank the reviewer for carefully comparing our work to existing literature. We will make sure previous work is properly acknowledged (including Morioka et al. (2021)) and modify the reference to other studies using general SDSs (which can be covered by Halva et al. 2021) by highlighting the assumptions on temporal dependence, rather than the assumptions on the transition model.
>
> Nonetheless, we believe our contribution is significant even though other works have studied similar models.
> - The SDS assumptions of Halva et al. (2021) are a combination of the ones used for HMM-ICA (Halva et al. 2020), and Hyvärinen et al. (2017) [2], thereby inheriting their limitations.
>   - Regarding HMM-ICA (Halva et al. 2020), the discrete states $s_t$ are assumed to follow a first-order stationary Markov chain with a full-rank transition matrix. The number of states are also required to be known. Instead, **our work makes no assumptions on knowing the number of states, or the structure of the dynamic model of $s_t$,** meaning it can be e.g., non-stationary and/or higher-order Markov.
>   - Regarding Hyvärinen et al. 2017 [2], while it is true that they do not impose linearity the assumptions are placed on the temporal dependencies between sources. Our strategy works in a different spirit, where we directly establish restrictions on the transition mapping itself, which is more intuitive.
> - The identifiability results of Halva et al. (2021), although stronger in terms of achieving disentanglement of the sources, are weaker in terms of identifying the transitions in the latent space.
>   - Their main results (Thms 1 & 2) prove the identifiability of the non-linear mixing and source distribution, but make **no identifiability statements about the transition dynamics**.
>   - The Linear SDS assumed in Halva et al. (2021) is overcomplete ( observation dimensions < latent dimensions), which is problematic in terms of achieving transition identifiability.
> - We believe the following claim “Identifiability of SDSs is thus achieved in Halva et al. (2021)” is not accurate.
>   - Identifiability results for HMMs include identifying the discrete transitions and source emission [3] (which would be transitions for SDSs). Halva et al. (2021) provides no result in this regard for SDS.
>   - Existing HMM identifiability results do not transfer to MSMs due to the explained limitation of Allman et al. (2009) technique.
> - Morioka et al. (2021) provides transition identifiability for vector autoregressive models (VAR), where only one transition function is used. On the contrary, Markov Switching Models propose using many transition functions conditioned on the discrete state variable. The extension IIA-HMM introduces an identifiable HMM from [3] thanks to source identification, but does not provide additional transition functions conditioned on the discrete variables.

---

> ### Author Response · Authors · 2023-11-16
> **Detailed response to reviewer (2/3)**
>
> [2] Hyvarinen, A. and Morioka, H., 2017, April. Nonlinear ICA of temporally dependent stationary sources. In Artificial Intelligence and Statistics (pp. 460-469). PMLR.
>
> [3] Gassiat, É., Cleynen, A. and Robin, S., 2016. Inference in finite state space non parametric hidden Markov models and applications. Statistics and Computing, 26, pp.61-71.
>
> > The identifiability results are weaker than acknowledged. or at least not sufficiently discussed
>
> Although from Kivva et al. (2022) results we might have weaknesses in terms of the identification results (transitions and emission up to affine transformations), we argue that they have other advantages that can be interesting depending on the application.
> - The results from Kivva allow identifiability up to affine transformations of the prior distribution (which is identifiable), and hence we still achieve identifiability of the discrete state distribution up to permutations.
> - The results from Kivva also allow weaker assumptions than the injective mapping assumed in Halva et al. (2021), which uses a weakly-injective (and continuous) mapping (e.g. ReLU networks).
> - Following the previous point, note that weaker assumptions generally imply weaker identifiability statements. Therefore, our approach does not achieve decoder component-wise identifiability as we are not assuming independence as in ICA.
>
> We follow this discussion point in Q2.
>
> > Experimental evaluation lacking in places
>
> We agree that the evaluation of salsa data is focused on qualitative visualizations, although we provide reconstruction error metrics for reference. We continue this point in Q3.
>
> > misc:
>
> We thank the reviewer for carefully revising our writing and we will make sure these changes are implemented accordingly. The only thing to note is that we already introduce the new indexing notation in the last sentence of page 3 as follows:
>
> “Combined with the path indexing function, this establishes an injective mapping $\phi \circ \varphi^{-1}$ to uniquely map a set of states $s_{1:T}$ to the $a_{1:T}$ indices, and we can view $p_{a_1^i}(z_1)$ and $p_{a_t^i}(z_t |z_{t-1})$ as equivalent notations of $p(z_1 | s_1)$ and $p(z_t |z_{t-1}, s_t)$ respectively for $s_{1:T} = \varphi(i)$. This notation shows that the MSM extends finite mixture models to temporal settings as a finite mixture of $K^T$ trajectories composed by (conditional) distributions in $\Pi_{\mathcal{A}}$ and $\mathcal{P}_{\mathcal{A}}$”
>
> We will add a sentence clarifying that the subscript $\theta$ is dropped for simplicity.

---

> ### Author Response · Authors · 2023-11-16
> **Detailed response to reviewer (3/3)**
>
> **Questions**
>
> Below we address the **questions** raised by the reviewer.
>
> > Q1: What is the justification for you estimation algorithm in light of the existence of the work [3] (see reference above) including benefits, disadvantages?
>
> We appreciate the suggestion of an alternative estimation algorithm. The main justification for using approaches similar to Dong et al. (2020) [4] and Ansari et al. (2021) [5] is resorting to more recent techniques found in the literature. We note that SVAE (Johnson et al. (2017)) has been discussed in Dong et al. (2020) mentioning some of its disadvantages, which we provide below:
> - Although they combine exact Kalman Filtering and smoothing, they assume **linear** dynamics.
> - They assume a mean field posterior over continuous and discrete dynamics, which can limit the expressiveness of the inference network.
> An additional reason to consider [4] and [5] is the fact that they directly address the problem of state collapse, which is a very important issue to consider when training SDSs (especially nonlinear SDSs).
>
> [4] Dong, Z., Seybold, B., Murphy, K. and Bui, H., 2020, November. Collapsed amortized variational inference for switching nonlinear dynamical systems. In International Conference on Machine Learning (pp. 2638-2647). PMLR.
>
> [5] Ansari, A.F., Benidis, K., Kurle, R., Turkmen, A.C., Soh, H., Smola, A.J., Wang, B. and Januschowski, T., 2021. Deep explicit duration switching models for time series. Advances in Neural Information Processing Systems, 34, pp.29949-29961.
>
> > Q2: Why do use the results of Khemakhem, Kivva that assumes that we know the noise terms distribution? Why do you not instead apply e.g. the results from Halva et al (2021), Theorem 1, that allows any arbitrary noise distribution?
>
> We can argue using Kivva et al. (2022) from 3 different perspectives
> - Practicality: The reason for using Kivva et al. (2022) results rather than Halva et al. (2021) is the problem application (we are not doing ICA). Our important contribution is the work on identifiable MSMs, for which we provide identification results in the non-parametric case (see Appendix B) using a novel technique, given that Allman et al. (2009) results are not applicable to our setup.
> - Simplicity: Using Kivva et al. (2022) for SDSs is a natural extension of the previous model, where we use the fact that the MSM prior is analytic and closed under factored affine transformations.
> - Assumptions: Some of the assumptions that appear in Theorem 1 are not attractive to us (e.g. marginal distribution of $f(s_t)$ is stationary; $s_t$ denote sources). Note that our identifiable MSM allows nonstationary marginal continuous latent distributions (and therefore nonstationary $f(s_t))$, thanks to imposing no restrictions on the discrete state distribution.
>
> We appreciate the reviewer’s interest in the work of Halva et al. (2021). In future work, we will consider embedding our identifiable MSM within the SNICA setup to achieve alternative ICA results.
>
>
> > Q3: Why is it not possible to apply a more quantitative performance metric in the salsa experiment?
>
> We understand the importance of providing quantitative performance metrics and will aim to provide forecasting results for the salsa dataset by the rebuttal deadline.

---

> ### Author Response · Authors · 2023-11-22
> **Reminder for Reviewer SfP9**
>
> Dear Reviewer SfP9,
>
> Once again, we appreciate your time in reviewing our work. As the discussion period is finishing soon, we would be grateful if you could confirm whether our reply and modifications to the document addressed your concerns.

---

> ### Comment · Reviewer_SfP9 · 2023-11-22
>
> **Thank you for your rebuttal! I have raised my score to "slightly below acceptance" which will be my final score and I think reflects this paper well as overall I think the paper has some nice results but not enough for this conference. Either stronger theoretical results are needed or the same results with more convincing experiments section.**
>
> Details:
> - the rebuttal has convinced me that there is more novelty in identifying e.g. the transition parameters of the MSM/SDS and that that this wasn't done to this general level in prior work
> - nevertheless, there is a lot of overlap from several previous works. e.g. while it's true that the results don't follow fully from Kivva, a large part of the paper does. The authors also claim "novel proof strategy" but I disagree still, sure there are parts that are perhaps novel. But outside of Kivva, but parts of the proofs have similarities to Khemakhem; Allman; Gassiat; Morioka.
> - further: while there is some new identifiability for the SDS/MSM transitions, the identifiability of latent variables is much worse actually than in previous works in nonlinear ICA as I mentioned. I have previously been critical of Kivva for this same reason. Assuming known variance distribution is not a reasonable assumption for a work in identifiability theory. Other problematic assumption is that the of the piecewise linear nonlinearity. The authors claim *"Kivva et al. (2022) argues piece-wise linear functions already have universal function approximation capabilities"*. I have always find this argumentation to be faulty: universal approximation only holds at limit and we know from previous theory (Hyvarinen and Pajunen) that identifiability fails at this very limit -- thus the results do not cover general "f(x)" and in fact it's very hard to say where it fails and where it doesn't (e.g. what if f is a Gaussian Process?)
> - experimental work is limited and not convincing enough, as discussed, and it's hard to change one's opinion without seeing new results etc.

---

> > ### Author Response · Authors · 2023-11-23
> > **Response to reviewer**
> >
> > We sincerely appreciate your reply and your re-assessment in terms of our paper contributions.
> >
> > As promised in our earlier reply, we have updated the manuscript with additional quantitative metrics for the salsa experiment in the appendix. Furthermore, we have include visualisations of forecasts generated by our model to showcase that despite the restrictions we assume, our iSDS serves well as a sequential generative model in such high-dimensional scenarios.
> >
> > We would also like to inform that we added results with a new dataset based on real dancing videos ([AIST Dance DB](https://aistdancedb.ongaaccel.jp/)). We hope this additional metrics and datasets can serve to improve the experimental section of our work.

---

> > ### Author Response · Authors · 2023-11-23
> > **Some more clarifications regarding our "novelty" claims**
> >
> > Thank you for your reply and recognition for our MSM/SDS theoretical contributions.
> >
> > While we may have disagreement in turns of using the framework of Kivva et al. (2022), and we respect your different opinions, we'd like to point out that our argument for novel & significant theoretical contributions still stands.
> >
> > Consider the following logic for our arguments:
> > 1. Kivva et al. (2022) result requires the prior on the latent continuous variable distribution to be an identifiable finite mixture. This means, without proving MSMs as identifiable finite mixture models, we cannot proceed on the SDS results.
> >
> > 2. Now for proving the identifiability of MSMs, we emphasise again our technique does not follow any of the techniques in Khemakhem; Allman; Gassiat; Morioka. Here are the differences:
> > - 2a) Khemakhem et al. (2020) assumes an exponential family prior conditioned on observed auxiliary variables $u$ (therefore not a finite mixture model) -- instead, our MSM prior does not include observed auxiliary variables.
> > - 2b) Allman et al. (2009) and Gassiat et al. (2016): as explained Allman et al. (2009) technique does not work, and our method has nothing to do with Kruskal's "three-way arrays". Gassiat et al. (2016) is a direct extension of Allman et al. (2019) to continuous observations, thereby inheriting the key limitations of "three-way arrays" proof strategies.
> > - 2c) Morioka et al. (2021) uses HMMs as the latent prior $p(z_{1:T})$ while we are using MSMs. Therefore the model classes considered in Morioka et al. and our submission are very different, so do the proof techniques.
> >
> > We look forward to hear your further concerns (if any) regarding our theoretical contributions for MSMs.

---

### Author Response · Authors · 2023-11-22
**Review Summary Statement by the authors**

We thank all the reviewers for their time and feedback on our work.

We would like to summarize this reviewing process by combining the reviewers’ assessments of our work in combination with some of our concerns.


**Novelty and significance**
- We prove identifiability of first-order Markov Switching Models using a **novel proof strategy** by giving conditions for the **nonparametric** case (in Appendix) and showcasing the **nonlinear Gaussian** case in the main text.
  - The reviewers generally acknowledge our identifiability result.
  - We are concerned that novelty and significance is not appreciated. **SfP9**: “ one could have assumed the results here to be already implied based on earlier works: the approach for proving MSM using mixtures model presentation is well known from the general work in [1]”. We consider this statement is not correct, as we believe the result for nonparametric MSMs cannot be derived from Allman et al. (2009). See our reply for an explanation.

[1] Allman et al.(2009) Identifiability of parameters in latent structure models with many observed variables

- We provide the **first identifiability proof** for non-linear Switching Dynamical Systems, in terms of transition and emission functions up to affine transformations. We are again worried that this has also not been acknowledged by the reviewers. We reflect on their assessments below.
  - **SfP9**: “Identifiability of SDSs is thus achieved in their work” (referring to Halva et al. (2021). We believe this is not an accurate assessment, as Halva et al. (2021) does not prove identifiability of the transition functions.
  - **JVt8**: “The results are a direct extension of Kivva et al. (2022)”. We consider this is not a fair summary of our contribution. Our extension to SDSs uses Kivva et al. (2022) building on top of the previous MSM results. Since our first result on iMSMs is non-trivial, one cannot assume SDS identifiability follows directly from Kivva et al. (2022).
- Below are some concerns in terms of assumptions:
  - **SfP9**: “Why do you not instead apply e.g. the results from Halva et al (2021), Theorem 1, that allows any arbitrary noise distribution?” While assuming a specific noise distribution can be strong, recent ICA results (e.g. Halva et al. (2021) cannot be applied to our case. Such works assume stationary temporal dependencies, while we allow a nonstationary MSM.
  - **nPz5**: “The conditions of identifiability such as invertibility are strong and thus less nontrivial, and would limit the practicality”. While invertibility is an assumption considered **only in the emission** of SDSs, recent works that study identifiability also assume invertibility. Moreover, we also allow weakly-injective functions (e.g. ReLU networks) thanks to Kivva et al. (2022) results. Regarding practicality, Kivva et al. (2022) argues piece-wise linear functions already have universal function approximation capabilities.

**Experiments**

- To follow up our theoretical contributions, we perform experiments on **synthetic data** for iMSMs and iSDSs. This is acknowledged by some reviewers. **QaMu**: “The study's findings are empirically verified with synthetic experiments”
- We summarize the reviewer’s assessments on real data experiments:
  - Reviewers generally appreciate experimenting with real datasets. **SfP9**: “It is also nice to see experiments, albeit somewhat limited (see below), on real data despite the theoretical focus of the paper”. However, they are generally not convinced by the justification of an identifiable model at this point. **nPz5**: “The empirical studies lack demonstration of identifiability or the benefits of identifiability”. We argue that identifiable models allow us to provide interpretability from the learned representations, as long as there is no model misspecification.
    - For causal discovery, we attempt to motivate this by confirming whether the learned representation agrees with the expert knowledge.
    - For videos, the approach is different. As it is difficult to perform interpretability analysis in this setting, the objective is to show that although identifiable models are more restrictive in terms of dependencies and assumptions, they are still competitive in segmentation.
- Some reviewers (**nPz5**) acknowledge the discovery of regime-dependent structures as a strength. Considering **identifiability in causal discovery for regime-dependent structures is not addressed by the literature**, we are concerned as this has not been acknowledged in the reviewing process. We would like to note that our approach allows regime-dependent causal discovery thanks to having access to the function derivatives in an iMSM.

---

> ### Author Response · Authors · 2023-11-23
> **Changes to revised manuscript**
>
> All the changes in the manuscript during this rebuttal period appear in red. We summarise our changes below.
>
> ---
>
>
> We changed the term inference to discovery when referring to causality. (raised by **QaMu**).
>
> ---
>
> We rephrased our claims on causal discovery to ensure we do not pursue (or achieve) structure identifiability (raised by **QaMu**).
>
> ---
>
> We corrected acknowledgment of previous work regarding ICA which cover SDSs by focusing on the assumptions of stationarity, and incorporated Morioka et al. (2021) [1] in the related work  (raised by **SfP9**).
>
> ---
>
> We clarified notations in the background section, and the initial part of section 3: "We drop the subscript $\theta$ for simplicity" (raised by **SfP9**).
>
> ---
>
> We modified Figure 2 to improve its quality (raised by **SfP9**).
>
> ---
>
> We clarify that for neural networks, identifiability is focused on the functions rather than network parameters (raised by **JVt8**).
>
> ---
>
> We incorporated **forecasting results and forecasting samples for the salsa video dataset** in the appendix (Appendix F.6), following suggestions from **SfP9**: "Why is it not possible to apply a more quantitative performance metric in the salsa experiment?". We believe this additional experiment shows the validity of iSDS in generating forecasts despite the restrictions implied by the assumptions that enable identifiability. New Figures 11 and 13 are included (pages 31 and 34 respectively). The previous Figure 11 is now Figure 12.
>
> ---
>
> We included results for a **new dataset concerning real dancing videos from the [AIST Dance DB](https://aistdancedb.ongaaccel.jp/)**, to further strengthen our empirical results. The new results are in Appendix F.7, and the change is reflected in the main text with the corresponding reference. New Figure 14 is included (page 35)
>
> - We believe this result is significant as it shows segmentation results for high-dimensional real data using state-space models. Our iSDS comes from a family of methods (e.g. [2], [3]), where applications in such challenging domains have not been addressed.
>
> ---
>
> [1] Morioka et al. (2021) Independent Innovation Analysis for Nonlinear Vector Autoregressive Process
>
> [2] Dong, Z., Seybold, B., Murphy, K. and Bui, H., 2020, November. Collapsed amortized variational inference for switching nonlinear dynamical systems. In International Conference on Machine Learning (pp. 2638-2647). PMLR.
>
> [3] Ansari, A.F., Benidis, K., Kurle, R., Turkmen, A.C., Soh, H., Smola, A.J., Wang, B. and Januschowski, T., 2021. Deep explicit duration switching models for time series. Advances in Neural Information Processing Systems, 34, pp.29949-29961.

---

### Meta-Review · Area_Chair_aGyZ · 2023-12-06

**Metareview:**

This paper establishes the identifiability of switching dynamical systems, especially focusing on their latent variables and non-linear mappings, and develops estimation algorithms for their practical application. The paper also includes empirical studies demonstrating the use of these models in high-dimensional time series analysis and causal discovery in climate data. The paper's strengths lie in its clear presentation and the expansion of our understanding of identifiability in these models. The identifiability shown for a general class of SDSs is noteworthy, especially as it does not assume independence of the latent components, diverging from the approach in the nonlinear ICA area. Additionally, the inclusion of experiments on real data, despite the paper's theoretical focus, adds to its value​​. However, there are significant weaknesses that overshadow these strengths. The conditions of identifiability, such as invertibility, are quite strong, which makes them less nontrivial and potentially limits the practicality of the approach. This limitation is critical as it narrows the applicability of the models in realistic scenarios. Another major issue is the lack of demonstration of identifiability or the benefits of identifiability in the empirical studies. This gap between the theoretical advancements and their practical demonstration weakens the overall impact of the paper​​. Considering these aspects, while the paper contributes to the theoretical understanding of identifiability in SDSs, the limitations in terms of practical applicability and empirical validation are significant. The strong theoretical conditions and the lack of convincing empirical evidence demonstrating the benefits of these identifiability results in practical scenarios lead to concerns about the paper's overall impact and relevance to the field. Therefore, I recommend rejecting this paper.

**Justification For Why Not Higher Score:**

while the paper contributes to the theoretical understanding of identifiability in SDSs, the limitations in terms of practical applicability and empirical validation are significant. The strong theoretical conditions and the lack of convincing empirical evidence demonstrating the benefits of these identifiability results in practical scenarios lead to concerns about the paper's overall impact and relevance to the field.

**Justification For Why Not Lower Score:**

N/A

---

### Decision · Program_Chairs · 2024-01-16

Reject